# Modulating adsorbed hydrogen drives electrochemical $CO_2$-to-$C_2$ products

Jiaqi Feng[1], Libing Zhang [1,2], Shoujie Liu [3], Liang Xu[1], Xiaodong Ma[1,2], Xingxing Tan[1,2], Limin Wu[1,2], Qingli Qian [1,2], Tianbin Wu[1,2], Jianling Zhang[1,2], Xiaofu Sun [1,2] ✉ & Buxing Han [1,2,4] ✉

Electrocatalytic $CO_2$ reduction is a typical reaction involving two reactants ($CO_2$ and $H_2O$). However, the role of $H_2O$ dissociation, which provides active *H species to multiple protonation steps, is usually overlooked. Herein, we construct a dual-active sites catalyst comprising atomic Cu sites and Cu nanoparticles supported on N-doped carbon matrix. Efficient electrosynthesis of multi-carbon products is achieved with Faradaic efficiency approaching 75.4% with a partial current density of 289.2 mA cm$^{-2}$ at −0.6 V. Experimental and theoretical studies reveal that Cu nanoparticles facilitate the C-C coupling step through *CHO dimerization, while the atomic Cu sites boost $H_2O$ dissociation to form *H. The generated *H migrate to Cu nanoparticles and modulate the *H coverage on Cu NPs, and thus promote *CO-to-*CHO. The dual-active sites effect of Cu single-sites and Cu nanoparticles gives rise to the catalytic performance.

Heterogeneous catalysis holds great promise for practical applications in both the petrochemical and fine chemical industries, primarily due to its ease of catalyst separation and recovery[1,2]. However, it remains challenging when two or multiple reactants coexist in the system. To achieve satisfactory catalytic performance, dual-/multi- active sites catalysts have attracted great attention recently[3,4]. These catalysts possess multiple active sites that endow them with improved catalytic properties, enabling the activation of two or more reactants, which provide more efficient reaction pathway and accelerate the entire reaction, leading to the significantly enhanced activity and selectivity for the target products.

As one of the representative reactions that involves two reactants ($CO_2$ and $H_2O$) and multiple protonation steps, the electrocatalytic $CO_2$ reduction reaction ($CO_2$RR) to chemicals and fuels offers a promising route to store renewable energy and manage the global carbon balance[5–8]. Substantial efforts have been dedicated to developing dual-active sites catalysts to enhance $CO_2$RR performance. Many reported catalysts consist of active sites responsible for $CO_2$ activation (e.g., Au,

Ag and Zn) and further hydrogenation or C-C coupling (Cu). It has been demonstrated that the intermediate migration (e.g., *CO) plays a crucial role on the selectivity of the reaction, which greatly influenced by the optimal distribution and distance between the dual active sites. Meanwhile, an increase in the *CO coverage over Cu sites has been shown to promote the formation of $C_{2+}$ products[9–11]. In addition, considering $H_2O$ molecule serves as the proton source for hydrogenation step, the active site for $H_2O$ dissociation has been designed to be part of the dual-active sites catalysts[12,13]. The introduction of $H_2O$ dissociation sites, such as metal single atoms, sulfur, and oxygen vacancies, has been demonstrated to accelerate the rate of $H_2O$ dissociation into *H species, which are subsequently fed to $CO_2$ conversion sites through *H spillover[14–16]. However, the existing findings have mainly focused on the influence of $H_2O$ dissociation in $CO_2$-to-$C_1$ product process (formate or methane), while it is of greater significance for the $C_2$ products formation process. This is due to the commonly used alkaline electrolyte in $CO_2$-to-$C_2$ product, which slows down the $H_2O$ dissociation step, resulting in a sluggish reaction kinetic

[1]Beijing National Laboratory for Molecular Sciences, Key Laboratory of Colloid and Interface and Thermodynamics, Center for Carbon Neutral Chemistry, Institute of Chemistry, Chinese Academy of Sciences, Beijing 100190, China. [2]School of Chemical Sciences, University of Chinese Academy of Sciences, Beijing 100049, China. [3]School of Materials Science and Engineering, Anhui University, Hefei 230601, China. [4]Shanghai Key Laboratory of Green Chemistry and Chemical Processes, School of Chemistry and Molecular Engineering, East China Normal University, Shanghai 200062, China. ✉e-mail: sunxiaofu@iccas.ac.cn; hanbx@iccas.ac.cn

process[15,17]. Therefore, the constructing dual-active sites electro-catalyst to accelerate $CO_2$ reduction and $H_2O$ dissociation respectively is a feasible strategy for achieving the desired electrochemical $CO_2$-to-$C_2$ products performance.

Cu is the most promising electrocatalyst for converting $CO_2$ into $C_2$ products, owing to its moderate adsorption capacity for key intermediates[18–20]. Various strategies have been investigated to improve the $C_2$ selectivity and current density, such as the manipulation of crystal facets, morphology, particle size, and oxidation state[21–23]. However, most of these studies mainly focus on regulating the Cu structure for $CO_2$ activation and C-C coupling, with limited reports on the impact of surface *H coverage through accelerated $H_2O$ dissociation. It is crucial to carefully control the coverage of *H species since an excessive amount promotes $H_2$ production through the competing hydrogen evolution reaction (HER). Recently, single-atom catalysts, characterized by the uniform dispersion of isolated metal atoms on a substrate, have received significant attention in the field of HER due to their high metal utilization, tunable electronic structures and structural stability. Notably, the HER activity of single-atom Cu catalyst in alkaline electrolyte can be regulated through modifying coordinated environment and support type[24–26]. Therefore, designing of a dual-active sites catalyst incorporating co-loaded Cu nanoparticles (NPs) and single-atom Cu sites offers the potential for simultaneous $CO_2$ conversion and $H_2O$ activation, favoring the high-performance for electroproduction of $C_2$ products.

Herein, we have constructed a dual-active sites catalyst containing moderate content ratio (0.25) of atomic Cu sites to Cu NPs (M-Cu$_1$/Cu$_{NP}$) for $CO_2$ electroreduction to $C_2$ products. The M-Cu$_1$/Cu$_{NP}$ exhibited a high $C_2$ products Faradaic efficiency (FE$_{C2}$) of 75.4% with corresponding a partial current density of $C_2$ products ($j_{C2}$) of 289.2 mA cm$^{-2}$ at −0.6 V versus reversible hydrogen electrode (vs. RHE, all potentials are referenced to RHE), Moreover, the stable $CO_2$-to-$C_2$ products conversion with FEs of >70% can maintain at a constant current density of 400 mA cm$^{-2}$ for a run time of 40 h. The experiments and density functional theory (DFT) calculations revealed that Cu NPs facilitated $CO_2$ activation, *CO hydrogenation into *CHO, and C-C coupling into $C_2$ products. The atomic Cu site accelerated $H_2O$ dissociation to provide *H, and the generated *H transferred to Cu NPs through N-doped carbon matrix and modulated the *H coverage on Cu NPs, which reduced the energy barrier for *CO to *CHO.

## Results

### Synthesis and characterization of M-Cu$_1$/Cu$_{NP}$ catalyst

The M-Cu$_1$/Cu$_{NP}$ catalyst was synthesized through a facile pyrolysis strategy by calcining Cu complexes in Ar flow and then reducing in $H_2$ flow. The Cu complexes were obtained by mixing $Cu(NO_3)_2 \cdot 3H_2O$ and guanidine thiocyanate with a 1:8 mole ratio in deionized water and subsequently evaporating the solvent. By varying the amount of $Cu(NO_3)_2 \cdot 3H_2O$, the catalysts containing the rich (R-Cu$_1$/Cu$_{NP}$) and poor (P-Cu$_1$/Cu$_{NP}$) amount of Cu$_1$ site were also prepared using the same procedure. The morphology of the as-prepared catalyst was characterized by transmission electron microscopy (TEM). The dark-field TEM image (Fig. 1a) shows that obvious nanoparticles with the average size of ~4 nm are uniformly dispersed on carbon matrix in M-Cu$_1$/Cu$_{NP}$. The corresponding high-resolution (HR) TEM image of the nanoparticle displays the distinct lattice fringes with an interplanar space of 0.208 nm, consistent with the (111) plane of metallic Cu (Fig. 1b), confirming the uniformly dispersed nanoparticles are Cu NPs. The dispersion of atomic Cu was then identified by the aberration-corrected high-angle annular dark-field scanning transmission electron microscopy (HAADF-STEM). As presented in Fig. 1c, the isolated bright dots can be recognized as well distributed Cu atoms in M-Cu$_1$/Cu$_{NP}$, which have been highlighted by red circles. The energy-dispersive X-ray spectroscopy (EDS) elemental mapping images (Fig. 1d) reveal a uniform distribution of C, N and Cu, indicating that

they are homogeneous over the entire architectures. Therefore, we can conclude that a dual-active sites catalyst containing atomically dispersed Cu sites adjacent to uniformly dispersed Cu NPs was successfully prepared. In addition, the TEM and aberration-corrected HAADF-STEM images of P-Cu$_1$/Cu$_{NP}$ and R-Cu$_1$/Cu$_{NP}$ are shown in Fig. S1 and Fig. S2, respectively. The TEM images showed that both P-Cu$_1$/Cu$_{NP}$ and R-Cu$_1$/Cu$_{NP}$ exhibited a similar average size of Cu NPs compared to M-Cu$_1$/Cu$_{NP}$, while the mount of Cu NPs obviously decreased over R-Cu$_1$/Cu$_{NP}$. Meanwhile, evident single atomic Cu can also be observed on HAADF-STEM images of P-Cu$_1$/Cu$_{NP}$ and R-Cu$_1$/Cu$_{NP}$. The X-ray diffraction (XRD) patterns of P-Cu$_1$/Cu$_{NP}$, M-Cu$_1$/Cu$_{NP}$ and R-Cu$_1$/Cu$_{NP}$ in Fig. 1e show that only the characteristic peaks of metallic Cu and carbon were observed, and the strong peak at 43.2° can be assigned to Cu(111) facet, which is consistent with the HRTEM results. The X-ray photoelectron spectroscopy (XPS) survey spectra show that the presence of Cu, N, C and O elements in P-Cu$_1$/Cu$_{NP}$, M-Cu$_1$/Cu$_{NP}$ and R-Cu$_1$/Cu$_{NP}$ (Fig. S3), where the existence of O element can be attributed to the adsorbed oxygen on surface. The Cu content in P-Cu$_1$/Cu$_{NP}$, M-Cu$_1$/Cu$_{NP}$, and R-Cu$_1$/Cu$_{NP}$ was determined to be 7.4, 4.3, and 1.9 at%, respectively, and the corresponding N and C content are displayed in Table S1. The Cu 2$p$ spectra (Fig. 1f) suggest that only Cu 2$p_{3/2}$ peak attributed to Cu$^{0/+}$ existed in the spectra, which moved to lower binding energy in the order of R-Cu$_1$/Cu$_{NP}$ (933.0 eV), M-Cu$_1$/Cu$_{NP}$ (932.7 eV) and P-Cu$_1$/Cu$_{NP}$ (932.5 eV), indicating a gradual increase in the valence state of Cu in the catalysts. The Cu LMM Auger spectra confirmed the coexistence of Cu$^+$ and Cu$^0$ in P-Cu$_1$/Cu$_{NP}$ M-Cu$_1$/Cu$_{NP}$ and R-Cu$_1$/Cu$_{NP}$ (Fig. S4). Considering that the valance state of Cu in Cu NP is 0, while that of atomically dispersed Cu site is Cu$^{\delta+}$, the content of Cu$_1$ site in the catalyst changed in order of P-Cu$_1$/Cu$_{NP}$ < M-Cu$_1$/Cu$_{NP}$ < R-Cu$_1$/Cu$_{NP}$. Meanwhile, the high resolution XPS N 1$s$ spectra of P-Cu$_1$/Cu$_{NP}$, M-Cu$_1$/Cu$_{NP}$, and R-Cu$_1$/Cu$_{NP}$ are displayed in Fig. S5. All spectra showed obvious Cu-N peak at around 399.2 eV, which suggests that the Cu$_1$ sites were coordinated by N atoms and other peaks could be attributed to pyridinic N (-398.5 eV), pyrrolic N (-399.9 eV), and graphitic (-401.0 eV), respectively.

To further determine the chemical state and local coordination environment, X-ray absorption near-edge structure (XANES) and extended X-ray absorption fine structure (EXAFS) measurements were conducted. The Cu K-edge XANES of R-Cu$_1$/Cu$_{NP}$, M-Cu$_1$/Cu$_{NP}$ and P-Cu$_1$/Cu$_{NP}$ with the reference samples Cu foil and Cu phthalocyanine (Cu Pc) were presented in Fig. 1g, the adsorption edge of all catalysts located between that of Cu foil and Cu Pc, indicating the average Cu valence state of Cu in the catalysts laid between 0 and +2. Meanwhile, the adsorption edge shifted to higher energy in the sequence of P-Cu$_1$/Cu$_{NP}$ < M-Cu$_1$/Cu$_{NP}$ < R-Cu$_1$/Cu$_{NP}$, confirming that the average Cu valence state increased with the increase in the Cu$_1$/Cu$_{NP}$ ratio. It is in accordance with the XPS results. The Fourier-transformed (FT) EXAFS spectra in the R-space of the three catalysts were processed and displayed in Fig. 1h, all three catalysts exhibited peaks at around 1.5 Å and 2.2 Å, attributed to Cu-N and Cu-Cu coordination, respectively. Therefore, both atomic Cu and metallic Cu NP coexisted in R-Cu$_1$/Cu$_{NP}$, M-Cu$_1$/Cu$_{NP}$ and P-Cu$_1$/Cu$_{NP}$. Notably, the intensity ratio of the Cu-N peak to the Cu-Cu peak increased in the order of P-Cu$_1$/Cu$_{NP}$ < M-Cu$_1$/Cu$_{NP}$ < R-Cu$_1$/Cu$_{NP}$, indicating a gradual increase in the content ratio of atomic Cu to Cu NP. In order to obtain the local structure of Cu species and the content ratio of atomic Cu to Cu NP, the quantitative analysis by the least-squares EXAFS fittings was performed (Fig. S6). The percentages (P) of Cu-N$_4$ (Cu$_1$) and Cu NPs in total Cu species of catalysts were displayed in Table S2. Thus, the content ratio of Cu$_1$ to Cu$_{NP}$ of P-Cu$_1$/Cu$_{NP}$, M-Cu$_1$/Cu$_{NP}$ and R-Cu$_1$/Cu$_{NP}$ were 0.05, 0.25 and 0.39, respectively, which is consistent with our catalyst design expectation.

### Electrocatalytic $CO_2$RR performance

The $CO_2$RR performance of the as-prepared catalysts were first evaluated by linear sweep voltammetry (LSV) in 5 M KOH aqueous solution

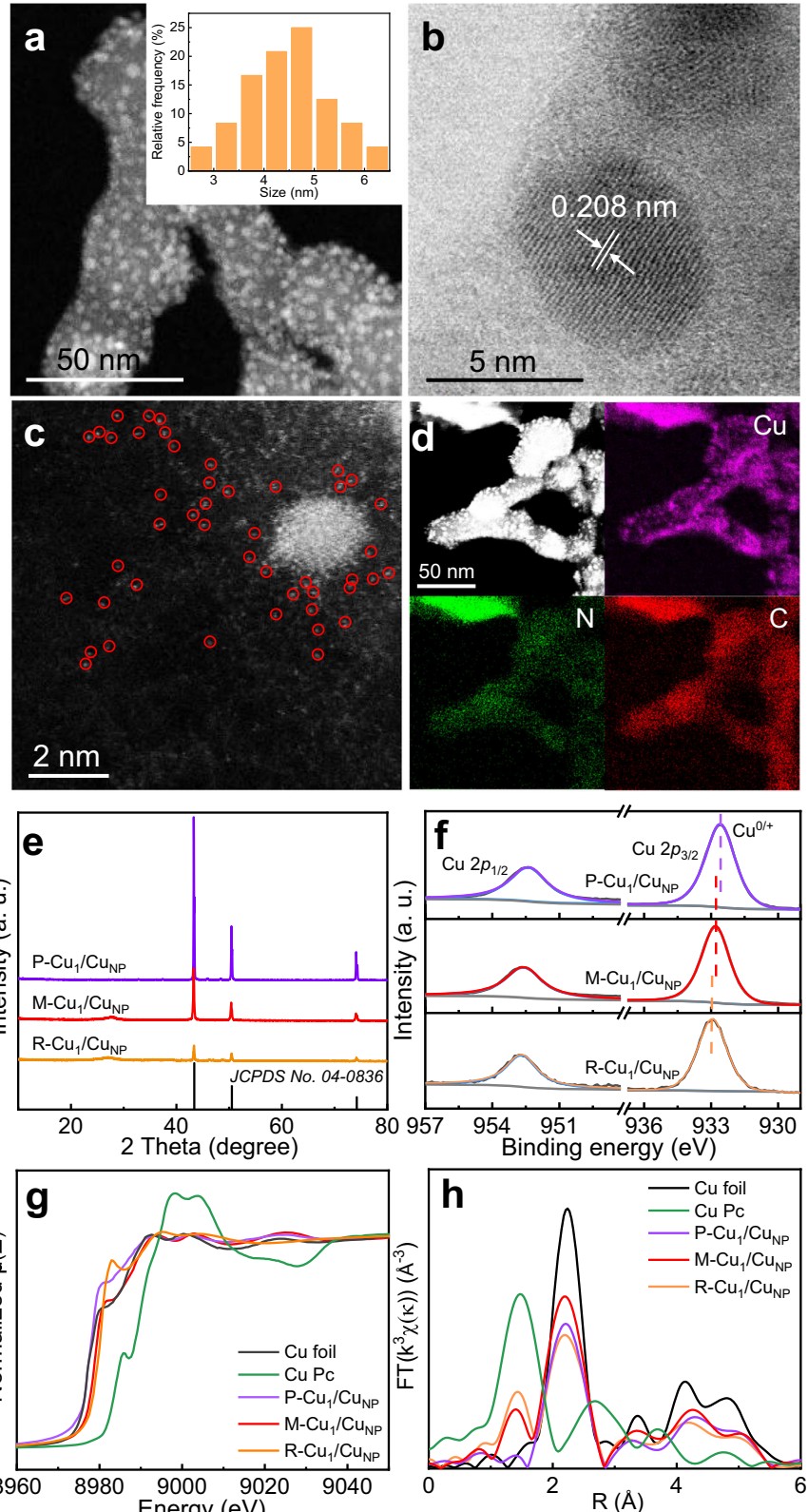

**Fig. 1 | Morphology and structure of the catalysts. a** DF-TEM image (inset shows particle size distribution), **b** HRTEM image, **c** aberration-corrected HAADF-STEM image, and **d** EDS element mappings of M-Cu₁/Cu_NP. **e** XRD patterns, **f** XPS spectra of Cu $2p$ orbits, **g** Cu K-edge XANES spectra, and **h** FT-EXAFS spectra of P-Cu₁/Cu_NP, M-Cu₁/Cu_NP and, R-Cu₁/Cu_NP.

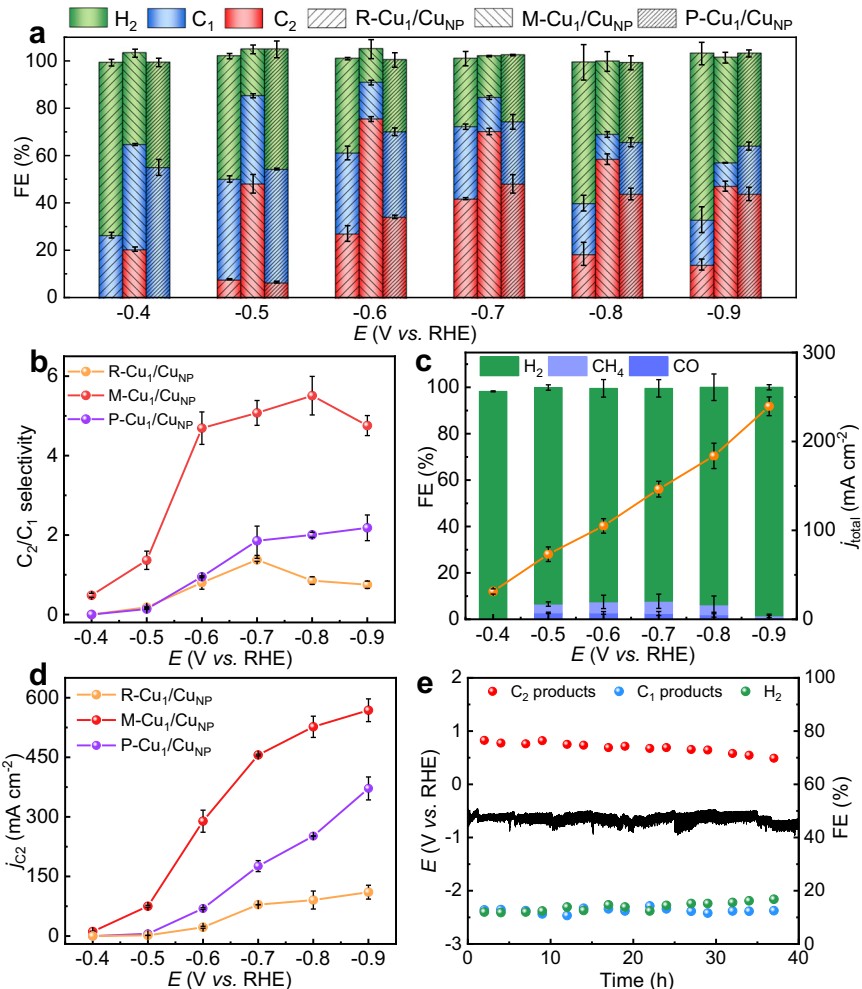

**Fig. 2 | CO₂ electroreduction performance. a** The FEs of C₁ products, C₂ products, H₂, and **b** C₂ and C₁ product selectivity ratio at different potentials over the as-prepared catalysts in 5 M KOH electrolyte. **c** The product FEs and total current density of Cu-N-C at different potentials in 5 M KOH electrolyte. **d** The C₂ partial current density at different potentials over the as-prepared catalysts in 5 M KOH

electrolyte. Values are means and error bars indicate s.d. (*n* = 3 replicates). **e** Long-term stability of M-Cu₁/Cu_NP at a constant current density of 400 mA cm⁻² in 5 M KOH electrolyte (the electrode was washed, then dried and the electrolyte was refreshed at intervals 5 h to address the issues of flooding and carbonation).

in a flow cell (Fig. S7). As shown in Fig. S8, all the catalysts exhibited significant increase in cathodic current density when the feed gas was change from N₂ to CO₂, indicating that CO₂RR occurred over R-Cu₁/Cu_NP, M-Cu₁/Cu_NP and P-Cu₁/Cu_NP. Meanwhile, M-Cu₁/Cu_NP exhibited the most positive onset potential and highest cathodic current density in the presence of CO₂ gas, and taking −0.8 V as an example, the current density reached up to 1093.0 mA cm⁻² over M-Cu₁/Cu_NP, which was roughly 2.5 and 1.7 times higher than that of R-Cu₁/Cu_NP and P-Cu₁/Cu_NP, respectively. Thus, the results of LSV experiments suggested that M-Cu₁/Cu_NP has higher CO₂ activity than R-Cu₁/Cu_NP and P-Cu₁/Cu_NP, which could be attributed to the moderate content ratio of Cu₁ to Cu_NP.

The control potential electrolysis was then performed to analyze the reduction products and the catalyst loading was 1 mg cm⁻² (Fig. S9). The gas-phase and liquid-phase products were analyzed by gas chromatography and ¹H nuclear magnetic resonance spectroscopy, respectively. H₂, CO, CH₄, formate, C₂H₄, ethanol and acetate were formed. Figure 2a shows that the FE of C₂ products (FE_C2) of M-Cu₁/Cu_NP exhibited a volcano-shaped dependence on the applied potential (Fig. S10, Tables S3−S5), and a maximum FE_C2 could reach 75.4% at −0.6 V, which is much higher than that over R-Cu₁/Cu_NP and P-Cu₁/Cu_NP. The maximum FE_C2 of P-Cu₁/Cu_NP was 47.3% at −0.7 V, which closed to the performance of Cu nanoparticles reported in the

literature[22,27]. Moreover, the M-Cu₁/Cu_NP had a lower onset potential for C₂ products formation. The FE_C2 could reach 20.5% at −0.4 V over M-Cu₁/Cu_NP, while C₂ products cannot be detected over R-Cu₁/Cu_NP and P-Cu₁/Cu_NP under the same potential. The ratio of FE_C2 to FE_C1 could keep >4.5 from −0.6 to −0.9 V over M-Cu₁/Cu_NP, while those of R-Cu₁/Cu_NP and P-Cu₁/Cu_NP were below 2 in the whole applied potential range (Fig. 2b). Furthermore, M-Cu₁/Cu_NP was treated by sulfuric acid solution to completely remove Cu NPs, resulting in a catalyst denoted as Cu-N-C. The results of TEM, XRD, XPS and XAS indicated that only Cu single atoms existed in the Cu-N-C catalyst (Figs. S11, S12). H₂ was the dominant product over Cu-N-C in the whole applied potentials range, suggesting that the atomic Cu sites mainly facilitated H₂O dissociation (Fig. 2c and Table S6). The formation of CO and CH₄ was also observed, in agreement with previous reports that Cu single atoms could generate H₂, CO, and CH₄[25,28,29]. These results suggest a dual-active sites effect between Cu sites and Cu NPs, and the proper ratio of Cu₁ to Cu_NP would obviously enhance the selectivity for C₂ products.

From FE_C2 and total current density (Fig. S13), the C₂ products partial current density (j_C2) of catalysts at different potentials was obtained and shown in Fig. 2d. It could reach 289.2 mA cm⁻² over M-Cu₁/Cu_NP at −0.6 V vs. RHE, exceed by up to 13.1-fold and 4.2-fold than that on R-Cu₁/Cu_NP (22.1 mA cm⁻²) and P-Cu₁/Cu_NP (69.3 mA cm⁻²), respectively. Besides, a high total current density of 1207.6 mA cm⁻²

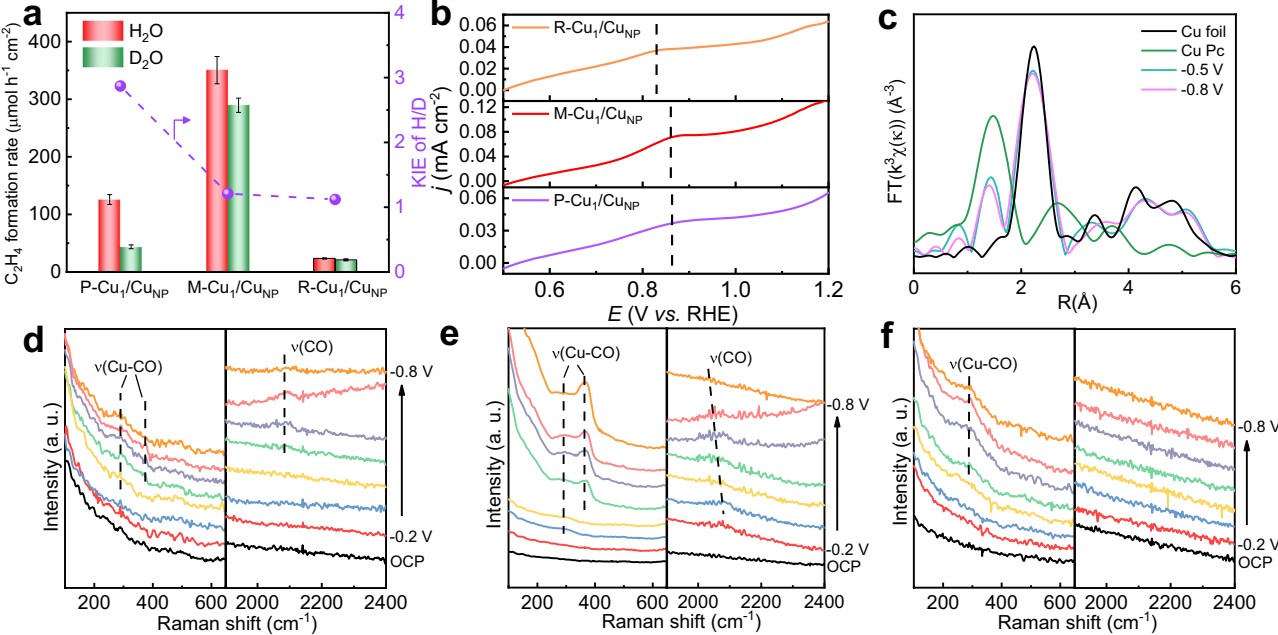

**Fig. 3 | The mechanism analysis. a** The kinetic isotope effect (KIE) of $H_2O/D_2O$ on P-Cu$_1$/Cu$_{NP}$, M-Cu$_1$/Cu$_{NP}$ and R-Cu$_1$/Cu$_{NP}$ at −0.6 V in flow cell with 5 M KOH electrolyte. **b** The electrochemical CO stripping voltammetry tests of P-Cu$_1$/Cu$_{NP}$, M-Cu$_1$/Cu$_{NP}$ and R-Cu$_1$/Cu$_{NP}$ in 0.1 M Na$_2$SO$_4$ electrolyte. **c** In situ FT EXAFS spectra at Cu K-edge over M-Cu$_1$/Cu$_{NP}$ at various potentials in 5 M KOH electrolyte. In situ surface-enhanced Raman spectra recorded at different applied potentials for (**d**) P-Cu$_1$/Cu$_{NP}$, (**e**) M-Cu$_1$/Cu$_{NP}$ and (**f**) R-Cu$_1$/Cu$_{NP}$ during CO$_2$RR in 5 M KOH electrolyte.

was obtained at −0.9 V over M-Cu$_1$/Cu$_{NP}$, delivering a FE$_{C2}$ of 47.1% with a partial current density of 568.5 mA cm$^{-2}$. We further normalized the $j_{C2}$ on the basis of the electrochemical active surface area (ECSA), which was measured by the double-layer capacitance (C$_{dl}$) method (Fig. S14). Even after normalization, M-Cu$_1$/Cu$_{NP}$ still exhibited the highest normalized $j_{C2}$ among the catalysts (Fig. S15), confirming that the intrinsic catalytic activity of M-Cu$_1$/Cu$_{NP}$ is higher than that of P-Cu$_1$/Cu$_{NP}$ and R-Cu$_1$/Cu$_{NP}$. Moreover, we also increased the loading of P-Cu$_1$/Cu$_{NP}$ and R-Cu$_1$/Cu$_{NP}$ in accordance with the ECSA results, ensuring the catalytic activity were measured under similar ECSA condition. The results suggested that M-Cu$_1$/Cu$_{NP}$ maintained the largest C$_2$ products selectivity and partial current density (Fig. S16), further confirming that the intrinsic catalytic activity of M-Cu$_1$/Cu$_{NP}$ exceeds that of both P-Cu$_1$/Cu$_{NP}$ and R-Cu$_1$/Cu$_{NP}$. Compared with reported electrocatalysts for CO$_2$RR to C$_{2+}$ products in the literature, M-Cu$_1$/Cu$_{NP}$ exhibited higher CO$_2$RR to C$_{2+}$ products activity at low applied potential, especially in term of C$_{2+}$ partial current density (Table S7).

The long-term stability test was conducted through chronopotentiometry for 40 h and the electrode was washed, then dried and the electrolyte was refreshed at intervals 5 h to address the issues of flooding and carbonation. Fig. 2e and Fig. S17 showed that no obvious change was observed over the potential and products selectivity, the high FE of C$_2$ products could be hold over 70% on M-Cu$_1$/Cu$_{NP}$ during the electrolysis. The morphology, valence states and crystal structure of M-Cu$_1$/Cu$_{NP}$ after 40 h CO$_2$RR experiment were characterized (Figs. S18, S19). TEM image suggested the size Cu NPs is similar to that before used and the aberration-corrected HAADF-STEM image confirmed that Cu$_1$ site and Cu NPs still coexisted in M-Cu$_1$/Cu$_{NP}$. The Cu LMM Auger spectra show that Cu$^+$ and Cu$^0$ coexisted in M-Cu$_1$/Cu$_{NP}$. Meanwhile, no obvious difference was observed in N 1$s$ XPS spectra between before and after CO$_2$RR experiment, and only the peaks corresponding to Cu were observed in XRD spectrum after CO$_2$RR experiment. All these results indicated the excellent stability of the catalyst.

The kinetic isotope effect (KIE) of $H_2O/D_2O$ (H/D) experiments were performed to get insights into the role of $H_2O$ dissociation in

CO$_2$-to-C$_2$ products (Fig. 3a). When $H_2O$ was replaced by $D_2O$ in the electrolyte, the formation rate of the product (e.g., C$_2$H$_4$) decreased over P-Cu$_1$/Cu$_{NP}$, M-Cu$_1$/Cu$_{NP}$ and R-Cu$_1$/Cu$_{NP}$, and the level of decrease closely related to the content ratio of Cu$_1$/Cu$_{NP}$. If the KIE value (defined as the ratio of C$_2$H$_4$ formation rates in $H_2O$ and $D_2O$) closes to 1, $H_2O$ dissociation is not the rate-determining step over the catalyst. The KIE values for R-Cu$_1$/Cu$_{NP}$, M-Cu$_1$/Cu$_{NP}$ and P-Cu$_1$/Cu$_{NP}$ were 1.12, 1.21 and 2.87, respectively, which indicated that $H_2O$ dissociation was accelerated gradually with an increasing Cu$_1$/Cu$_{NP}$ content ratio. These results confirmed that the atomic Cu sites were responsible for accelerating $H_2O$ dissociation and provided proton to adjacent Cu NPs, thus affecting CO$_2$-to-C$_2$ products. Meanwhile, the N-doped carbon matrix catalysts has been reported to favor the migration of proton[30]. In addition, the H$_2$ formation rate in 5 M KOH $H_2O$ solution and $D_2O$ solution was presented in Fig. S20. An obvious decrease was observed over all the catalysts, while the decreasing degree followed the sequence of P-Cu$_1$/Cu$_{NP}$ > M-Cu$_1$/Cu$_{NP}$ > R-Cu$_1$/Cu$_{NP}$, suggesting that the increase of Cu single atom content can reduce the influence of isotope effect on H$_2$ formation rate.

Furthermore, we studied the influence of electrolyte pH on CO$_2$RR performance over R-Cu$_1$/Cu$_{NP}$, M-Cu$_1$/Cu$_{NP}$ and P-Cu$_1$/Cu$_{NP}$. Three different concentrations of KOH aqueous electrolytes, i.e., 0.1 M, 3 M and 5 M, were employed to adjust the pH environment at the electrode/electrolyte interface. The results in Fig. S21 show that the FE and formation rate of C$_2$ products increased with the increasing pH value of electrolyte, suggesting that strong basic local environment favored the C$_2$ products formation. Although CO$_2$RR activity of M-Cu$_1$/Cu$_{NP}$ in low concentration KOH electrolytes was lower than that in 5 M KOH electrolyte, M-Cu$_1$/Cu$_{NP}$ exhibited higher FE and formation rate of C$_2$ products than P-Cu$_1$/Cu$_{NP}$ and R-Cu$_1$/Cu$_{NP}$ in low concentration KOH electrolytes, which suggested that the cooperative effect of Cu NPs and atomic Cu sites still facilitated C$_2$ products formation. The ratio of C$_2$ products formation rate over M-Cu$_1$/Cu$_{NP}$ to P-Cu$_1$/Cu$_{NP}$, i.e., Rate$_M$/Rate$_P$, in 5 M KOH (3.4) is higher than those in 3 M KOH (3.0) and 0.1 M KOH (2.3). This demonstrated that the role of atomic Cu sites in accelerating the $H_2O$ dissociation process was significantly more

pronounced at higher pH values, leading to the enhanced $C_2$ products formation, even if the dissociation of $H_2O$ in higher pH electrolyte was a sluggish step.

We then investigated the capacity adsorption and activation of $CO_2$ and CO molecules on R-$Cu_1$/$Cu_{NP}$, M-$Cu_1$/$Cu_{NP}$ and P-$Cu_1$/$Cu_{NP}$ via gas electro-response experiments in a self-designed gas adsorption electro-response device (Fig. S22)[31]. The results in Fig. S23 and Fig. S24 show that the $CO_2$ and CO adsorption responses changed in the sequence of R-$Cu_1$/$Cu_{NP}$ < M-$Cu_1$/$Cu_{NP}$ < P-$Cu_1$/$Cu_{NP}$, suggesting that the adsorption and activation of $CO_2$ and CO were promoted as the ratio of $Cu_1$ to $Cu_{NP}$ decreased. On the other hand, the electrochemical CO stripping voltammetry tests of R-$Cu_1$/$Cu_{NP}$, M-$Cu_1$/$Cu_{NP}$ and P-$Cu_1$/$Cu_{NP}$ were also performed to study the CO adsorption ability. Figure 3b shows a peak in the potential range of 0.8–0.9 V for all three catalysts. No peak was observed in the LSV curve without CO adsorption (Fig. S25). According to prior literature[32–34], the peak at around 0.8–0.9 V can be attributed to the CO stripping peak. Interestingly, the CO stripping peak occurred at around 0.89 V for M-$Cu_1$/$Cu_{NP}$ and P-$Cu_1$/$Cu_{NP}$, while it was around 0.82 V for R-$Cu_1$/$Cu_{NP}$. The positive shift suggested that M-$Cu_1$/$Cu_{NP}$ and P-$Cu_1$/$Cu_{NP}$ had a stronger CO binding ability than R-$Cu_1$/$Cu_{NP}$, which have higher Cu NP content. The results above indicated that the Cu NPs was beneficial for the adsorption and activation of $CO_2$ and CO.

In situ X-ray absorption spectroscopy (XAS) experiments were performed on M-$Cu_1$/$Cu_{NP}$ to investigate the changes of Cu valence state and structure during $CO_2$RR (Fig. S26). In the XANES spectra (Fig. S27), the Cu K-edge adsorption spectra did not show obvious difference under different potentials, indicating that the average Cu valence state kept stable during the reaction. Moreover, the peaks corresponding to Cu-N and Cu-Cu coordination still existed in FT EXAFS spectra (Fig. 3c), with no notable change was observed in peak intensity, suggesting the stability of the content ratio of $Cu_1$ to $Cu_{NP}$. The results of in situ XAS experiments showed that the structure and content of $Cu_1$ and $Cu_{NP}$ remained stable during $CO_2$RR.

In situ surface-enhanced Raman spectroscopy (SERS) was employed to reveal the interactions between Cu species and the reaction intermediates during $CO_2$RR (Fig. S28)[35–38]. As shown in the Raman spectra under different potentials (Fig. 3d–f), the peaks assigned to the restricted rotation of adsorbed CO (298 cm$^{-1}$, v(Cu-CO)), Cu-CO stretching (365 cm$^{-1}$, v(Cu-CO)) and C-O stretching of atop *CO (2000–2100 cm$^{-1}$, v(CO)) were observed from −0.3 to −0.8 V over M-$Cu_1$/$Cu_{NP}$. In contrast, the v(Cu-CO) and v(CO) peaks over P-$Cu_1$/$Cu_{NP}$ appeared from −0.4 V and −0.5 V, respectively. These observation could be related to the lower onset potential for $C_2$ products formation over M-$Cu_1$/$Cu_{NP}$. Only CO rotation peaks can be observed on R-$Cu_1$/$Cu_{NP}$ from −0.4 to −0.8 V, meaning poor $CO_2$ reduction activity. In situ SERS was also conducted over Cu-N-C (Fig. S29), whereas no peaks associated with *CO intermediate were observed, suggesting that single atomic Cu was not conducive to the conversion of $CO_2$ to adsorbed CO. The results of in situ SERS indicated that $CO_2$-to-$C_2$ products proceeded via CO intermediate process and a moderate content ratio of $Cu_1$ to $Cu_{NP}$ facilitated the conversion of $CO_2$ to adsorbed CO.

Online differential electrochemical mass spectrometry (DEMS) was conducted to investigate reaction mechanisms (Fig. S30), which extracts volatile intermediates and products generated on the electrode surface into the mass spectrometer within milliseconds, utilizing the pressure difference in a vacuum as the driving force[39,40]. During five continuous cycles at −0.6 V vs. RHE, the m/z signal of 29 that correspond to CHO could be detected over M-$Cu_1$/$Cu_{NP}$, while it was absent in P-$Cu_1$/$Cu_{NP}$ and R-$Cu_1$/$Cu_{NP}$ (Fig. S31). The results of online DEMS demonstrated that the crucial role of the proper $Cu_1$ to $Cu_{NP}$ content ratio in CHO formation. Meanwhile, in situ attenuated total reflection-surface-enhanced IR absorption spectroscopy (ATR-SEIRAS) spectra were further collected to trace the evolution of reaction intermediates during $CO_2$RR from −0.2 to −0.9 V over R-$Cu_1$/$Cu_{NP}$, M-$Cu_1$/$Cu_{NP}$ and P-$Cu_1$/$Cu_{NP}$ (Figs. S32, S33). The peak at around 2100 cm$^{-1}$ can be attributed to electrogenerated CO adsorbed (*CO) on catalyst surface[41,42], which did not appear until the applied potential reached −0.5 V on R-$Cu_1$/$Cu_{NP}$ and P-$Cu_1$/$Cu_{NP}$. However, the v(*CO) peak over M-$Cu_1$/$Cu_{NP}$ obviously existed from −0.3 to −0.9 V and the peak intensity was stronger than that of R-$Cu_1$/$Cu_{NP}$ and P-$Cu_1$/$Cu_{NP}$, suggesting that M-$Cu_1$/$Cu_{NP}$ preferred to generate *CO intermediate. Meanwhile, a peak at 1748 cm$^{-1}$ observed over M-$Cu_1$/$Cu_{NP}$ from −0.3 to −0.9 V can be ascribed to the *CHO intermediate[12,38,39], and its change trend was similar to that of *CO intermediate, indicating that the *CHO was originated from the hydrogenation of *CO with the assistance of *H. Nevertheless, the *CHO peak was not observed on R-$Cu_1$/$Cu_{NP}$ and P-$Cu_1$/$Cu_{NP}$, meaning that proper content ratio of $Cu_1$ to $Cu_{NP}$ is necessary for *CHO formation.

## DFT calculations

DFT calculations were performed to gain insights into the $CO_2$RR mechanism. According to the HRTEM and EXAFS fitting results, the Cu(111) facet and the Cu-$N_4$ were used as the models of Cu NPs and atomic Cu sites, respectively. All the computational structure models and the detailed data are shown in Figs. S34–S40. We first calculated $CO_2$ to adsorbed CO (*CO) process to verify if the catalyst is beneficial for $CO_2$ activation, as *CO is a crucial intermediate for $C_2$ products formation. The calculations revealed that $CO_2$ could be converted to *CO via a *COOH intermediate. However, the energy barrier of $CO_2$-to-*CO over the Cu(111) facet and Cu-$N_4$ was found to be 0.44 and 1.40 eV, respectively (Fig. 4a and Fig. S35). This indicated that $CO_2$ preferred to be activated on Cu(111) facet, which is accordance with the experiment results.

In general, the C-C coupling reaction was considered as the rate-determining step for $CO_2$-to-$C_{2+}$ products[23,43]. Although $CO_2$ could be easily converted to *CO on Cu(111) facet, high uphill reaction energy was needed to form *O*CCO intermediate, meaning that the C-C coupling reaction through *CO dimerization is difficult. We further calculated the C-C coupling reaction through different $C_1$ species (*CO, *CHO, *COH). As shown in Fig. 4b, the uphill reaction energy for the *CHO formation was 0.65 eV lower than that of the *COH formation (1.07 eV). This indicated that the *CO preferred to hydrogenate into *CHO rather than *COH, which is consistent with the ATR-SEIRAS results. Afterwards, both *CHO-*CO and *COH-*COH coupling were endoergic, while *CHO dimerization to form *OHCCHO* intermediate was exergonic. Therefore, the hydrogenation of *CO into *CHO then coupling of *CHO into *OHCCHO* is the most favorable routes in C-C coupling reaction.

Considering that the *CHO intermediate was originated from the hydrogenation of *CO with the assistance of *H, the influence of *H coverage on *CHO formation over Cu(111) facet were investigated. Our calculations revealed that that increasing the *H coverage on Cu(111) up to 1/4 decreased the reaction energy of *CO hydrogenation to *CHO (Fig. 4c and Fig. S41). However, a further increase of the *H coverage to 1/2 disfavored the formation of *CHO. These findings confirm that a moderate *H coverage facilitated the formation of *CHO, thus promoted the C-C coupling reaction and improved the $CO_2$-to-$C_2$ products efficiency. The moderate atomic Cu sites in M-$Cu_1$/$Cu_{NP}$ can accelerate the $H_2O$ dissociation process in alkaline electrolyte to provide the moderate *H coverage on Cu NP surface (Fig. 4d), therefore, the M-$Cu_1$/$Cu_{NP}$ exhibited excellent $CO_2$RR to $C_2$ products performance.

## Discussion

In summary, a series of dual-active sites catalysts with co-loaded Cu NPs and atomic Cu sites on N-doped carbon matrix have been successfully synthesized and evaluated for $CO_2$ electroreduction. Among them, M-$Cu_1$/$Cu_{NP}$ exhibited a $C_2$ products FE of 75.4% with a

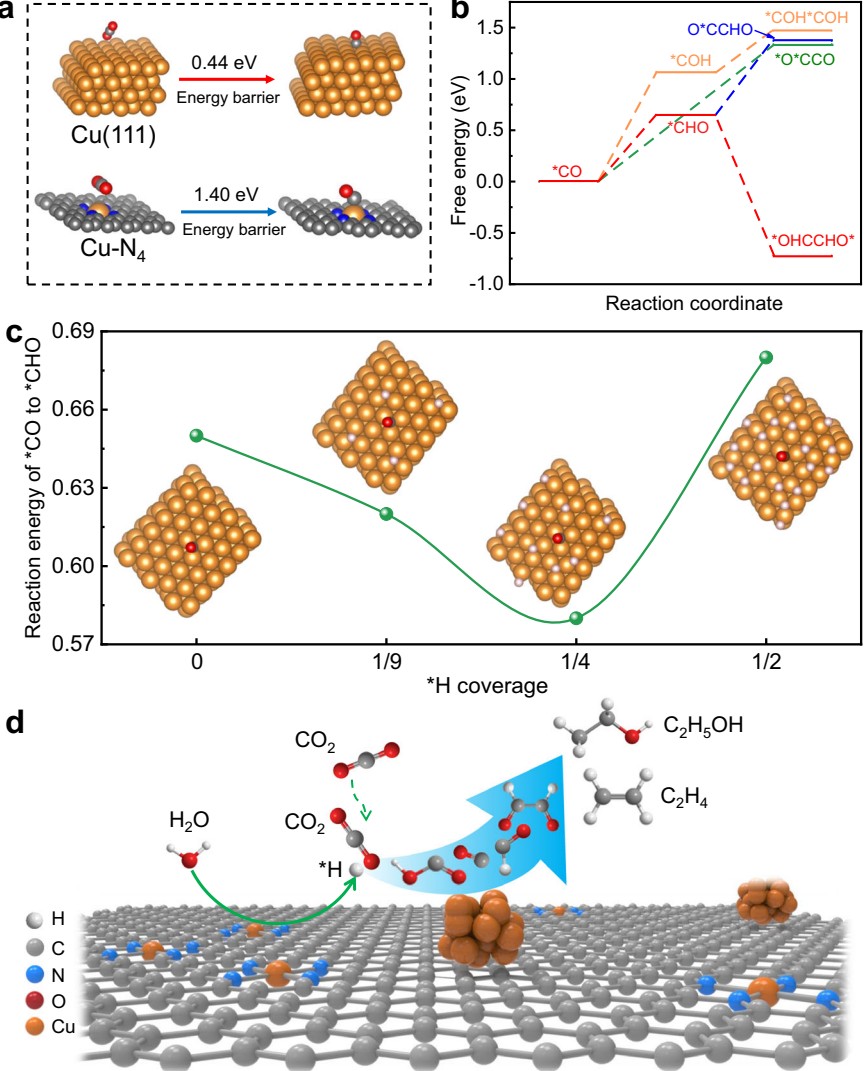

**Fig. 4 | The DFT calculations. a** The energy barrier of $CO_2$ was converted to *CO over Cu(111) and Cu-$N_4$. **b** The free energy diagram for the $CO_2$RR to describe the possible C−C coupling step from *CO over Cu(111). **c** Reaction energy of *CO hydrogenation to *CHO on Cu(111) as a function of *H coverage. **d** Proposed reaction mechanism for $CO_2$RR to $C_2$ products on the M-$Cu_I$/$Cu_{NP}$.

corresponding partial current density of 289.2 mA cm$^{-2}$ at −0.6 V, as well as remarkable long-term stability. The detailed study revealed that atomic Cu sites could promote $H_2O$ dissociation to provide *H, and the Cu NPs was beneficial for the adsorption and conversion of $CO_2$. In the in situ spectroscopic characterization showed that moderate content ratio of atomic Cu sites to Cu NPs facilitated $CO_2$ conversion to adsorbed CO, resulting in M-$Cu_I$/$Cu_{NP}$ possessing a lower onset potential for $C_2$ products. Furthermore, online DEMS and in situ ATR-SEIRAS revealed the presence of the intermediate *CHO over M-$Cu_I$/$Cu_{NP}$, which was originated from the hydrogenation of *CO with the assistance of *H. DFT calculations demonstrated that the C-C coupling step was promoted through *CHO dimerization reaction on Cu NPs, and the moderate *H coverage facilitated the formation of *CHO. Therefore, the excellent catalytic performance of as-fabricated dual-active sites catalyst originated from the dual-active sites effect of atomic Cu sites and Cu NPs. The atomic Cu sites promoted $H_2O$ dissociation to provide *H, which in turn migrated to Cu NPs and facilitated *CO protonation to form *CHO by modulating the *H coverage on Cu NPs, leading to high activity for $C_2$ products production. This work puts forward rational concept for promoting conversion $CO_2$ to $C_2$ products through modulate the adsorbed hydrogen coverage on Cu-based catalysts. We believe that it will inspire the design of more dual-/multi-active sites catalysts for multi-step reactions.

## Methods

### Materials
Copper nitrate hydrate (Cu(NO$_3$)$_2$·3H$_2$O, purity > 99%), guanidine thiocyanate (C$_2$H$_6$N$_4$S, purity > 99%), potassium hydroxide (KOH, purity > 85%), potassium bicarbonate (KHCO$_3$, purity > 99.5%), sodium sulfate (Na$_2$SO$_4$, purity > 99%), 2, 2-dimethyl-2-silapentane-5-sulfonate (DSS, 99%), Deuterium oxide (D$_2$O, purity > 99.9), gas diffusion electrode (YLS-30) with 10% PTFE and microporous layer, anion exchange membrane (FAA-3-PK-130) and Ni foam (purity > 99.8%, thickness 0.5 mmm) were purchased from Alfa Aesar China Co., Ltd. CO$_2$ (99.999%), N$_2$ (99.999%) and 10% H$_2$/Ar were provided by Beijing Analytical Instrument Company. All materials were used directly without further purification and all the aqueous solutions were prepared by Milli Q water (18.2 MΩ cm, 298 K).

### Catalysts characterization
The morphologies of as-synthesized catalysts were characterized by a JEOL JEM-2100F high-resolution transmission electron microscopy

(HR-TEM). The high-angle annular dark-field scanning transmission electron microscopy (HAADF-STEM) characterization and corresponding energy-dispersive spectroscopy (EDS) were conducted on aberration-corrected JEM-ARM300F operated at 300 kV. Powder X-ray diffraction (XRD) analysis of the samples were performed on the X-ray diffraction (Model D/MAX2500, Rigaka) with Cu-Kα radiation and the scattering range of 2θ was from 5° to 90°, with a scanning rate of 5° min⁻¹. X-ray photoelectron spectroscopy (XPS) analysis was performed on Thermo Fisher Scientific ESCA Lab 250Xi using 200 W monochromatic Al Kα (1486.6 eV) radiation under a pressure of $3 \times 10^{-10}$ mbar, and the binding energy was referenced to the C 1 s peak at 284.4 eV. The X-ray absorption data at the Cu K-edge of the catalysts were recorded in transmission mode using ion chambers at the 4B9A beamline of Beijing Synchrotron Radiation Facility (BSRF) and the radiation was monochromatized by a Si (111) double-crystal monochromator. All collected spectra were analyzed using Athena and Artemis program within the IFEFFIT software packages.

### Catalyst synthesis

50.8 mmol of guanidine thiocyanate and a quantity of $Cu(NO_3)_2 \cdot 3H_2O$ were added to a flask containing 50 mL deionized water with vigorously stirring at 80 °C. After stirring for 4 h, the temperature was raised to 100 °C to evaporated water. The collected solid product was grounded evenly and pyrolyzed at 600 °C with 10 °C min⁻¹ for 3.5 h under $N_2$ atmosphere, heated at 400 °C with 2 °C min⁻¹ for 2 h under 10% $H_2$/Ar atmosphere. After cooled down to room temperature, the as-obtained solid was washed with ethanol and deionized water for several times and dried in vacuum at 60 °C overnight. Specifically, for R-Cu₁/Cu_NP, M-Cu₁/Cu_NP, and P-Cu₁/Cu_NP, the amount of used $Cu(NO_3)_2 \cdot 3H_2O$ was 3.81, 6.35 and 8.89 mmol, respectively. For comparison, Cu-N-C was prepared by treating M-Cu₁/Cu_NP with acid treatment. The M-Cu₁/Cu_NP was added into 50 mL 1 M sulfuric acid aqueous solution and heated at 80 °C for 48 h, then washed with deionized water several times and dried at 80 °C overnight.

### Electrocatalytic CO₂ reduction

To prepared gas diffusion electrode, 1 mg catalyst was suspended in 500 μL isopropanol with 10 μL Nafion D-521 dispersion (5 wt%) to form a homogeneous ink. Then the catalyst ink was spread onto the gas diffusion electrode of $0.5 \times 2$ cm² in area by a micropipette to make sure the mass loading of the catalyst was 1 mg cm⁻², and then dried under room temperature. All the electrochemical experiments were conducted on the electrochemical workstation (CHI 660E) equipped with a high current amplifier (CHI 680 C) and the CO₂RR performance was investigated in flow cell (Fig. S7). The prepared gas diffusion electrode and Ni foam were used as the cathode and anode, respectively. An anion exchange membrane was used to separate the cathode and anode. Aqueous KOH solution (5 M) was used as the electrolyte solution and the electrolyte volume was 30 mL, and the electrolyte volume in the cell is 0.5 mL. The catholyte solution and anolyte solution were recirculated by two pumps with flow rates of 10 mL min⁻¹ and 30 mL min⁻¹, respectively. Meanwhile, CO₂ gas was continuously supplied to the gas chamber by using a mass flow controller with a flow rate of 40 mL min⁻¹. All the potentials were measured against a Hg/HgO reference electrode and converted to versus RHE with iR (80%) compensations, i represents the current obtained at corresponding potential, R is ohmic resistance of the cell measured by electrochemical workstation:

$$E_{RHE} = E_{Hg/HgO} + 0.098 + 0.059 \times pH - iR \times 80\% \qquad (1)$$

The chronopotentiometry method was used for evaluating long-term activity stability of electrocatalyst in 5 M KOH electrolyte. In order to address the issues of flooding and carbonation accumulation of GDE, the CO₂RR was interrupted every 5 h and the GDE were removed,

washed with deionized water and followed by dryness under $N_2$ atmosphere. Meanwhile, the electrolyte solution was refreshed for each interval.

### Product analysis

The gaseous products were collected using a gas bag and quantified by gas chromatography (GC 7890B). A thermal conductivity detector (TCD) and a flame ionization detector (FID) were used to quantify $H_2$, CO, and other alkane contents, respectively. The Faradaic efficiency (FE) of gaseous products was calculated by the equation:

$$FE = znVF/Q \times 100\% \qquad (2)$$

Where z represents the number of electrons transferred for product formation, n is the volume concentration from GC, V is the total volume calculated by outlet flow rate, F is Faraday constant (96485 C/mol) and the Q is the amount of cumulative charge recorded by the electrochemical workstation.

The liquid product was analyzed by ¹H NMR (Bruker Advance III 400 HD spectrometer) in deuteroxide. To accurately integrate the products in NMR analysis, the sodium 2, 2-dimethyl-2-silapentane-5-sulfonate (DSS) was the reference for ethanol and acetic acid, and phenol was the reference for formate. The FE of liquid products was calculated by the equation:

$$FE = znVF/Q \times 100\% \qquad (3)$$

Where z represents the number of electrons transferred for product formation, C is the liquid concentration obtained from NMR, V is electrolyte volume, F is Faraday constant (96485 C/mol) and the Q is the amount of cumulative charge recorded by the electrochemical workstation.

### Double layer capacitance measurements

The electrochemical active surface area is proportional to double layer capacitance, which measures the capacitive current associated with double-layer charging from the scan-rate dependence of cyclic voltammogram. The double layer capacitance was determined in a single-compartment electrolytic cell with 0.5 M KHCO₃ aqueous solution as electrolyte. Ag/AgCl eletrode and graphite rod were used as reference electrode and counter electrode, respectively. The scan rates of cyclic voltammogram were 20, 40, 60, 80, 100 mV s⁻¹.

### Gas electro-response experiments

A self-designed gas adsorption electroresponse device (Fig. S19) was used to perform gas electro-response experiments. To prepared the electrode, 5 mg catalyst was suspended in 1 mL isopropanol to form a homogeneous ink. Then the catalyst ink was spread onto the Cu foam with area of $1 \times 2$ cm² by a micropipette and then dried under room temperature. The as-prepared electrode was put into a sealed container and connected with electrochemical workstation though two electrode system. Before Ar, CO₂ or CO gas was injected into the sealed container, the container was kept in vacuum state by a vacuum pump. Various potentials were applied on the electrode to observed the change of current curve as a function of time under different atmosphere. The adsorption of various gas on the electrode surface would induce the change of current response. Considering the catalyst loading and the size of the Cu foam used in each experiment is the same, the difference of current density under Ar and CO₂ (or CO) atmosphere can reflect the adsorbed capacity of CO₂ (or CO) on the catalyst surface.

### CO stripping test

CO stripping experiments were conducted in a single-chamber electrolytic cell with three electrode system through linear sweep voltammetry method. To prepared working electrode, 1 mg catalyst was

suspended in 500 μL isopropanol with 10 μL Nafion D-521 dispersion (5 wt%) to form a homogeneous ink. Then the catalyst ink was spread onto the carbon paper of $1 \times 1$ cm$^2$ in area and then dried under room temperature. Ag/AgCl electrode and graphite rod were used as reference electrode and counter electrode, respectively. The electrolyte was 0.1 M Na$_2$SO$_4$ aqueous solution. The CO adsorption procedure was accomplished by polarizing the working electrode at +0.2 V and bubbling the electrolyte with CO for 10 min and subsequently with N$_2$ for another 10 min. Then the linear sweep voltammetry was conducted from 0.5 to 1.2 V with scan rate of 10 mV s$^{-1}$.

### In situ XAS measurements
A custom-designed flow cell was used to conducted in situ XAS measurements (Fig. S23), the gas diffusion electrode loaded with catalyst (1 mg cm$^{-2}$), Ni foam and Hg/HgO electrode were chosen as the working electrode, counter electrode and reference electrode, respectively. The 5 M KOH aqueous solution was used as electrolyte and were recirculated by pump with flow rates of 20 mL min$^{-1}$. CO$_2$ gas was continuously supplied to the gas chamber with a flow rate of 40 mL min$^{-1}$. Data were recorded in fluorescence excitation mode using a Lytle detector. During the in situ XAS experiments, the spectrum was recorded following the application of a corresponding potential to the electrode for a duration of 600 s. Beamline of Beijing Synchrotron Radiation Facility (BSRF) and the radiation was monochromatized by a Si (111) double-crystal monochromator. All collected spectra were analyzed using Athena and Artemis program within the IFEFFIT software packages.

### In situ Raman measurements
A flow cell with a quartz window by GaossUnion (Tianjin) Photoelectric Technology Company was used to carry out in situ Raman measurements using a Horiba LabRAM HR Evolution Raman microscope (Fig. S25). A 785 nm laser was used and signals were recorded using a 20 s integration and by averaging two scans. The gas diffusion electrode sprayed with the catalyst was used as working electrode and a graphite rod and a Hg/HgO electrode were used as counter and reference electrodes, respectively. The counter electrode was separated from the working electrode by anion exchange membrane. The 5 M KOH aqueous solution was used as electrolyte and were recirculated by pump with flow rates of 20 mL min$^{-1}$. Meanwhile, CO$_2$ gas was continuously supplied to the gas chamber with a flow rate of 40 mL min$^{-1}$.

### In situ ATR-SEIRAS measurements
The experiments were conducted in a modified electrochemical cell that integrated into a Nicolet 6700 FTIR spectrometer equipped with MCT detector cooled by liquid nitrogen (Fig. S27). The catalysts ink was dropped on a germanium ATR crystal deposited with Au film. A platinum wire and a Hg/HgO electrode were used as counter and reference electrodes. Each spectrum was collected with 32 times with a resolution of 4 cm$^{-1}$. The CO$_2$-saturated 3 M KOH aqueous solution was used as electrolyte to avoid that the germanium ATR crystal was damaged by strong basic electrolyte. The background spectrum was collected at the potential of +0.1 V vs RHE.

### Online DEMS measurement
A custom-designed flow cell was used to conducted online DEMS measurements (Fig. S30). The gas diffusion electrode loaded with catalyst, platinum wire and Hg/HgO electrode were chosen as the working electrode, counter electrode, and reference electrode, respectively. The 5 M KOH aqueous solution was used as electrolyte and were recirculated by pump with flow rates of 20 mL min$^{-1}$. CO$_2$ gas was continuously supplied to the gas chamber with a flow rate of 40 mL min$^{-1}$. A potentiostatic test at −0.6 V (vs. RHE) was performed in an alternating manner. Following the completion of each electrochemical test and the return of the mass signal to its baseline level, the subsequent cycle commenced, employing identical experimental conditions to minimize the possibility of inadvertent errors. The experiment was terminated after the completion of five cycles.

### DFT calculations
Density function theory calculation were performed by using the CP2K package[44]. PBE functional[45] with Grimme D3 correction[46] was used to describe the system. Unrestricted Kohn-Sham DFT has been used as the electronic structure method in the framework of the Gaussian and plane waves method[47,48]. The Goedecker-Teter-Hutter (GTH) pseudopotentials[49,50], DZVPMOLOPT-GTH basis sets were utilized to describe the molecules. A plane-wave energy cut-off of 500 Ry has been employed. The potential-dependence of reaction free energies in elementary steps involving proton-electron transfers was evaluated using the computational hydrogen electrode (CHE) approach[51].

The Gibbs free energy is calculated using:

$$\Delta G_{free} = \Delta E_{DFT} - T\Delta S + \text{ZPE} \tag{4}$$

Where $\Delta G_{free}$ is Gibbs free energy, $\Delta E_{DFT}$ is energy calculate from DFT, $S$ is entropy, $T$ is temperature (300 K), and ZPE is zero-point energy.

## Data availability
The data that support the plots within this paper are available in the Source data file. Additional data available from authors upon request. Source data are provided with this paper.

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

## Acknowledgements

The work was supported by National Natural Science Foundation of China (22203099, 22002172, 22033009, 21890761, 22121002), Beijing Natural Science Foundation (J210020), China Postdoctoral Science Foundation (2022M713200), S&T Program of Hebei (B2021208074), and Photon Science Center for Carbon Neutrality. The X-ray absorption spectroscopy measurements were performed at Beamline 4B9A at Beijing Synchrotron Radiation Facility (BSRF).

## Author contributions

J.F., X.S., and B.H. proposed the project, designed the experiments, and wrote the manuscript; J.F. performed the whole experiments; S.L. assisted in analysing the experimental data; L.Z., L.X., X.M., X.T. and L.W. conducted a part of characterizations. Q.Q., T.W., and J.Z. participated in discussions. X.S. and B.H. supervised the whole project.

## Competing interests

The authors declare no competing interests.
