## [Peer Review File · Nature Communications]

Modulating adsorbed hydrogen drives electrochemical
CO₂-to-C₂ productsREVIEWER COMMENTS

Reviewer #1 (Remarks to the Author):

In this work, Feng et al reported a dual-active sites catalyst comprising atomic Cu sites and Cu nanoparticles for CO₂RR. Among them, the Cu nanoparticles facilitated the C-C coupling step by *CHO dimerization, and the atomic Cu sites boosted H₂O dissociation to form *H and further migrated to Cu nanoparticles. As a result, the catalyst shows the CO₂-to-C₂+ performance with a FE of 75.4%. Some concerns remain regarding the suitability of the authors' conclusions, for which the reviewer would like to see additional data to address and then can justify whether this work is suitable for publication.

Detailed comments are given below:

1. I am confused about the Figure 3B. The CO stripping peak is determined at approximately -0.7 V vs SHE (Nature Catalysis 2020, 3, 797–803). The peak around 0.8-0.9 V vs RHE should not be classified as CO stripping.
2. It is uncertain that why the author chose different reference samples in the tests. M-Cu₁/CuNP was compared with P-Cu₁/CuNP in KIE experiment, but compared with R-Cu₁/CuNP in CO stripping experiment. Why not compare the three samples together?
3. The assignment of C≡O Raman bands at 2000-2100 cm⁻¹ is problematic. In fact, the CObridge appears 1830-1900cm⁻¹. (ACS Central Science 2016, 2, 522-528, The Journal of Physical Chemistry C 2017, 121, 12337-12344, ACS Catal. 2018, 8, 7507-7516, ACS Catal. 2019, 9, 6305-6319, Nat. Commun. 2022, 13, 2656.)
4. It is not clear about the details of the in-situ Raman/ATR-SEIRAS process. What electrolyte did the authors use for in-situ tests? The CO₂RR performance of the catalysts were evaluated in KOH aqueous solution. If the authors also used alkaline electrolytes for the in situ experiments, why did the CO₃²⁻ signal appear?
5. The authors compared the catalytic performance with other literatures (Figure 2D), but the performances of this work are not outstanding, especially in alkaline electrolytes. Moreover, this kind of comparison does not help much with the advance of the research community, as when researchers get used to them, they just target higher metrics but ignoring science and insights behind the experiments.
6. Regarding catalytic properties, the reviewer is not sure whether the authors have adopted rigorous protocol to ensure the accuracy of CO₂RR performance data. The outlet flowrates could be very different from the inlet flowrates due to the reaction between CO₂ and KOH, leading to problematic FE data of the gas products.

Reviewer #2 (Remarks to the Author):

The authors present a multi-active site catalyst by mixing Cu Nanoparticles and atomic Cu sites, both supported on an N-doped carbon matrix. Additionally, they have wisely chosen to additionally prepare catalysts comprising only Cu NPs or Cu atomic sites. The catalysts were thoroughly prepared and fully characterized. The CO₂ reaction reaction was performed in alkaline media, in a flow cell, focusing on the detection of C₂ products. A wide variety of insitu/operando techniques have been used to understand the evolution of the catalysts under reaction conditions. The authors claim a mixture of CuNP and Cu atomic sites to be more beneficial than both individual systems comprising a synergistic effect. By characterizing the catalysts in the as prepared state and during reaction, the authors claim to find that C-C coupling occurs mostly on CuNP, while H₂O dissociation appears mostly at atomic Cu sites, *H would migrate to the CuNP and thus promote the protonation of *CO.

To achieve these results, the authors have comprehensively used different techniques to understand the catalysts under reaction conditions. The authors have started strongly demonstrating the interesting nature of the catalysts and characterizing them well in the pristine state. The data sets and their analysis of the measured data under reaction conditions or after reaction is, however, incomplete. The data for at least one catalyst for most of the presented insitu/operando techniques is not presented. Thus, the interpretation is weakly supported on the basis of these investigations. It is suggested to integrate the missing data and to revise the presented interpretation in this manuscript majorly.

INTRODUCTION

The introduction should be revised providing more information and state-of-the-art knowledge about multiactive sites for multiple reactants in CO₂RR (L33). A lot of studies have been performed to study the role of particle size, distance or shape. A more through summary on the literature should be presented with a focus on the relevant parameters for this work (L49-51). The cited literature should show similar reaction systems, e.g. in L56, the authors claim H₂O activation on Cu single atoms, but the literature presents their data in an acidic system, while the authors study alkaline systems. A discussion why this result like this is transferable, is thus required. A specification on the ratio of CuNP compared to Cu single atoms should be given already in L60.

TEM ANALYSIS

The Cu nanoparticle size is a relevant parameter for the catalytic performance. It would thus be relevant to mention the obtained sizes in the main text and present particle size distribution for all catalysts. The authors may also provide a NP size distribution for the R-Cu1/CuNP as they have nicely done for the other two samples. Additionally a more exact measure for the ratio between Cu NP and single atom sizes from the TEM data should be given and discussed against the EXAFS fitted data and the observed percentages between the coordination numbers. In this regard, the HAADF-STEM images showing the single atom Cu sites are missing in Figure S1 and S2 and are asked to be provided.

XPS ANALYSIS

Please comment further on the observed differences and its relevance in the binding energies of the Cu2p_{3/2} peaks (L94). Additionally, fits that indicate the small amounts of Cu²⁺ species in the M and P catalysts should be presented. The authors also may provide an analysis of the Auger peaks for distinguishing the metallic Cu and the Cu¹⁺ oxidation states of the catalysts, if possible. The content of Cu should be set into context with the content of C and N in the catalysts (L95), and any impurities should be ruled out (e.g. sulfur from guanidine thiocyanate). The analysis of N1s for the other two catalysts should be presented (Figure S4).

XES ANALYSIS

The authors may provide a motivation why they used this technique and how it would be helpful to understand the catalysts better.

L103 & Figures 1F and 1G: The authors may comment why their references are so different from literature (see e.g. Vegelius et al., *J. Anal. At. Spectrom.*, 2012,27, 1882).

L104-105 & Figure 1G: The authors may also comment why their data quality is so low compared to their mentioned reference literature and what they could significantly learn from the XES experiments, which they did not learn from XAS or XPS.

XAS ANALYSIS

The XAS analysis has been thoroughly performed. However, the descriptive text starting from L106 is not clear. E.g., in L108, it is unclear, which catalyst is meant. As much as the Cu valence state increases in a sequence it is more vital in the reviewers opinion, that the P- and the R-catalyst show very similar intensity transients than the two references, indicating that both could be used to represent the reference catalysts. Please also comment on the artifact(?) peaks around 1 Å (L112, Figure 1I) and on the calculation of the percentage between Cu NP and single Cu sites (L117). Please also comment on the maximum observed coordination number of the Cu NPs and if an estimation of the particle size can be extracted from it.

ELECTROCATALYTIC CO₂RR PERFORMANCE

The authors present very interesting and carefully studied CO₂ reduction reaction results. The P- and R-catalysts should however be set into the context of literature, as Cu NPs and Cu single atoms are already well studied.

How were the presented LSVs in Figure S6 normalized? The reviewer assumes that a normalization to the ECSA results, which are presented in Figure S10, would change the results and thus the interpretation. A thoughtful analysis of the electrochemical surface area is necessary here to evaluate the observed current densities. Additionally, Figure S6, and in general for most Figures, specifically the electrochemistry figures, lack a complete description of the used electrolyte, scan rate and normalization, which should be provided.

It is unclear, which figure is described here in L136.

The data in Figure 2A are nicely evaluated. Please also add the obtained total currents to the graphs. For transparency, please also add a table with the obtained values and errorbars (or add them to the plots). Describe in the experimental data how many datapoints are comprised in one errorbar.

The additional synthesis procedure for the Cu-N-C catalyst should be described in the experimental data

and in the figure descriptions Figures S7-S9). It looks unorganized if this catalyst is only introduced here, but not in the general catalyst characterization in Figure 1. It is proposed that this Cu-N-C catalyst is used as a 4th main catalyst, and the whole characterization is also performed on this one. Please comment on the huge H₂ production on the Cu-N-C and the much better performance of the R-Cu₁/CuNP (L145). Please also relate the results of Cu-N-C to literature. Discuss that Cu-N-C catalysts should in principle hamper any C-C coupling, but still small amounts of ethylene are formed around -0.6V.

The discrepancy of the obtained results from DL Capacitance are huge (L145). If the M- and P-catalysts would be normalized to the R-catalyst, roughness factors of 1, 2 and 5 would emerge (R, M, P). It is thus not enough to just normalize the CO₂RR data to these values due to many other parameters that could bias the results (loading, particle distances, agglomeration, ...). Please justify your results by remeasuring one or two potentials of your catalysts with similar ECSA (e.g. by lowering the loading).

Please establish your choice to measure chronopotentiometry for the long-term stability test instead of chronoamperometry (L155). Discuss the long-term stability results on the results of the other two (or three catalysts). How would you expect that the R- and P-catalysts behave during 40h? Please also show the results of the other products, especially hydrogen. Unfortunately, the presented TEM, XPS and XRD data (Figures S12-S14) after the long-term stability test are not analyzed. For the TEM, an evaluation of the NP size is lacking, as well as an HAADF-STEM image showing the presence and amount of the single atom sites. The authors should provide the same type of data, that they have nicely provided in Figure 1A-D. XPS: An evaluation of the oxidation state (Cu2p with fits and, if possible Auger data) and chemical composition (Cu/C/N ratio) should be given. Differences, e.g. of the N1s composition change before and after reaction should be discussed. Here, the CO₂RR conditions should be described in the descriptions of these Figures. Similar for all shown data after reaction. Please specify how the regular refreshing of the catalyst could influence the stability and product distribution (L164). This important experimental detail should be noted in the experimental data, and maybe even in the main text as it may be deceptive for the reader of this manuscript. Please discuss and demonstrate that flooding might not alter the catalyst, but that carbonation does and that it can be reversed.

KINETIC ISOTOPE EFFECT

The authors provide a decent study on the origin of the hydrogen used for C₂ products. These results could be justified by KIE results of R-Cu₁/CuNP and Cu-NC (L172). The authors should provide these investigations. It should also be discussed how the hydrogen formation is altered with D₂O experiments. Again, please specify the experimental conditions in Figure 3. (potential/scan rate, electrolyte, CO₂ saturation).

The authors also present a small study on the influence of the electrolyte concentration on the formation of ethylene. Please describe why the formation rate instead of the FE or partial current densities are chosen here for the analysis and how it justifies the interpretation. Results for R-Cu₁/CuNP should also be given. The potential and electrochemical details should be provided in Figure S15.

ELECTRORESPONSE & CO Stripping

Discuss why the used Cu foam as support does not take part in the experiment. Discuss the relevance of

catalyst loading and ECSA for this experiment. Please provide the results for the missing catalysts. The shown results indicate intrinsic differences between the catalysts, but could also very likely be just a roughening/loading effect. Please add experimental details also in the experimental section and add a scheme of the device (Figure S17). Similarly, please provide experimental details of the CO stripping voltammetry in the experimental section. Explain, how the CO was adsorbed on the surface and please add the LSV of the double layer without the CO stripping peak. Again, please also provide the results for the missing catalyst.

The interpretation of the electroresponse and CO stripping experiments are vague, and need clarification, providing that the "reference catalyst" P-Cu1/CuNP was left out. Please set the results in the context of literature.

OPERANDO XAS:

The shown results for operando XAS are incomplete and the interpretation is based on weak presented data.

The authors should quantify the increase of the Cu valence state for example with Linear-Combination fitting (L194). It is not clear, how long the catalyst was set at -0.6V before the shown spectrum was measured. It could very likely be that the experiment did not perform as wished. Different (experimental) parameters can play a role, why the catalyst did not reduce completely or oxidizes again. Parameters that could show a working system could be the following: i) current similar to the experimental conditions in the lab ii) similar product detection iii) different potentials to see the evolution of the catalyst. Please discuss. Thus the assumption towards adsorbed CO₂ and H₂O on the basis of the shown data is unfortunately highly overinterpreted. Please show the data of several potentials and justify with the other catalysts e.g. that Cu NPs can reduce completely, and/or that Cu single atoms behave different.

L196: The authors should also provide the least-square fittings (L196) and clarify how you attribute/correlate the increased Cu-N/O coordination number to adsorbed CO₂ and H₂O (L198). It is quite hard from EXAFS data to see the local coordination of adsorbed species on the data. In some cases, however, it is evaluable from XANES data or from XES data.

INSITU-SERS and ATR-SEIRAS

The authors should specify the CO stretching and CO rotation in L201 and provide the results of the other two catalysts (L203). It would be intriguing to see how the CO adsorption changes for the other catalysts. What did the authors learn from these experiments? It is well known that CO is an intermediate for CO₂ reduction. Would the COads be lacking for the R-Cu1/CuNP or the Cu-N-C catalyst? Are differences in the CO onset potential seen that would be relatable to the earlier onset for C₂+ products for the M-catalyst?

The interpretation on the experiments of ATR-SEIRAS are weak without the measurements of the other catalysts. The authors should provide complete data sets of the other catalysts.

DISCUSSION & SUMMARY

The discussion is weak, providing that the assumptions were mostly made on the basis of just one sample. Mixing Cu NPs and Cu atomic sites require good knowledge and verification of both individual

systems. The manuscript has started well by presenting three different samples and analyzing the electrocatalytic data of each quite well. However, some more work has to be done on the spectroscopic site to understand the catalytic systems and to extract a relevant conclusion from it.

EXPERIMENTAL

In general, the experimental section lacks several details that are important to follow the experiments. E.g., different types of flow cells exist, a more explicit description should be provided. If a gas-fed CO₂ type flow cell was used, please also add the side from which the gas products were determined and how often gas products were injected into the GC (if an online-GC was used). Please provide a scheme or a reference of the used cell. Please state type and company of membrane and used chemicals, type of the used gas diffusion electrode was used (type, content of PTFE, with or without microporous layer), type of the Ni foam. Please state the total volume of electrolyte (re)circulated, and how big the electrolyte volume in the cell is.

The authors may also provide more details of the electrolyte (purity of the water, purity and company of the KOH, possible presaturation with CO₂). Experimental details on the ohmic resistance should be added. The authors may also provide the experimental details the double layer capacitance measurements. Ni foam as counter electrode not the state of the art to be used as counter electrode for CO₂RR, the authors may comment on their choice. The authors may also comment on their choice of and Hg/HgO reference electrode instead of other stable and non-toxic reference electrodes, such as RHE or Ag/AgCl.

The authors may provide schemes or references to the used operando and in situ XAS, Raman and ATR-SEIRAS cells and a description the catalysts are exposed to the spectroscopic techniques. The authors may state which definition are the terms operando and in-situ chosen for each technique. The authors may also combine the Methods in the main manuscript and the methods in the Supporting information into one section.

Reviewer #3 (Remarks to the Author):

Authors reported the dual-active sites catalyst for electrocatalytic CO₂ reduction reaction. The atomic Cu sites and Cu NPs exhibited an efficient CO₂-to-C₂ reaction. M-Cu₁/CuNP delivered a high C₂ products FE of 75.4% with a partial current density of 289.2 mA cm⁻² and remarkable long-term stability. Based on the experimental and theoretical studies, the boosted H₂O dissociation and formed*CHO play the crucial role in the CO₂-to-C₂ reaction. However, the description of X-ray and FTIR spectroscopy is not convincing. The reviewer has several questions regarding the mechanistic insights into the proposed reaction model. Please address these issues in detail.

1.

Line 92: XPS results show that the valence state of Cu in the catalysts increased in the order of P-Cu₁/CuNP < M-Cu₁/CuNP < R-Cu₁/CuNP. Can authors provide quantitative analysis?

2.

Line 100: "Figure 1F shows that the characteristic peaks of Cu Kβ_{1,3} for Cu₁/CuNP samples lied between

0 and +2 in the sequence of P-Cu₁/CuNP < M-Cu₁/CuNP < R-Cu₁/CuNP". In fact, it is difficult to distinguish the position of these characteristic peaks in figure 1F. Authors should present this figure clearly and provide detailed description.

3.

In Figure 3C, the increase of Cu valence state was studied using the absorption energy of white line position of XANES spectra. However, the average Cu valence state of pristine materials was examined using first derivative of XANES spectra (Figure 1H). Please provide the detailed XANES spectra of pristine materials in the supporting information.

4.

In Figure 3C, the feature of Cu K-edge XANES spectrum of M-Cu₁/CuNP at OCP is very similar with that of Cu foil (Cu₀). Also, the shift in the absorption energy of white line position of M-Cu₁/CuNP and Cu foil is close. These results suggest that Cu valence state of M-Cu₁/CuNP could be metallic Cu (or close to Cu₀) at OCP, which is not consistent with the XPS/XAS results in Figure 1H and Figure S3. Authors should add XANES spectrum of pristine M-Cu₁/CuNP and Cu foil in figure 3C. The detailed discussion should be provided. More information could be obtained using linear combination fitting.

5.

Line 193-195 (Figure 3C), authors claimed that the increase of Cu valence state could be attributed to the adsorption of CO₂ and H₂O molecular on Cu NPs and Cu sites. It is surprising that Cu valence state increases during CO₂RR. This result is not consistent with previous studies. Also, Cu K-edge XAS provides bulk information not surface information. These results and statements are not convincing. If the adsorption of CO₂ and H₂O does cause the oxidation process, authors should provide a series of operando XANES spectra obtained from positive potentials to negative potentials. The shifts in the absorption energy of white line position can be therefore discussed.

6.

Figure S12-S14 show the TEM/XPS/XRD results of M-Cu₁/CuNP after CO₂RR. Is it after 5 hours or 40 hours? Please provide the information.

7.

The experimental description of in situ XAS/Raman/SEIRAS is not clear. Were all in situ XAS/Raman/SEIRAS experiments performed in the flow cell with KOH electrolyte? Since cell configuration of traditional SEIRAS measurement is H-type cell, how did authors perform SEIRAS experiment with flow cell? Please provide the cell configuration. The changes in the OH-associated peak should be further addressed.

If authors performed the SEIRAS experiment with traditional cell configuration (H-type cell), the pH value of electrolyte changes during CO₂RR. Also, the pH value of electrolyte used in SEIRAS measurement is different from that in XAS/Raman experiments. How did authors correlate the FTIR results with XAS/Raman results?

8.

Cu will be oxidized in the alkaline solution. Why did not XAS (Figure 3C)/Raman (Figure 3D) results show the formation of CuO/Cu(+2)?

9.

Since the S/N ratio of CHO-associated peak is poor in figure 3F, the role of CHO in the CO₂ reduction

mechanism is questionable. To prove the proposed reaction model, authors should provide SEIRAS of P-Cu₁/CuNP and R-Cu₁/CuNP materials. The potential dependence of CHO-associated peak (at 1548 cm⁻¹) in different materials should be further discussed.

Reviewer #4 (Remarks to the Author):

In this work, the authors report the design of catalysts for electrochemical CO₂ reduction to C₂⁺ products based on a dual-active site strategy. They combine a water dissociation catalyst (single atom Cu) and a C-C coupling one (Cu nanoparticles) to produce Cu nanoparticles supported on Cu-N-C single atom matrix. They found that this combination enhances CO₂ reduction to C₂⁺, achieving a C₂ Faradaic efficiency of 75% and a partial current density of 289 mA with a total FE at -0.6 V vs RHE. While the fabrication of the dual catalysts could be interesting, I do not find the designed catalysts represent a significant advance in the field of electrochemical CO₂ conversion. My conclusion comes from both fundamental and applied perspectives as I explain below:

1. The electrochemical CO₂ conversion performance of the designed catalysts in this work is comparable to most of normal Cu nanoparticles catalysts. In a basic electrolyte, C₂ selectivity of 70-80% in the current density range of 200-500 mA/cm² has been frequently reported using Cu nanoparticle catalysts. A good catalyst for C₂ production should exhibit an FE of at least over 80%.
2. The high performance in this work was achieved with very basic electrolyte (5M KOH). Many recent analyses have pointed out the cost related to carbonate formation with alkaline electrolyte and suggest that this condition is not suitable for practical application. In this work, the author designed dual-active site catalysts that work well only in strong alkaline electrolyte (Figure S15). Therefore, it is unclear how the designed catalysts can be used in practical systems.
3. The mechanism for enhanced C₂⁺ production based on dual-active sites is rather speculative as there are not enough experimental and theoretical evidence to support that claim. For example, the authors claim that Cu-N-C matrix can enhance water dissociation to provide protons for C₂ formation, but they did not provide a clear experimental and theoretical evidence for the enhanced water dissociation. Some obvious questions are: How does the water dissociate on Cu sites? What happens to the OH⁻? How does the H⁺ transport to the surface of Cu nanoparticles (in the form of H⁺ or H)?
4. The conversion of Cu single atom to Cu nanoparticle/clusters during electrochemical CO₂ reduction are well-documented (Nat. Commun. 9, 415 (2018); Angewandte Chemie, 58, 15098 (2019)). The presence of Cu single site with operando XAS could be originated from non-electrically conductive sites (Nat. Commun. 13, 4190 (2022)). Therefore, it is unclear how the Cu single sites can help CO₂ reduction if they are not stable.
5. Additional controlled samples should be tested to confirm the proposed mechanism. For example, mixing Cu-N-C with a commercially available Cu nanoparticles or other metals such as Ag or Au to see if this mechanism is universal (i.e., for CO₂ conversion to other products).

Responses to the comments of the reviewers

Reviewer 1:

In this work, Feng et al reported a dual-active sites catalyst comprising atomic Cu sites and Cu nanoparticles for CO₂RR. Among them, the Cu nanoparticles facilitated the C-C coupling step by *CHO dimerization, and the atomic Cu sites boosted H₂O dissociation to form *H and further migrated to Cu nanoparticles. As a result, the catalyst shows the CO₂-to-C₂₊ performance with a FE of 75.4%. Some concerns remain regarding the suitability of the authors' conclusions, for which the reviewer would like to see additional data to address and then can justify whether this work is suitable for publication.

Response: We thank the referee very much for the comment, which help and guide us to think about and polish our work greatly. We have tried our best to answer the questions from the reviewers. We hope that we have addressed all of the questions satisfactorily.

1. I am confused about the Figure 3B. The CO stripping peak is determined at approximately -0.7 V vs. SHE (Nature Catalysis 2020, 3, 797-803). The peak around 0.8-0.9 V vs RHE should not be classified as CO stripping.

Response: We thank the referee again for the comment. The potential of CO stripping peak in the as-mentioned paper (*Nat. Catal.* 2020, 3, 797-803) was presented versus Standard Hydrogen Electrode (SHE) and 0.1 M NaOH was used as electrolyte, while the CO stripping experiments in our work were conducted in 0.1 M Na₂SO₄ electrolyte and the potential of CO stripping peak was presented versus Reversible Hydrogen Electrode (RHE). Unlike the SHE, the potential measured by RHE does not change with the pH, so it is a more widely used (*J. Phys. Chem. B* 2004, 108, 28, 9829-9833; *Proc. Natl. Acad. Sci. U.S.A.* 1919, 5, 160-163). In many literature (*Angew. Chem. Int. Ed.* 2017, 56, 8828-8833; *Nat. Commun.* 2021, 12, 6261; *Adv. Mater.* 2023, 35, 2208799), the CO stripping peak on metal-based catalyst located at range of 0.6-0.9 V vs. RHE. Moreover, we have added the LSV curve without CO adsorption on the catalyst as Figure S22 in the revised supporting information, and no stripping peak was observed at around 0.8-0.9 V vs. RHE. Therefore, the peak around 0.8-0.9 V vs. RHE in Figure 3B could be assigned to CO stripping.

In the revised manuscript, we have added the above literature as references and discussed them as “On the other hand, the electrochemical CO stripping voltammetry tests of R-Cu₁/Cu_{NP}, M-Cu₁/Cu_{NP} and P-Cu₁/Cu_{NP} were also performed to study the CO adsorption ability. Figure 3B shows a peak in the potential range of 0.8-0.9 V for R-Cu₁/Cu_{NP}, M-Cu₁/Cu_{NP} and P-Cu₁/Cu_{NP}. No peak was observed in the LSV curve without CO adsorption (Figure S22). Further according to pervious literature,³²⁻³⁴ the peak at around 0.8-0.9 V can be attributed to the CO stripping peak. Interestingly, the CO stripping peak occurred around 0.89 V for M-Cu₁/Cu_{NP} and P-Cu₁/Cu_{NP}, while it was around 0.82 V for R-Cu₁/Cu_{NP}. The positive shift suggested that M-Cu₁/Cu_{NP} and P-Cu₁/Cu_{NP} had stronger CO

binding ability than R-Cu₁/Cu_{NP}, which have higher Cu NP content. The results above indicated that the Cu NPs was beneficial for the adsorption and activation of CO₂ and CO.” Please see them in page 13 of the revised manuscript.

2. It is uncertain that why the author chose different reference samples in the tests. M-Cu₁/Cu_{NP} was compared with P-Cu₁/Cu_{NP} in KIE experiment, but compared with R-Cu₁/Cu_{NP} in CO stripping experiment. Why not compare the three samples together?

Response: We thank the referee again for the comment. According to the comment, we have conducted the KIE experiments, gas electro-response experiments and CO stripping experiments over R-Cu₁/Cu_{NP}, M-Cu₁/Cu_{NP} and P-Cu₁/Cu_{NP}, and discussed them in the revised manuscript by “The kinetic isotope effect (KIE) of H₂O/D₂O (H/D) experiments were performed to get insights into the role of H₂O dissociation in CO₂-to-C₂ products (Figure 3A). When H₂O was replaced by D₂O in the electrolyte, the formation rate of the product (for example C₂H₄) decreased over P-Cu₁/Cu_{NP}, M-Cu₁/Cu_{NP} and R-Cu₁/Cu_{NP}, and the level of its decrease was closely related to the content ratio of Cu₁/Cu_{NP}. If the KIE value (defined as the ratio of C₂H₄ formation rates in H₂O and D₂O) closes to 1, H₂O dissociation is not the rate-determining step over the catalyst. The KIE value for R-Cu₁/Cu_{NP}, M-Cu₁/Cu_{NP} and P-Cu₁/Cu_{NP} were 1.12, 1.21 and 2.87, respectively, which indicated that H₂O dissociation was accelerated gradually with the increase of Cu₁/Cu_{NP} content ratio. These results confirmed that the atomic Cu sites were responsible for accelerating H₂O dissociation and providing proton to adjacent Cu NPs, thus affecting CO₂-to-C₂ products. Meanwhile, the N-doped carbon matrix catalysts has been reported to favor the migration of proton.³⁰ In addition, the H₂ formation rate in 5 M KOH H₂O solution and D₂O solution was presented in Figure S17. An obvious decrease was observed over all the catalysts, while the decreasing degree followed the sequence of P-Cu₁/Cu_{NP} > M-Cu₁/Cu_{NP} > R-Cu₁/Cu_{NP}, suggesting that the increase of Cu single atom content can reduce the influence of isotope effect on H₂ formation rate.

Furthermore, we studied the influence of electrolyte pH on CO₂RR performance over R-Cu₁/Cu_{NP}, M-Cu₁/Cu_{NP} and P-Cu₁/Cu_{NP}. Three different concentrations of KOH aqueous electrolytes, i.e., 0.1 M, 3 M and 5 M, were employed to adjust the pH environment at the electrode/electrolyte interface. The results in Figure S18 show that the FE and formation rate of C₂ products increased with the increasing pH value of electrolyte, suggesting that strong basic local environment favored the C₂ products formation. Although CO₂RR activity of M-Cu₁/Cu_{NP} in low concentration KOH electrolytes was lower than that in 5 M KOH electrolyte, M-Cu₁/Cu_{NP} exhibited higher FE and formation rate of C₂ products than P-Cu₁/Cu_{NP} and R-Cu₁/Cu_{NP} in low concentration KOH electrolytes, which suggested that the cooperative effect of Cu NPs and atomic Cu sites still facilitated C₂ products formation. The ratio of C₂ products formation rate over M-Cu₁/Cu_{NP} to P-Cu₁/Cu_{NP}, i.e., Rate_M/Rate_P, in 5 M KOH (3.4) is higher than those in 3 M KOH (3.0) and 0.1 M KOH (2.3). It demonstrated that the role of atomic Cu sites in accelerating the H₂O dissociation process was significantly more pronounced at higher pH values, leading to the enhanced C₂

products formation, even if the dissociation of H₂O in higher pH alkaline electrolyte was a sluggish step.

We then investigated the capacity adsorption and activation of CO₂ and CO molecules on R-Cu₁/Cu_{NP}, M-Cu₁/Cu_{NP} and P-Cu₁/Cu_{NP} via gas electro-response experiments in a self-designed gas adsorption electroresponse device (Figure S19).³¹ The results in Figure S20 and Figure S21 show that the CO₂ and CO adsorption responses changed in the sequence of R-Cu₁/Cu_{NP} < M-Cu₁/Cu_{NP} < P-Cu₁/Cu_{NP}, suggesting that the adsorption and activation of CO₂ and CO were promoted as the ratio of Cu₁ to Cu_{NP} decreased. On the other hand, the electrochemical CO stripping voltammetry tests of R-Cu₁/Cu_{NP}, M-Cu₁/Cu_{NP} and P-Cu₁/Cu_{NP} were also performed to study the CO adsorption ability. Figure 3B shows a peak in the potential range of 0.8-0.9 V for R-Cu₁/Cu_{NP}, M-Cu₁/Cu_{NP} and P-Cu₁/Cu_{NP}. No peak was observed in the LSV curve without CO adsorption (Figure S22). Furthermore, according to pervious literature,³²⁻³⁴ the peak at around 0.8-0.9 V can be attributed to the CO stripping peak. Interestingly, the CO stripping peak occurred around 0.89 V for M-Cu₁/Cu_{NP} and P-Cu₁/Cu_{NP}, while it was around 0.82 V for R-Cu₁/Cu_{NP}. The positive shift suggested that M-Cu₁/Cu_{NP} and P-Cu₁/Cu_{NP} had stronger CO binding ability than R-Cu₁/Cu_{NP}, which have higher Cu NP content. The results above indicated that the Cu NPs was beneficial for the adsorption and activation of CO₂ and CO.” Please see them in page 12-13 of the revised manuscript.

3. The assignment of C≡O Raman bands at 2000-2100 cm⁻¹ is problematic. In fact, the CO bridge appears 1830-1900 cm⁻¹. (ACS Central Science 2016, 2, 522-528, The Journal of Physical Chemistry C 2017, 121, 12337-12344, ACS Catal. 2018, 8, 7507-7516, ACS Catal. 2019, 9, 6305-6319, Nat. Commun. 2022, 13, 2656.)

Response: We thank the referee again for the comment. We agree with the referee. The Raman band of CO bridge configuration locates at 1830-1900 cm⁻¹, but the peak at 2000-2100 cm⁻¹ can be assigned to the CO atop configuration (ACS Catal. 2018, 8, 7507-7516; ACS Catal. 2019, 9, 6305-6319; Nat. Commun. 2022, 13, 2656). In the revised manuscript, we have added the above literature as references and we modified the description of *in situ* SERS as “*In situ* surface-enhanced Raman spectroscopy (SERS) was employed to reveal the interaction between Cu species and the reaction intermediates during CO₂RR (Figure S25).³⁵⁻³⁸ As shown in the Raman spectra under different potentials (Figures 3D-3F), the peaks assigned to the restricted rotation of adsorbed CO (298 cm⁻¹, ν(Cu-CO)), Cu-CO stretching (365 cm⁻¹, ν(Cu-CO)) and C-O stretching of atop *CO (2000-2100 cm⁻¹, ν(CO)) were observed from -0.3 to -0.8 V over M-Cu₁/Cu_{NP}, while ν(Cu-CO) and ν(CO) peaks over P-Cu₁/Cu_{NP} appeared from -0.4 V and -0.5 V, respectively. This observation could be related to the lower onset potential for C₂ products for M-Cu₁/Cu_{NP}. Only CO rotation peaks can be observed on R-Cu₁/Cu_{NP} from -0.4 to -0.8 V, meaning poor CO₂ reduction activity. *In situ* SERS was also conducted over Cu-N-C (Figure S26), whereas no peaks associated with *CO intermediate were observed, suggesting that single atomic Cu was not conducive to the conversion of CO₂ to adsorbed CO. The results of *in situ* SERS suggested that CO₂-to-C₂ products proceeded via CO intermediate and moderate ratio of Cu₁ to Cu_{NP} facilitated CO₂ conversion of adsorbed CO.” Please see them in Page14 of the revised

manuscript.

4. It is not clear about the details of the *in situ* Raman/ATR-SEIRAS process. What electrolyte did the authors use for in-situ tests? The CO₂RR performance of the catalysts were evaluated in KOH aqueous solution. If the authors also used alkaline electrolytes for the *in situ* experiments, why did the CO₃²⁻ signal appear?

Response: We thank the referee again for the comment, and we answer the two questions as follows.

(1) 5 M KOH aqueous solution was used as electrolyte during the *in situ* Raman spectroscopy experiments. Considering strong basic electrolyte would damage the germanium ATR crystal, 3 M KOH aqueous solution was used as electrolyte during the *in situ* ATR-SEIRAS experiments. The details of *in situ* surface-enhanced Raman and ATR-SEIRAS experiment have been added in the Methods section of the revised manuscript: “*In situ* Raman measurements. A flow cell with a quartz window by GaossUnion (Tianjin) Photoelectric Technology Company was used to carry out *in-situ* Raman measurements using a Horiba LabRAM HR Evolution Raman microscope (Figure S25). A 785 nm laser was used and signals were recorded using a 20 s integration and by averaging two scans. The gas diffusion electrode sprayed with the catalyst was used as working electrode and a graphite rod and a Hg/HgO electrode were used as counter and reference electrodes, respectively. The counter electrode was separated from the working electrode by anion exchange membrane. The 5 M KOH aqueous solution was used as electrolyte and were recirculated by pump with flow rates of 20 mL min⁻¹. Meanwhile, CO₂ gas was continuously supplied to the gas chamber with a flow rate of 40 mL min⁻¹.”

In situ ATR-SEIRAS measurements: The experiments were conducted in a modified electrochemical cell that integrated into a Nicolet 6700 FTIR spectrometer equipped with MCT detector cooled by liquid nitrogen (Figure S27). The catalysts ink was dropped on a germanium ATR crystal deposited with Au film. A platinum wire and a Hg/HgO electrode were used as counter and reference electrodes. Each spectrum was collected with 32 times with a resolution of 4 cm⁻¹. The CO₂-saturated 3 M KOH aqueous solution was used as electrolyte to avoid that the germanium ATR crystal was damaged by strong basic electrolyte. The background spectrum was collected at the potential of +0.1 V vs RHE.” Please see them in Page 21-22 of the revised manuscript.

The optical photograph and schemes of cells used in *in situ* surface-enhanced Raman and *in situ* ATR-SEIRAS have been also provided in the revised supporting information as Figure S25 and Figure S27, respectively.

(2) There are following reactions for carbonate equilibria in electrolyte (*Science* 2021, 372, 1074-1078; *Nat. Catal.* 2022, 5, 564-570).

According to above reactions, it can be concluded that CO₃²⁻ exists as the main species in alkaline condition. The

pH value of CO₂-saturated 5 M KOH and 3 M KOH aqueous solution are 11.02 and 10.21, respectively, so CO₃²⁻ signal appeared in *in situ* Raman spectra. It was also reported in many literature (*J. Am. Chem. Soc.* 2022, 144, 14936-14944; *Angew. Chem. Int. Ed.* 2022, 61, e202110657).

5. The authors compared the catalytic performance with other literatures (Figure 2D), but the performances of this work are not outstanding, especially in alkaline electrolytes. Moreover, this kind of comparison does not help much with the advance of the research community, as when researchers get used to them, they just target higher metrics but ignoring science and insights behind the experiments.

Response: We thank the referee very much for the comment. We appreciate the reviewer's comment regarding the significance of revealing the science and insights behind the experiments, rather than solely pursuing high metrics. We have conducted additional experiments to further reveal the mechanism and revised the manuscript carefully. Additionally, it is significant to obtain high CO₂RR activity at low applied potential, which means low energy consumption. The as-prepared M-Cu₁/Cu_{NP} exhibited large C₂₊ products partial current density at applied potential, which is lower than most of the reported electrocatalysts for CO₂RR to C₂₊ products in literature. In response to the reviewer's suggestion to focus less on the catalytic performance, we have removed the figure comparing performance and instead included Table S7 in the revised supporting information and discussed them in the revised manuscript by "Compared with reported electrocatalysts for CO₂RR to C₂₊ products in the literature, M-Cu₁/Cu_{NP} exhibited higher CO₂RR to C₂₊ products activity at low applied potential, especially in term of C₂₊ partial current density (Table S7)." Please see them in Page 10 of the revised manuscript.

6. Regarding catalytic properties, the reviewer is not sure whether the authors have adopted rigorous protocol to ensure the accuracy of CO₂RR performance data. The outlet flowrates could be very different from the inlet flowrates due to the reaction between CO₂ and KOH, leading to problematic FE data of the gas products.

Response: We thank the referee again for the comment. Adopting rigorous protocol is important for obtaining accurate CO₂RR performance data. Our CO₂RR activity testing procedure is consistent with the methods described in the literature. We took into account that the outlet flow rate is not consistent with inlet flow rate due to the reaction between CO₂ and KOH. Therefore, we calculated the product FEs using the outlet flow rate to ensure accuracy.

In the revised manuscript, we have added further experiments details: "Electrocatalytic CO₂ reduction. To prepared gas diffusion electrode, 1 mg catalyst was suspended in 500 μL isopropanol with 10 μL Nafion D-521 dispersion (5 wt%) to form a homogeneous ink. Then the catalyst ink was spread onto the gas diffusion electrode of 0.5 × 2 cm² in area by a micropipette and then dried under room temperature. All the electrochemical experiments were conducted on the electrochemical workstation (CHI 660E) equipped with a high current amplifier (CHI 680C) and the CO₂RR performance was investigated in flow cell (Figure S7). The prepared gas diffusion electrode and Ni

foam were used as the cathode and anode, respectively. An anion exchange membrane was used to separate the cathode and anode compartment. Aqueous KOH solution (5 M) was used as the electrolyte solution and the electrolyte volume was 30 mL. The catholyte solution and anolyte solution were recirculated by two pumps with flow rates of 10 mL min⁻¹ and 30 mL min⁻¹, respectively. Meanwhile, CO₂ gas was continuously supplied to the gas chamber by using a mass flow controller with a flow rate of 40 mL min⁻¹. All the potentials were measured against a Hg/HgO reference electrode and converted to versus RHE with *iR* (80%) compensations, *i* represents the current obtained at corresponding potential, *R* is ohmic resistance of the cell measured by electrochemical workstation:

$$E_{\text{RHE}} = E_{\text{Hg/HgO}} + 0.098 + 0.059 \times \text{pH} - iR \times 80\%$$

The chronopotentiometry method is used for evaluating long-term CO₂RR activity stability of electrocatalyst in a flow cell. In order to address the issues of flooding and carbonation accumulation of GDE, the CO₂RR was interrupted every 5 hours and the GDE were removed, washed with deionized water and followed by dryness under N₂ atmosphere. Meanwhile, the electrolyte solution was refreshed for each interval.

Product Analysis. The gaseous products were collected using a gas bag and quantified by gas chromatography (GC 7890B). A thermal conductivity detector (TCD) and a flame ionization detector (FID) were used to quantify H₂, CO, and other alkane contents, respectively. The Faradaic efficiency (FE) of gaseous products was calculated by the equation:

$$FE = \frac{znVF}{Q} \times 100$$

Where *z* represents the number of electrons transferred for product formation, *n* is the volume concentration from GC, *V* is the total volume calculated by outlet flow rate, *F* is Faraday constant (96485 C/mol) and the *Q* is the amount of cumulative charge recorded by the electrochemical workstation.

The liquid product was analyzed by ¹H NMR (Bruker Advance III 400 HD spectrometer) in deuterioxide. To accurately integrate the products in NMR analysis, the sodium 2, 2-dimethyl-2-silapentane-5-sulfonate (DSS) was the reference for ethanol and acetic acid, and phenol was the reference for formate. The FE of liquid products was calculated by the equation:

$$FE = \frac{zCVF}{Q} \times 100$$

Where *z* represents the number of electrons transferred for product formation, *C* is the liquid concentration obtained from NMR, *V* is electrolyte volume, *F* is Faraday constant (96485 C/mol) and the *Q* is the amount of cumulative charge recorded by the electrochemical workstation.” Please see them in pages 19-20 of the revised manuscript.

Reviewer 2:

The authors present a multi-active site catalyst by mixing Cu Nanoparticles and atomic Cu sites, both supported on an N-doped carbon matrix. Additionally, they have wisely chosen to additionally prepare catalysts comprising only Cu NPs or Cu atomic sites. The catalysts were thoroughly prepared and fully characterized. The CO₂ reaction was performed in alkaline media, in a flow cell, focusing on the detection of C₂ products. A wide variety of *in situ/operando* techniques have been used to understand the evolution of the catalysts under reaction conditions. The authors claim a mixture of Cu_{NP} and Cu atomic sites to be more beneficial than both individual systems comprising a synergistic effect. By characterizing the catalysts in the as prepared state and during reaction, the authors claim to find that C-C coupling occurs mostly on Cu_{NP}, while H₂O dissociation appears mostly at atomic Cu sites, *H would migrate to the Cu_{NP} and thus promote the protonation of *CO. To achieve these results, the authors have comprehensively used different techniques to understand the catalysts under reaction conditions. The authors have started strongly demonstrating the interesting nature of the catalysts and characterizing them well in the pristine state. The data sets and their analysis of the measured data under reaction conditions or after reaction is, however, incomplete. The data for at least one catalyst for most of the presented *in situ/operando* techniques is not presented. Thus, the interpretation is weakly supported on the basis of these investigations. It is suggested to integrate the missing data and to revise the presented interpretation in this manuscript majorly.

Response: We thank the referee very much for the comment, and we have carefully revised the manuscript based on the comments. The datapoint values, error bars and total current density of CO₂RR activity test have been added. The characterization of the catalyst after reaction have been supplemented and discussed in detail, such as TEM, XRD, XPS, HAADF-TEM. More details of experiment and cell structure have been included in the method section of the manuscript. The KIE experiments, gas electro-response experiments and CO stripping experiments, *in situ* spectroscopic characterizations over all the three catalysts have been presented and discussed in details. Based on the results presented above, we have revised the manuscript carefully, which can be known from the detailed discussion below.

Finally, we would like to take this chance to thank the referee for the invaluable advice and suggestions, which help and guide us to improve and polish our work greatly. We have tried our best to answer the questions from the reviewers and modify the manuscript. We hope that we have addressed all of the questions satisfactorily.

1. INTRODUCTION : The introduction should be revised providing more information and state-of-the-art knowledge about multi-active sites for multiple reactants in CO₂RR (L33). A lot of studies have been performed to study the role of particle size, distance or shape. A more through summary on the literature should be presented with a focus on the relevant parameters for this work (L49-51). The cited literature should show similar reaction systems, e.g. in L56, the authors claim H₂O activation on Cu single atoms, but the literature presents their data in an acidic

system, while the authors study alkaline systems. A discussion why this result like this is transferable, is thus required. A specification on the ratio of Cu_{NP} compared to Cu single atoms should be given already in L60.

Response: We thank the referee again for the comment. According to the reviewer's suggestion, we have revised the Introduction section: "Significant efforts have been devoted to the development of dual-active sites catalysts for boosting CO₂RR performance. Many reported catalysts are composed the active sites that are responsible for CO₂ activation (e.g. Au, Ag and Zn) and further hydrogenation or C-C coupling (Cu), respectively. It has been demonstrated that the intermediate migration (e.g. *CO) plays a crucial role on the selectivity of the reaction, which greatly depends on the optimal site distribution and distance between the dual active sites. Meanwhile, an increase in the *CO coverage over Cu sites has been shown to promote the formation of C₂₊ products.⁹⁻¹¹ In addition, considering H₂O molecule serves as the proton source for hydrogenation step, the active site for H₂O dissociation has been designed to be part of the dual-active sites catalysts.^{12, 13} The introduction of H₂O dissociation sites, such as metal single atoms, sulfur, and oxygen vacancies, has been shown to accelerate H₂O dissociation into *H species, which are subsequently fed to CO₂ conversion sites through *H spillover.¹⁴⁻¹⁶ However, these findings have mainly focused on the influence of H₂O dissociation in CO₂-to-C₁ product process (formate or methane), while it is of greater significance for the C₂ products formation process. This is due to the commonly used alkaline electrolyte in CO₂-to-C₂ product, which slows down the H₂O dissociation step, resulting in a sluggish reaction kinetic procedure.^{15, 17} Therefore, constructing dual-active sites electrocatalyst to accelerate CO₂ reduction and H₂O dissociation respectively is a feasible strategy for achieving the desired electrochemical CO₂-to-C₂ products performance.

Cu is the most promising electrocatalyst for converting CO₂ into C₂ products, due to its moderate adsorption capacity for the key intermediates.¹⁸⁻²⁰ Diverse strategies have been investigated to improve the C₂ selectivity and current density, such as the manipulation of crystal facets, morphology, particle size, and oxidation state.²¹⁻²³ However, most of these studies mainly focus on regulating the Cu structure on CO₂ activation and C-C coupling, and the reports about the effect of surface *H coverage through accelerating H₂O dissociation are very limited. It is important to note that *H coverage must be controlled rigorously, since excessive *H favors H₂ production *via* the competing hydrogen evolution reaction (HER). Recently, single-atom catalyst characterized by the uniform dispersion of isolated metal atoms on a substrate has received significant attention in HER, owing to utmost metal utilization, tunable electronic structures and structural stability. Among them, the HER activity of single-atom Cu catalyst in alkaline electrolyte can be regulated through modifying coordinated environment and support type.²⁴⁻²⁶ Therefore, designing a dual-active sites catalyst with co-loaded Cu nanoparticles (NPs) and single-atom Cu sites could realize CO₂ conversion and H₂O activation simultaneously, favoring the high performance for electroproduction of C₂ products." Please see them in Page 4 of the revised manuscript. In addition, the content ratio of atomic Cu sites to Cu NPs of M-Cu₁/Cu_{NP} has been added in the last paragraph of the Introduction section.

2. TEM ANALYSIS : The Cu nanoparticle size is a relevant parameter for the catalytic performance. It would thus be relevant to mention the obtained sizes in the main text and present particle size distribution for all catalysts. The authors may also provide a NP size distribution for the R-Cu₁/Cu_{NP} as they have nicely done for the other two samples. Additionally, a more exact measure for the ratio between Cu NP and single atom sizes from the TEM data should be given and discussed against the EXAFS fitted data and the observed percentages between the coordination numbers. In this regard, the HAADF-STEM images showing the single atom Cu sites are missing in Figure S1 and S2 and are asked to be provided.

Response: We thank the referee again for the comment. The XAFS is a commonly used characterization method for single-atom catalysts, which can show target element valence state by comparing with reference samples and details of the atomic structure through fitting (e.g., coordination number, bond length) (*Science* 2019, 364, 1091–1094; *Nat. Catal.* 2018, 11, 870-877; *Rev. Mod. Phys.* 2000, 72, 621-654). TEM is a commonly method to determine the particle size, shape and distribution of sample. Although the aberration-corrected HAADF-STEM can observe the atomic image, it cannot show atomic structure information. Therefore, the results obtained by XAFS and TEM are not the same scale. Additionally, the number of single atoms cannot be counted by TEM. Therefore, it is difficult to calculate the ratio of nanoparticle to single atom through TEM results (*Adv. Mater.* 2022, 34, 2200057; *ACS Catal.* 2022, 12, 971-981).

According to the comment, the nanoparticle size distribution of the R-Cu₁/Cu_{NP} was presented as Figure S2 in the revised supporting information, which showed that the average nanoparticle size is around 5 nm. The HAADF-STEM images of P-Cu₁/Cu_{NP} and R-Cu₁/Cu_{NP} have been added in Figure S1 and S2 of the revised supporting information. We have discussed them in the revised manuscript by “In addition, the TEM and aberration-corrected HAADF-STEM images of P-Cu₁/Cu_{NP} and R-Cu₁/Cu_{NP} are shown in Figure S1 and Figure S2, respectively. The TEM images showed that both P-Cu₁/Cu_{NP} and R-Cu₁/Cu_{NP} exhibited similar average Cu NPs size to M-Cu₁/Cu_{NP}, while the amount of Cu NPs obviously decreased over R-Cu₁/Cu_{NP}. Meanwhile, evident single atomic Cu can also be observed on HAADF-STEM images of P-Cu₁/Cu_{NP} and R-Cu₁/Cu_{NP}.” Please see them in page 5 of the revised manuscript.

3. XPS ANALYSIS : Please comment further on the observed differences and its relevance in the binding energies of the Cu_{2p}3/2 peaks (L94). Additionally, fits that indicate the small amounts of Cu²⁺ species in the M and P catalysts should be presented. The authors also may provide an analysis of the Auger peaks for distinguishing the metallic Cu and the Cu¹⁺ oxidation states of the catalysts, if possible. The content of Cu should be set into context with the content of C and N in the catalysts (L95), and any impurities should be ruled out (e.g. sulfur from guanidine thiocyanate). The analysis of N1s for the other two catalysts should be presented (Figure S4).

Response: We thank the referee again for the comment. To avoid the influence of long-term exposure in air, we have

resynthesized the three catalysts and conducted XPS characterization. The XPS survey spectra of the catalysts have been added in the revised supporting information as Figure S3, which showed that only Cu, N, C and O elements existed in P-Cu₁/Cu_{NP}, M-Cu₁/Cu_{NP} and R-Cu₁/Cu_{NP}. No other element could be detected. The existence of O element can be attributed to the adsorbed oxygen on surface. The contents of Cu in P-Cu₁/Cu_{NP}, M-Cu₁/Cu_{NP}, and R-Cu₁/Cu_{NP} were 7.4, 4.3, and 1.9 at%, respectively, and the corresponding N and C content were also displayed in Table S1 of the revised supporting information. The Cu 2*p* peaks were fit and displayed in Figure 1F in the revised manuscript. Only Cu 2*p*_{3/2} peak attributed to Cu^{0/+} existed in the spectra, and the peak moved to lower binding energy in the order of R-Cu₁/Cu_{NP} (933.0 eV), M-Cu₁/Cu_{NP} (932.7 eV) and P-Cu₁/Cu_{NP} (932.5 eV), indicating that the valance state of Cu in the catalysts increased gradually. We have also provided the Cu LMM Auger spectra in the revised supporting information as Figure S4, which confirmed that Cu⁺ and Cu⁰ co-existed in P-Cu₁/Cu_{NP}, M-Cu₁/Cu_{NP} and R-Cu₁/Cu_{NP}. Considering the valance state of Cu in Cu NP is 0, and that of atomically dispersed Cu site is Cu^{δ+}, the content of Cu₁ site in the catalyst changed in order of P-Cu₁/Cu_{NP} < M-Cu₁/Cu_{NP} < R-Cu₁/Cu_{NP}. The N 1*s* spectra of R-Cu₁/Cu_{NP}, M-Cu₁/Cu_{NP} and P-Cu₁/Cu_{NP} were presented in the revised supporting information as Figure S5 and obvious Cu-N peak can be observed in all spectra.

In the revised manuscript, we have discussed XPS results by “The XPS survey spectra show that the presence of Cu, N, C and O elements in P-Cu₁/Cu_{NP}, M-Cu₁/Cu_{NP} and R-Cu₁/Cu_{NP} (Figure S3), where the existence of O element can be attributed to the adsorbed oxygen on surface. The contents of Cu in P-Cu₁/Cu_{NP}, M-Cu₁/Cu_{NP}, and R-Cu₁/Cu_{NP} were 7.4, 4.3, and 1.9 at%, respectively, and the corresponding N and C content are displayed in Table S1. The Cu 2*p* spectra (Figure 1F) suggest that only Cu 2*p*_{3/2} peak attributed to Cu^{0/+} existed in the spectra, which moved to lower binding energy in the order of R-Cu₁/Cu_{NP} (933.0 eV), M-Cu₁/Cu_{NP} (932.7 eV) and P-Cu₁/Cu_{NP} (932.5 eV), indicating that the valance state of Cu in the catalysts increased gradually. The Cu LMM Auger spectra confirmed that Cu⁺ and Cu⁰ coexisted in P-Cu₁/Cu_{NP} M-Cu₁/Cu_{NP} and R-Cu₁/Cu_{NP} (Figure S4). Considering the valance state of Cu in Cu NP is 0, while that of atomically dispersed Cu site is Cu^{δ+}, thereby the content of Cu₁ site in the catalyst changed in order of P-Cu₁/Cu_{NP} < M-Cu₁/Cu_{NP} < R-Cu₁/Cu_{NP}. Meanwhile, the high resolution XPS N 1*s* spectra of P-Cu₁/Cu_{NP}, M-Cu₁/Cu_{NP}, and R-Cu₁/Cu_{NP} are displayed in Figure S5. All the spectra showed obvious Cu-N peak at around 399.2 eV, which suggested that the Cu₁ sites were coordinated by N atoms and other peaks could be attributed to pyridinic N (~398.5 eV), pyrrolic N (~399.9 eV), and graphitic (~401.0 eV), respectively.” Please see them in page 6 of the revised manuscript.

4. XES ANALYSIS : The authors may provide a motivation why they used this technique and how it would be helpful to understand the catalysts better. L103 & Figures 1F and 1G: The authors may comment why their references are so different from literature (see e.g. Vegelius et al., J. Anal. At. Spectrom., 2012,27, 1882). L104-105 & Figure 1G: The authors may also comment why their data quality is so low compared to their mentioned reference literature

and what they could significantly learn from the XES experiments, which they did not learn from XAS or XPS.

Response: We thank the referee again for the comment. The XES experiments was adopted to identify the coordination atoms of Cu single atoms and Cu valence state. However, the data quality was much lower than that reported in literature due to the limitation of the detector accuracy in XES experiments, which affected the validity of the conclusions. Additionally, according to the results of XPS and XAS, it can be identified that Cu valence state varied in the sequence of P-Cu₁/Cu_{NP} < M-Cu₁/Cu_{NP} < R-Cu₁/Cu_{NP}, and Cu single atoms were coordinated by N atoms. Therefore, according to the comment, we have deleted the XES experiments data, because we have already obtained the same conclusion by the more accurate technique (XPS and XAS).

5. XAS ANALYSIS : The XAS analysis has been thoroughly performed. However, the descriptive text starting from L106 is not clear. E.g., in L108, it is unclear, which catalyst is meant. As much as the Cu valence state increases in a sequence it is more vital in the reviewer's opinion, that the P- and the R-catalyst show very similar intensity transients than the two references, indicating that both could be used to represent the reference catalysts. Please also comment on the artifact(?) peaks around 1 Å (L112, Figure 1I) and on the calculation of the percentage between Cu NP and single Cu sites (L117). Please also comment on the maximum observed coordination number of the Cu NPs and if an estimation of the particle size can be extracted from it.

Response: We thank the referee again for the comment. The XAS characterization of R-Cu₁/Cu_{NP} has been re-performed and Cu valence state was analyzed through XANES spectra and the results were presented as Figure 1G in the revised manuscript. The Cu K-edge XANES of the three catalysts was compared with reference samples to determine the average Cu valence state, which was found to lie between 0 and +2. The average Cu valence state varied in the order of P-Cu₁/Cu_{NP} < M-Cu₁/Cu_{NP} < R-Cu₁/Cu_{NP}, consistent with XPS results. The FT EXAFS of the three catalysts was analyzed and Figure 1H in the revised manuscript shows that all three catalysts exhibited peaks at around 1.5 Å and 2.2 Å, attributed to Cu-N and Cu-Cu coordination, respectively. Therefore, atomic Cu and metallic Cu NP co-existed in the three catalysts. Notably, the intensity ratio of the Cu-N peak to the Cu-Cu peak increased with the sequence progressed from P-Cu₁/Cu_{NP}, M-Cu₁/Cu_{NP} to R-Cu₁/Cu_{NP}, indicating a gradual increase in the content ratio of Cu₁ to Cu NP. The peak located below 1 Å, i.e., low-frequency structure, can be ascribed to the resonant scattering from the electronic states within the central atom. The structure is not part of the structural information. These non-structural contributions to the total absorption are considered to be part of the background (*Phys. Rev. B* 1993, 47,14126-14131). The percentage of Cu NP and single Cu site was calculated according to the method reported in the literature (*J. Phys. Chem. C* 2012, 116, 24999-25003). The structural information provided by XAFS is the average value and the coordination number obtained through EXAFS fitting is the information of atomic structure. It has not been reported that estimating particle size from EXAFS fitting coordination number.

In the revised manuscript, we have discussed them by “To further determine the chemical state and local coordination environment, X-ray absorption near-edge structure (XANES) and extended X-ray absorption fine structure (EXAFS) measurements were conducted. The Cu K-edge XANES edge of R-Cu₁/Cu_{NP}, M-Cu₁/Cu_{NP} and P-Cu₁/Cu_{NP} with the reference samples Cu foil and Cu phthalocyanine (CuPc) were presented in Figure 1G, the adsorption edge of all catalysts located between that of Cu foil and CuPc, indicating the average Cu valence state of Cu in the catalysts laid between 0 and +2. Meanwhile, adsorption edge moved to higher energy in the sequence of P-Cu₁/Cu_{NP} < M-Cu₁/Cu_{NP} < R-Cu₁/Cu_{NP}, confirming that the average Cu valence state increased with the increase of Cu₁/Cu_{NP} ratio. It is in accordance with the XPS results. The Fourier-transformed (FT) EXAFS spectra in the R-space of the three catalysts was processed and displayed in Figure 1H, all three catalysts exhibited peaks at around 1.5 Å and 2.2 Å, attributed to Cu-N and Cu-Cu coordination, respectively. Therefore, atomic Cu and metallic Cu NP co-existed in R-Cu₁/Cu_{NP}, M-Cu₁/Cu_{NP} and P-Cu₁/Cu_{NP}. Notably, the intensity ratio of the Cu-N peak to the Cu-Cu peak increased in the order of P-Cu₁/Cu_{NP} < M-Cu₁/Cu_{NP} < R-Cu₁/Cu_{NP}, indicating a gradual increase in the content ratio of atomic Cu to Cu NP. In order to obtain the local structure of Cu species and the content ratio of atomic Cu to Cu NP, the quantitative analysis by the least-squares EXAFS fittings was performed (Figure S6). The percentages (P) of Cu-N₄ (Cu₁) and Cu NPs in total Cu species of catalysts were displayed in Table S2. Thus, the content ratio of Cu₁ to Cu_{NP} of P-Cu₁/Cu_{NP}, M-Cu₁/Cu_{NP} and R-Cu₁/Cu_{NP} were 0.39, 0.25 and 0.05, respectively, which is consistent with our catalyst design expectation.” Please see them in Page 6-7 of the revised manuscript.

6. ELECTROCATALYTIC CO₂RR PERFORMANCE : The authors present very interesting and carefully studied CO₂ reduction reaction results. The P- and R- catalysts should however be set into the context of literature, as Cu NPs and Cu single atoms are already well studied.

Response: We thank the referee again for the comment. We have discussed them in the revised manuscript by “The control potential electrolysis was then performed to analyze the reduction products and the catalyst loading was 1 mg cm⁻² (Figure S9). The gas-phase and liquid-phase products were analyzed by gas chromatography and ¹H nuclear magnetic resonance spectroscopy, respectively. H₂, CO, CH₄, formate, C₂H₄, ethanol and acetate were formed. Figure 2A shows that the FE of C₂ products (FE_{C2}) of M-Cu₁/Cu_{NP} exhibited a volcano-shaped dependence on the applied potential (Figure S10, Tables S3-S5), and a maximum FE_{C2} could reach 75.4% at -0.6 V, which is much higher than that over R-Cu₁/Cu_{NP} and P-Cu₁/Cu_{NP}. The maximum FE_{C2} of P-Cu₁/Cu_{NP} was 47.3% at -0.7 V, which closed to the performance of Cu nanoparticles reported in the literature.^{22, 27} Moreover, the M-Cu₁/Cu_{NP} had a lower onset potential for C₂ products formation. The FE_{C2} could reach 20.5% at -0.4 V over M-Cu₁/Cu_{NP}, while C₂ products cannot be detected over R-Cu₁/Cu_{NP} and P-Cu₁/Cu_{NP} under the same potential. The ratio of FE_{C2} to FE_{C1} could keep > 4.5 from -0.6 to -0.9 V over M-Cu₁/Cu_{NP}, while those of R-Cu₁/Cu_{NP} and P-Cu₁/Cu_{NP} were below 2 in the whole applied potential range (Figure 2B). Furthermore, M-Cu₁/Cu_{NP} was treated by sulfuric acid solution to completely

remove Cu NPs (denoted as Cu-N-C) and the results of TEM, XRD and XAS indicated that only Cu single atoms existed in the Cu-N-C catalyst (Figures S11, S12). H₂ was the dominant product over Cu-N-C in the whole applied potentials range, suggesting that the atomic Cu sites mainly facilitated H₂O dissociation (Figure 2C and Table S6). The formation of CO and CH₄ was also observed, in agreement with previous reports that Cu single atoms could generate H₂, CO, and CH₄.^{25, 28, 29} These results suggest that there is synergistic effect between Cu sites and Cu NPs, and the proper ratio of Cu₁ to Cu_{NP} would obviously enhance the selectivity for C₂ products.” Please see them in Page 9-10 of the revised manuscript.

7. How were the presented LSVs in Figure S6 normalized? The reviewer assumes that a normalization to the ECSA results, which are presented in Figure S10, would change the results and thus the interpretation. A thoughtful analysis of the electrochemical surface area is necessary here to evaluate the observed current densities. Additionally, Figure S6, and in general for most Figures, specifically the electrochemistry figures, lack a complete description of the used electrolyte, scan rate and normalization, which should be provided.

Response: We thank the referee again for the comment. The LSV curves presented in Figure S6 were not normalized by ECSA, while the potentials were treated by 80% ohmic compensation. Although the Hg/HgO electrode was used as reference electrode during experiments, the applied potentials were converted to versus reversible hydrogen electrode (RHE) with 80% ohmic compensation in figures. The formula for the conversion as:

$$E_{\text{RHE}} = E_{\text{Hg/HgO}} + 0.098 + 0.059 \times \text{pH} - iR \times 80\%$$

where $E_{\text{Hg/HgO}}$ represents the potential versus Hg/HgO electrode, pH is the pH value of electrolyte, i represents the current obtained at corresponding potential, R is ohmic resistance of the cell measured by electrochemical workstation. The experiment details have been added in the Method section of revised manuscript and we have marked the test conditions in the caption of electrochemistry figures in revised manuscript and supporting information.

The LSV curves are commonly used to judge whether CO₂RR occurred and measure the onset potential. It is seldom reported to normalize LSV curve by using ECSA. As shown in Figure S8 of the revised supporting information, R-Cu₁/Cu_{NP}, M-Cu₁/Cu_{NP} and P-Cu₁/Cu_{NP} showed significant increase in cathodic current density when feed gas was change from N₂ to CO₂, indicating that CO₂RR occurred over all the three catalysts. Meanwhile, M-Cu₁/Cu_{NP} exhibited the most positive onset potential and highest current density in the presence of CO₂. The current density reached 1093.0 mA cm⁻² over M-Cu₁/Cu_{NP} at -0.8 V, which was roughly 2.5 and 1.7 times higher than that over R-Cu₁/Cu_{NP} and P-Cu₁/Cu_{NP}, respectively. The results suggested that M-Cu₁/Cu_{NP} has higher CO₂ activity than R-Cu₁/Cu_{NP} and P-Cu₁/Cu_{NP}, which could be attributed to the moderate content ratio of Cu₁ to Cu_{NP}.

In the revised manuscript, we have discussed them by “The CO₂RR performance of the as-prepared catalysts were first evaluated by linear sweep voltammetry (LSV) in 5 M KOH aqueous solution in a flow cell (Figure S7). As

shown in Figure S8, all the catalysts exhibited significant increase in cathodic current density when feed gas was change from N₂ to CO₂, indicating that CO₂RR occurred over R-Cu₁/Cu_{NP}, M-Cu₁/Cu_{NP} and P-Cu₁/Cu_{NP}. Meanwhile, M-Cu₁/Cu_{NP} exhibited the most positive onset potential and highest cathodic current density in the presence of CO₂ gas, and taking -0.8 V as an example, the current density reached up to 1093.0 mA cm⁻² over M-Cu₁/Cu_{NP}, which was roughly 2.5 and 1.7 times higher than that of R-Cu₁/Cu_{NP} and P-Cu₁/Cu_{NP}, respectively. Therefore, the results of LSV experiments suggested that M-Cu₁/Cu_{NP} has higher CO₂ activity than R-Cu₁/Cu_{NP} and P-Cu₁/Cu_{NP}, which could be attributed to the moderate content ratio of Cu₁ to Cu_{NP}.” Please see them in page 9 of the revised manuscript.

8. It is unclear, which figure is described here in L136. The data in Figure 2A are nicely evaluated. Please also add the obtained total currents to the graphs. For transparency, please also add a table with the obtained values and error bars (or add them to the plots). Describe in the experimental data how many datapoints are comprised in one error bar.

Response: We thank the referee again for the comment. The figure is described in L136 is Figure 2A. We apologize for the confusion caused by unclear expression and have revised the manuscript as “Figure 2A shows that the FE of C₂ products (FE_{C2}) of M-Cu₁/Cu_{NP} exhibited a volcano-shaped dependence on the applied potential (Figure S10, Tables S3-S5), and a maximum FE_{C2} could reach 75.4% at -0.6 V, which is much higher than that over R-Cu₁/Cu_{NP} and P-Cu₁/Cu_{NP}. The maximum FE_{C2} of P-Cu₁/Cu_{NP} was 47.3% at -0.7 V, which closed to the performance of Cu nanoparticles reported in the literature.^{22, 27}” According to the comment, we have added total current density in Figure S13 in the revised supporting information. Three datapoints are comprised in one error bar. We have described it in the method section of the revised manuscript and added tables with datapoint values and error bars (Tables S3-S5) in the revised supporting information.

9. The additional synthesis procedure for the Cu-N-C catalyst should be described in the experimental data and in the figure descriptions (Figures S7-S9). It looks unorganized if this catalyst is only introduced here, but not in the general catalyst characterization in Figure 1. It is proposed that this Cu-N-C catalyst is used as a 4th main catalyst, and the whole characterization is also performed on this one. Please comment on the huge H₂ production on the Cu-N-C and the much better performance of the R-Cu₁/Cu_{NP} (L145). Please also relate the results of Cu-N-C to literature. Discuss that Cu-N-C catalysts should in principle hamper any C-C coupling, but still small amounts of ethylene are formed around -0.6 V.

Response: We thank the referee again for the comment. The synthesis procedure of Cu-N-C has been supplemented in the Methods section of the revised manuscript and figure description of Figures S11, S12 in the revised supporting information. The Cu-N-C catalyst was not used as a 4th main catalyst, and it was only a sample obtained by acid treatment of M-Cu₁/Cu_{NP}. The activity of Cu-N-C catalyst was measured to confirm the role of Cu single atom, so

the Cu nanoparticles or clusters in Cu-N-C catalyst must be completely removed. We have re-optimized the conditions of acid treatment and the related part of the manuscript is changed into “The M-Cu₁/Cu_{NP} was added into 50 mL 1 M sulfuric acid aqueous solution and heated at 80 °C for 48 h, then washed with deionized water several times and dried at 80°C overnight.” in the Methods section of revised manuscript and the figure descriptions of Figures S11, S12 in the revised supporting information.

TEM, XRD, XPS and XAS characterization have been conducted for Cu-N-C catalyst. No obvious nanoparticles were observed in TEM image and a large amount of isolated bright dots recognized as Cu single atoms were observed in aberration-corrected HAADF-STEM image. No diffraction peaks related to crystalline Cu species can be identified in XRD pattern. XPS spectra showed that Cu element existed in Cu-N-C catalyst and Cu⁺ was main species. More importantly, the EXAFS profiles in the R-space of Cu-N-C suggested that only the peak attributed to Cu-N coordination located at around 1.4 Å could be observed, while the Cu-Cu coordination peak at around 2.2 Å disappeared, which confirmed that the Cu species only existed as single atom form. The results of least-squares EXAFS fitting confirmed that the Cu-N coordination numbers in Cu-N-C was 4.0 (Table S2 of the revised supporting information), implying that the atomic Cu species mainly existed as Cu-N₄ structure. We have discussed the characterization results of Cu-N-C in the figure description of Figures S11 and S12.

The activity of the Cu-N-C obtained through acid treatment was remeasured and the data was presented in the revised manuscript as Figure 2C and the revised supporting information as Table S6. The results indicated that H₂ was the dominant product in the whole applied potentials range. CO and CH₄ was also detected, which was in agreement with previous reports in the literature (*Small Struct.* 2021, 2, 2000058; *Nat. Commun.* 2021, 12, 586; *Appl. Surf. Sci.* 2023, 610, 155506). It indicated that Cu single atoms could generate H₂, CO, and CH₄. We have discussed them in the revised manuscript as “Furthermore, M-Cu₁/Cu_{NP} was treated by sulfuric acid solution to completely remove Cu NPs (denoted as Cu-N-C) and the results of TEM, XRD, XPS and XAS indicated that only Cu single atoms existed in the Cu-N-C catalyst (Figures S11, S12). H₂ was the dominant product over Cu-N-C in the whole applied potentials range, suggesting that the atomic Cu sites mainly facilitated H₂O dissociation (Figure 2C and Table S6). The formation of CO and CH₄ was also observed, in agreement with previous reports that Cu single atoms could generate H₂, CO, and CH₄.^{25, 28, 29}”. Please see them in pages 9-10 of the revised manuscript. As for R-Cu₁/Cu_{NP}, we have reformed characterization over R-Cu₁/Cu_{NP}, and found Cu nanoparticles on R-Cu₁/Cu_{NP} through TEM (Figure S2 in the revised supporting information). The results of XAS were showed in the Figures 1G and 1H of the revised manuscript, which indicated that Cu single atoms and Cu nanoparticles co-existed in R-Cu₁/Cu_{NP} and the content ratio of Cu₁ to Cu NP was 0.39 (Table S2 in the revised supporting information).

10. The discrepancy of the obtained results from DL Capacitance are huge (L145). If the M- and P-catalysts would be normalized to the R-catalyst, roughness factors of 1, 2 and 5 would emerge (R, M, P). It is thus not enough to

just normalize the CO₂RR data to these values due to many other parameters that could bias the results (loading, particle distances, agglomeration, ...). Please justify your results by remeasuring one or two potentials of your catalysts with similar ECSA (e.g. by lowering the loading)

Response: We thank the referee again for the comment. We agree with reviewer's opinion that just normalizing CO₂RR data by ECSA is limit. Therefore, the normalized C₂ products partial current density by ECSA was abandoned in the revised manuscript. Meanwhile, we have investigated the influence of M-Cu₁/Cu_{NP} loading on CO₂RR activity. The CO₂RR activity of the working electrode with 0.25, 0.5, 1, 1.5 and 2 mg cm⁻² M-Cu₁/Cu_{NP} were measured in 5 M KOH electrolyte at -0.6 V in the flow cell. The catholyte solution and anolyte solution were recirculated by two pumps with flow rates of 10 mL min⁻¹ and 30 mL min⁻¹, respectively. CO₂ gas was continuously supplied to the gas chamber by using a mass flow controller with a flow rate of 40 mL min⁻¹. The results were added as Figure S9 in the revised supporting information. Both C₂ products selectivity and total current density increased as the increasing loading of M-Cu₁/Cu_{NP} until 1 mg cm⁻², and no obvious difference was observed among 1, 1.5 and 2 mg cm⁻². Therefore, we chose 1 mg cm⁻² as the catalyst loading to conduct activity experiments.

In the revised manuscript, we have discussed them by "The control potential electrolysis was then performed to analyze the reduction products and the catalyst loading was 1 mg cm⁻² (Figure S9)." Please see them in Page 9 of the revised manuscript.

11. Please establish your choice to measure chronopotentiometry for the long-term stability test instead of chronoamperometry (L155). Discuss the long-term stability results on the results of the other two (or three catalysts). How would you expect that the R- and P-catalysts behave during 40 h? Please also show the results of the other products, especially hydrogen. Unfortunately, the presented TEM, XPS and XRD data (Figures S12-S14) after the long-term stability test are not analyzed. For the TEM, an evaluation of the NP size is lacking, as well as an HAADF-STEM image showing the presence and amount of the single atom sites. The authors should provide the same type of data, that they have nicely provided in Figure 1A-D. XPS: An evaluation of the oxidation state (Cu2p with fits and, if possible, Auger data) and chemical composition (Cu/C/N ratio) should be given. Differences, e.g. of the N1s composition change before and after reaction should be discussed. Here, the CO₂RR conditions should be described in the descriptions of these Figures. Similar for all shown data after reaction. Please specify how the regular refreshing of the catalyst could influence the stability and product distribution (L164). This important experimental detail should be noted in the experimental data, and maybe even in the main text as it may be deceptive for the reader of this manuscript. Please discuss and demonstrate that flooding might not alter the catalyst, but that carbonation does and that it can be reversed.

Response: We thank the referee again for the comment. In order to diminish the impact of solution resistance during long-term electroreduction, chronopotentiometry is a common method used for evaluating long-term CO₂RR

activity stability of electrocatalyst in a flow cell (J. Am. Chem. Soc. 2022, 144, 259-269; Angew. Chem. Int. Ed. 2022,61, e202113498; Angew. Chem. Int. Ed. 2022,61, e202110657). The selectivity of C₁, C₂ products and H₂ during the long-term stability test were added and the long-term stability results of the other two catalysts have been presented as Figure S14 in the revised supporting information. R-Cu₁/Cu_{NP} and P-Cu₁/Cu_{NP} showed lower C₂ products selectivity than that of M-Cu₁/Cu_{NP} and no obvious changes in product distribution or potential were observed over the course of the long-term stability tests.

In the revised manuscript, we have discussed them by “The long-term stability test was conducted through chronopotentiometry for 40 h and the electrode was washed, then dried and the electrolyte was refreshed at intervals 5 h to address the issues of flooding and carbonation. Figure 2E and Figure S14 showed that no obvious change was observed over the potential and products selectivity, and the high FE of C₂ products could be held over 70% on M-Cu₁/Cu_{NP} during the electrolysis.” Please see them in page 10 of the revised manuscript.

According to the comment, we have added more characterization over M-Cu₁/Cu_{NP} after long-term CO₂RR experiment and discussed them by “The morphology, valence states and crystal structure of M-Cu₁/Cu_{NP} after 40 h CO₂RR experiment were characterized (Figures S15, S16). TEM image suggested that the size Cu NPs is similar to that before used and the aberration-corrected HAADF-STEM image confirmed that Cu₁ site and Cu NPs still coexisted in M-Cu₁/Cu_{NP}. The Cu LMM Auger spectra show that Cu⁺ and Cu⁰ coexisted in M-Cu₁/Cu_{NP}. Meanwhile, no obvious difference was observed in N 1s XPS spectra before and after CO₂RR experiment and only the peaks corresponding to Cu were observed in XRD pattern after CO₂RR experiment. All these results indicated the excellent stability of the catalyst.” Please see them in Page 10 of the revised manuscript. It is noteworthy that Nafion solution was used in the process fabricating working electrode, which contains C, H, O, S and F elements, thereby, EDS mapping and chemical composition of the catalyst after long-term CO₂RR experiment cannot provide value information.

The CO₂RR stability is affected by catalyst, electrolyte, gas diffusion electrode (GDE) and so on. In flow cell, CO₂RR occurred at the three-phase interface formed by GDE with CO₂ gas channel and hydrophobic layer. CO₂ can react with hydroxide in electrolyte to form carbonate precipitation during CO₂RR process, and the accumulation of carbonate at three-phase interface would block CO₂ gas channel and the gas diffusion electrode suffers from losing hydrophobicity and being flooded over time. Both CO₂ gas channel blockage and flooding would destroy GDE stability, and thus affect the catalytic performance. Meanwhile, the water dissociation, accumulation of liquid products and carbonate formation would change the properties of electrolyte, and also affect the catalyst performance. Therefore, in order to verify the stability of the catalyst, the influence of GDE and electrolyte should be minimized. To issue the problem of gas channel blockage and flooding, the CO₂RR was interrupted every 5 hours and the GDE were removed, washed with deionized water and followed by dryness under N₂ atmosphere. Meanwhile, the fresh electrolyte solution was pumped into flow cell for each interval to keep the electrolyte property stable. The results

of long-term CO₂RR experiment showed that C₂ products selectivity and potential had no obvious change after minimizing effect of GDE and electrolyte, demonstrating the catalyst had high stability. In the revised manuscript, we have supplemented the experiment details in Methods section of the revised manuscript: “The chronopotentiometry method was used for evaluating long-term stability of electrocatalyst in 5 M KOH electrolyte. In order to address the issues of flooding and carbonation accumulation of GDE, the CO₂RR was interrupted every 5 hours and the GDE were removed, washed with deionized water and followed by dryness under N₂ atmosphere. Meanwhile, the electrolyte solution was refreshed for each interval.” Please see them in page 20 of the revised manuscript.

12. KINETIC ISOTOPE EFFECT : The authors provide a decent study on the origin of the hydrogen used for C₂ products. These results could be justified by KIE results of R-Cu₁/Cu_{NP} and Cu-NC (L172). The authors should provide these investigations. It should also be discussed how the hydrogen formation is altered with D₂O experiments. Again, please specify the experimental conditions in Figure 3. (potential/scan rate, electrolyte, CO₂ saturation). The authors also present a small study on the influence of the electrolyte concentration on the formation of ethylene. Please describe why the formation rate instead of the FE or partial current densities are chosen here for the analysis and how it justifies the interpretation. Results for R-Cu₁/Cu_{NP} should also be given. The potential and electrochemical details should be provided in Figure S15.

Response: We thank the referee again for the comment, and we answer the two questions as follows.

(1) The KIE results of R-Cu₁/Cu_{NP} have been added in Figure 3A of the revised manuscript. The KIE value of R-Cu₁/Cu_{NP} was 1.12, which was less than that of P-Cu₁/Cu_{NP} (2.87) and M-Cu₁/Cu_{NP} (1.21), further indicating that H₂O dissociation was accelerated gradually as ratio of Cu₁ to Cu_{NP} increased. Additionally, the change of H₂ formation rate have been displayed as Figure S17 in the revised supporting information. When the H₂O was replaced by D₂O in the electrolyte, an obvious decrease was observed over all the catalysts, while the decreasing degree followed the sequence of P-Cu₁/Cu_{NP} > M-Cu₁/Cu_{NP} > R-Cu₁/Cu_{NP}, suggesting that the increase of Cu single atom content can reduce the influence of isotope effect on H₂ formation rate.

In the revised manuscript, we have discussed them by “The kinetic isotope effect (KIE) of H₂O/D₂O (H/D) experiments were performed to get insights into the role of H₂O dissociation in CO₂-to-C₂ products (Figure 3A). When the H₂O was replaced by D₂O in the electrolyte, the formation rate of the product (for example C₂H₄) decreased over P-Cu₁/Cu_{NP}, M-Cu₁/Cu_{NP} and R-Cu₁/Cu_{NP}, and the level of its decrease was closely related to the content ratio of Cu₁/Cu_{NP}. If the KIE value (defined as the ratio of C₂H₄ formation rates in H₂O and D₂O) closes to 1, H₂O dissociation is not the rate-determining step over the catalyst. The KIE value for R-Cu₁/Cu_{NP}, M-Cu₁/Cu_{NP} and P-Cu₁/Cu_{NP} were 1.12, 1.21 and 2.87, respectively, which indicated that H₂O dissociation was accelerated gradually as the increasing of Cu₁/Cu_{NP} content ratio. These results confirmed that the atomic Cu sites were

responsible for accelerating H₂O dissociation and provided proton to adjacent Cu NPs, thus affecting CO₂-to-C₂ products. Meanwhile, the N-doped carbon matrix catalysts has been reported to favor the migration of proton.³⁰ In addition, the H₂ formation rate in 5 M KOH H₂O solution and D₂O solution was presented in Figure S17. An obvious decrease was observed over all the catalysts, while the decreasing degree followed the sequence of P-Cu₁/Cu_{NP} > M-Cu₁/Cu_{NP} > R-Cu₁/Cu_{NP} suggesting that the increase of Cu single atom content can reduce the influence of isotope effect on H₂ formation rate.” Please see them in page 12 of the revised manuscript. Besides, we have added the experiments conditions in the caption of Figure 3.

(2) The results of the electrolyte concentration influence over R-Cu₁/Cu_{NP} have been added in Figure S18 of the revised supporting information. We have discussed the results in the revised manuscript as “Furthermore, we studied the influence of electrolyte pH on CO₂RR performance over R-Cu₁/Cu_{NP}, M-Cu₁/Cu_{NP} and P-Cu₁/Cu_{NP}. Three different concentrations of KOH aqueous electrolytes, i.e., 0.1M, 3M and 5M, were employed to adjust the pH environment at the electrode/electrolyte interface. The results in Figure S18 show that the FE and formation rate of C₂ products increased with the increasing pH value of electrolyte, suggesting that strong basic local environment favored the C₂ products formation. Although CO₂RR activity of M-Cu₁/Cu_{NP} in low concentration KOH electrolytes was lower than that in 5 M KOH electrolyte, M-Cu₁/Cu_{NP} exhibited higher FE and formation rate of C₂ products than P-Cu₁/Cu_{NP} and R-Cu₁/Cu_{NP} in low concentration KOH electrolytes, which suggested that the cooperative effect of Cu NPs and atomic Cu sites still facilitated C₂ products formation. The ratio of C₂ products formation rate over M-Cu₁/Cu_{NP} to P-Cu₁/Cu_{NP}, i.e., Rate_M/Rate_P, in 5 M KOH (3.4) is higher than those in 3M KOH (3.0) and 0.1 M KOH (2.3). It demonstrated that the role of atomic Cu sites in accelerating the H₂O dissociation process was significantly more pronounced at higher pH values, leading to the enhanced C₂ products formation, even if the dissociation of H₂O in higher pH alkaline electrolyte was a sluggish step.” Please see them in pages 12-13 of the revised manuscript.

Both FE and formation rate were chosen here for the analysis, and the C₂ products FE and formation rate over P-Cu₁/Cu_{NP}, M-Cu₁/Cu_{NP} and R-Cu₁/Cu_{NP} increased with the increasing pH value of electrolyte, suggesting that strong basic local environment favored the C₂ products formation. FE reflects the products selectivity of catalyst, while formation rate reflects products formation activity. The products formation rate is calculated from the products partial current density considering the time parameter, so they reflect similar property of electrocatalyst. However, the products formation rate is more intuitive than the partial current density, so that we used “formation rate”. It is also a commonly used parameter in the literature (*Nat. Commun.* 2022, 13, 1965; *Nat. Commun.* 2019, 10, 892). In the revised supporting information, the potential and electrochemical details were provided in Figure S18.

13. ELECTRORESPONSE & CO Stripping : Discuss why the used Cu foam as support does not take part in the experiment. Discuss the relevance of catalyst loading and ECSA for this experiment. Please provide the results for

the missing catalysts. The showed results indicate intrinsic differences between the catalysts, but could also very likely be just a roughening/loading effect. Please add experimental details also in the experimental section and add a scheme of the device (Figure S17). Similarly, please provide experimental details of the CO stripping voltammetry in the experimental section. Explain, how the CO was adsorbed on the surface and please add the LSV of the double layer without the CO stripping peak. Again, please also provide the results for the missing catalyst. The interpretation of the electroresponse and CO stripping experiments are vague, and need clarification, providing that the "reference catalyst" P-Cu₁/Cu_{NP} was left out. Please set the results in the context of literature.

Response: We thank the referee again for the comment, and we answer the two questions as follows.

(1) In the process of the gas electro-response experiments, the used Cu foam took part in the gas electro-response experiments. We kept the same Cu foam size and catalyst loading for each experiment, thereby the differences in current density shown in the experiment was attributed to the intrinsic property of the catalysts. The results of P-Cu₁/Cu_{NP} and the device scheme have been added in the revised supporting information (Figures S19-S21). Since the experiment is a gas adsorption process on solid surface, the concept of ECSA is not involved. In the revised manuscript, we have discussed them by “We then investigated the capacity adsorption and activation of CO₂ and CO molecules on R-Cu₁/Cu_{NP}, M-Cu₁/Cu_{NP} and P-Cu₁/Cu_{NP} via gas electro-response experiments in a self-designed gas adsorption electroresponse device (Figure S19).³¹ The results in Figure S20 and Figure S21 show that the CO₂ and CO adsorption responses changed in the sequence of R-Cu₁/Cu_{NP} < M-Cu₁/Cu_{NP} < P-Cu₁/Cu_{NP}, suggesting that the adsorption and activation of CO₂ and CO were promoted as the ratio of Cu₁ to Cu_{NP} decreased.” Please see them in page 13 of the revised manuscript.

In addition, the experimental details of gas electro-response experiments have been added in the Methods section of the revised manuscript: “A self-designed gas adsorption electroresponse device (Figure S19) was used to perform gas electroresponse experiments. To prepare the electrode, 5 mg catalyst was suspended in 1 mL isopropanol to form a homogeneous ink. Then the catalyst ink was spread onto the Cu foam with area of 1 × 2 cm² by a micropipette and then dried under room temperature. The as-prepared electrode was put into a sealed container and connected with electrochemical workstation through two electrode system. Before Ar, CO₂ or CO gas was injected into the sealed container, the container was kept in vacuum state by a vacuum pump. Various potentials were applied on the electrode to observed the change of current curve as a function of time under different atmosphere. The adsorption of various gas on the electrode surface would induce the change of current response. Considering the catalyst loading and the size of the Cu foam used in each experiment is the same, the difference of current density under Ar and CO₂ (or CO) atmosphere can reflect the adsorbed capacity of CO₂ (or CO) on the catalyst surface.” Please see them in page 21 of the revised manuscript.

(2) During the CO stripping experiments, the CO adsorption procedure was accomplished by polarizing the working electrode at +0.2 V and bubbling the electrolyte with CO for 10 min. The experimental details of the CO stripping

test have been added in the Methods section of the revised manuscript: “CO stripping experiments were conducted in a single-chamber electrolytic cell with three electrode system through linear sweep voltammetry method. To prepare the working electrode, 1 mg catalyst was suspended in 500 μL isopropanol with 10 μL Nafion D-521 dispersion (5 wt%) to form a homogeneous ink. Then the catalyst ink was spread onto the carbon paper of $1 \times 1 \text{ cm}^2$ in area and then dried under room temperature. Ag/AgCl electrode and graphite rod were used as reference electrode and counter electrode, respectively. The electrolyte was 0.1 M Na_2SO_4 aqueous solution. The CO adsorption procedure was accomplished by polarizing the working electrode at +0.2 V and bubbling the electrolyte with CO for 10 min and subsequently with N_2 for another 10 min. Then the linear sweep voltammetry was conducted from 0.5 to 1.2 V with scan rate of 10 mV s^{-1} .” Please see them in page 21 of the revised manuscript.

According to the comment, the CO stripping result of P-Cu₁/Cu_{NP} has been added in Figure 3B of the revised manuscript and the LSV curve without CO stripping has displayed as Figure S22 in the revised supporting information. In the revised manuscript, we have discussed them by “On the other hand, the electrochemical CO stripping voltammetry tests of R-Cu₁/Cu_{NP}, M-Cu₁/Cu_{NP} and P-Cu₁/Cu_{NP} were also performed to study the CO adsorption ability. Figure 3B shows a peak in the potential range of 0.8-0.9 V for R-Cu₁/Cu_{NP}, M-Cu₁/Cu_{NP} and P-Cu₁/Cu_{NP}. No peak was observed in the LSV curve without CO adsorption (Figure S22). Further according to pervious literature,³²⁻³⁴ the peak at around 0.8-0.9 V can be attributed to the CO stripping peak. Interestingly, the CO stripping peak occurred around 0.89 V for M-Cu₁/Cu_{NP} and P-Cu₁/Cu_{NP}, while it was around 0.82 V for R-Cu₁/Cu_{NP}. The positive shift suggested that M-Cu₁/Cu_{NP} and P-Cu₁/Cu_{NP} had stronger CO binding ability than R-Cu₁/Cu_{NP}, which have higher Cu NP content. The results above indicated that the Cu NPs was beneficial for the adsorption and activation of CO₂ and CO.” Please see them in page 13 of the revised manuscript.

14. OPERANDO XAS : The show results for operando XAS are incomplete and the interpretation is based on weak presented data. The authors should quantify the increase of the Cu valence state for example with Linear-Combination fitting (L194). It is not clear, how long the catalyst was set at -0.6 V before the shown spectrum was measured. It could very likely be that the experiment did not perform as wished. Different (experimental) parameters can play a role, why the catalyst did not reduce completely or oxidizes again. Parameters that could show a working system could be the following: i) current similar to the experimental conditions in the lab ii) similar product detection iii) different potentials to see the evolution of the catalyst. Please discuss. Thus the assumption towards adsorbed CO₂ and H₂O on the basis of the shown data is unfortunately highly overinterpreted. Please show the data of several potentials and justify with the other catalysts e.g. that Cu NPs can reduce completely, and/or that Cu single atoms behave different. L196: The authors should also provide the least-square fittings (L196) and clarify how you attribute/correlate the increased Cu-N/O coordination number to adsorbed CO₂ and H₂O (L198). It is quite hard from EXAFS data to see the local coordination of adsorbed species on the data. In some cases, however, it is evaluable

from XANES data or from XES data.

Response: We thank the referee again for the comment. An optical photo of *in situ* XAS experiment has been included as Figure S23 in the revised supporting information. In order to overcome the condition limitation of *in situ* XAS test and ensure data quality, the flow cell used in the *in situ* XAS experiments was designed differently from that used in CO₂RR activity tests. The differences between the two flow cells included the distance between the working electrode and counter electrode, as well as the site of the electrolyte inlet and outlet, resulting in variations in the measured current. We endeavored to simulate laboratory test conditions as closely as possible. Based on the definitions of *operando/in situ* characterization, we described the experiments as *in situ* XAS experiments in the revised manuscript. During the *in situ* XAS experiments, the spectrum was recorded following the application of a corresponding potential to the electrode for a duration of 600 s. More experiment details have been included in the Methods section of the revised manuscript as “A custom-designed flow cell was used to conducted *in situ* XAS measurements (Figure S23), the gas diffusion electrode loaded with catalyst (1 mg cm⁻²), Ni foam and Hg/HgO electrode were chosen as the working electrode, counter electrode and reference electrode, respectively. The 5 M KOH aqueous solution was used as electrolyte and were recirculated by pump with flow rates of 20 mL min⁻¹. CO₂ gas was continuously supplied to the gas chamber with a flow rate of 40 mL min⁻¹. Data were recorded in fluorescence excitation mode using a Lytle detector. During the *in situ* XAS experiments, the spectrum was recorded following the application of a corresponding potential to the electrode for a duration of 600 s. Beamline of Beijing Synchrotron Radiation Facility (BSRF) and the radiation was monochromatized by a Si (111) double-crystal monochromator. All collected spectra were analyzed using Athena and Artemis program within the IFEFFIT software packages.” Please see them in page 21 of the revised manuscript.

We have reconducted *in situ* XAS experiments over M-Cu₁/Cu_{NP}, and the spectra of XANES and FT EXAFS were presented as Figure S24 in the revised supporting information and Figure 3C in the revised manuscript. The Cu K-edge adsorption spectra did not show any significant difference under different potentials, indicating the stable nature of the average Cu valence state during CO₂RR. Moreover, the peaks attributed to Cu-N and Cu-Cu coordination still existed in FT EXAFS, with no notable changes observed in peak intensity, indicating the stability of the content ratio of Cu₁ to Cu_{NP} during CO₂RR. These results suggested that the structure and content of Cu₁ and Cu_{NP} remained stable during CO₂RR. The *in situ* XAS experiments of P-Cu₁/Cu_{NP} and R-Cu₁/Cu_{NP} were omitted, considering that the difference among the three catalysts is Cu₁ and Cu_{NP} content and the limited XAS testing time. In the revised manuscript, we have discussed the *in situ* XAS experiments by “*In situ* X-ray absorption spectroscopy (XAS) experiments were performed on M-Cu₁/Cu_{NP} to investigate the changes of Cu valence state and structure during CO₂RR (Figure S23). In the XANES spectra (Figure S24), the Cu K-edge adsorption spectra did not show obvious difference under different potentials, indicating that the average Cu valence state kept stable during the reaction. Moreover, the peaks attributed to Cu-N and Cu-Cu coordination still existed in FT EXAFS spectra (Figure

3C), with no notable changes observed in peak intensity, suggesting the stability of the content ratio of Cu₁ to Cu_{NP}. The results of *in situ* XAS experiments showed that the structure and content of Cu₁ and Cu_{NP} remained stable during CO₂RR.” Please see them in pages 13-14 of the revised manuscript.

15. INSITU-SERS and ATR-SEIRAS : The authors should specify the CO stretching and CO rotation in L201 and provide the results of the other two catalysts (L203). It would be intriguing to see how the CO adsorption changes for the other catalysts. What did the authors learn from these experiments? It is well known that CO is an intermediate for CO₂ reduction. Would the CO_{ads} be lacking for the R-Cu₁/Cu_{NP} or the Cu-N-C catalyst? Are differences in the CO onset potential seen that would be relatable to the earlier onset for C₂₊ products for the M-catalyst? The interpretation on the experiments of ATR-SEIRAS are weak without the measurements of the other catalysts. The authors should provide complete data sets of the other catalysts.

Response: We thank the referee again for the comment, and we answer the two questions as follows.

(1) The photograph and scheme of cell used for *in situ* SERS spectroscopy were showed as Figure S25 in the revised supporting information. The *in-situ* SERS spectroscopy of R-Cu₁/Cu_{NP}, M-Cu₁/Cu_{NP} and P-Cu₁/Cu_{NP} were presented in Figures 3D-3F in the revised manuscript. The peaks at round 298, 365 and 2100 cm⁻¹ were assigned to adsorbed CO restricted rotation ($\nu(\text{Cu-CO})$), Cu-CO stretching ($\nu(\text{Cu-CO})$) and C-O stretching of atop *CO ($\nu(\text{CO})$), respectively. The peaks can be observed from -0.3 to -0.8 V over M-Cu₁/Cu_{NP}, while $\nu(\text{Cu-CO})$ and $\nu(\text{CO})$ peaks appeared at -0.4 V and -0.5 V, respectively, on P-Cu₁/Cu_{NP}. This observation could be relatable to the lower onset potential for C₂ products for M-Cu₁/Cu_{NP}. Only CO rotation peaks can be observed on R-Cu₁/Cu_{NP} from -0.4 to -0.8 V, meaning that high ratio of Cu₁ to Cu_{NP} did not facilitate CO₂ reduction. *In situ* SERS of Cu-N-C has been added as Figure S26 in the revised supporting information, and no peaks associated with *CO intermediate were observed on Cu-N-C, suggesting that single atomic Cu was not conducive to the conversion of CO₂ to adsorbed CO. In the revised manuscript, we have discussed them by “*In situ* surface-enhanced Raman spectroscopy (SERS) was employed to reveal the interactions between Cu species and the reaction intermediates during CO₂RR (Figure S25).³⁵⁻³⁸ As shown in the Raman spectra under different potentials (Figures 3D-3F), the peaks assigned to the restricted rotation of adsorbed CO (298 cm⁻¹, $\nu(\text{Cu-CO})$), Cu-CO stretching (365 cm⁻¹, $\nu(\text{Cu-CO})$) and C-O stretching of atop *CO (2000-2100 cm⁻¹, $\nu(\text{CO})$) were observed from -0.3 to -0.8 V over M-Cu₁/Cu_{NP}, while $\nu(\text{Cu-CO})$ and $\nu(\text{CO})$ peaks over P-Cu₁/Cu_{NP} appeared from -0.4 V and -0.5 V, respectively. This observation could be relatable to the lower onset potential for C₂ products for M-Cu₁/Cu_{NP}. Only CO rotation peaks can be observed on R-Cu₁/Cu_{NP} from -0.4 to -0.8 V, meaning poor CO₂ reduction activity. *In situ* SERS was also conducted over Cu-N-C (Figure S26), whereas no peaks associated with *CO intermediate were observed, suggesting that single atomic Cu was not conducive to the conversion of CO₂ to adsorbed CO. The results of *in situ* SERS indicated that CO₂-to-C₂ products proceeded a CO intermediate process, and moderate content ratio of Cu₁ to Cu_{NP} facilitated CO₂

conversion to adsorbed CO.” Please see them in page 14 of the revised manuscript.

(2) The photograph and scheme of cell used for *in situ* ATR-SEIRAS spectroscopy are showed as Figure S27 in the revised supporting information. *In situ* ATR-SEIRAS spectroscopy of R-Cu₁/Cu_{NP}, M-Cu₁/Cu_{NP} and P-Cu₁/Cu_{NP} are presented in Figures 3G-3I in the revised manuscript. The peak attributed to *CO existed on all the three catalysts. However, as the potentials moved to negative value, the *CO peak over M-Cu₁/Cu_{NP} appeared earlier than that of R-Cu₁/Cu_{NP} and P-Cu₁/Cu_{NP}, and the peak intensity was stronger, which indicated that M-Cu₁/Cu_{NP} facilitated CO₂ conversion to adsorbed CO. Additionally, the peak assigned *CHO only appeared over M-Cu₁/Cu_{NP}, and its change trend was similar to that of *CO intermediate, indicating that the *CHO was originated from the hydrogenation of *CO with the assistance of *H. The results of *in situ* ATR-SEIRAS suggested that proper content ratio of Cu₁ to Cu_{NP} is necessary for *CHO formation. In the revised manuscript, we have discussed them by “*In situ* attenuated total reflection-surface-enhanced IR absorption spectroscopy (ATR-SEIRAS) spectra were further collected to trace the evolution of reaction intermediates during CO₂RR from -0.2 to -0.9 V over R-Cu₁/Cu_{NP}, M-Cu₁/Cu_{NP} and P-Cu₁/Cu_{NP} (Figures 3G-3I, Figure S27). The peak at around 2100 cm⁻¹ can be attributed to electrogenerated CO adsorbed (*CO) on catalyst surface,^{39, 40} which did not appear until the applied potential reached -0.5 V on R-Cu₁/Cu_{NP} and P-Cu₁/Cu_{NP}. However, the ν (*CO) peak over M-Cu₁/Cu_{NP} obviously existed from -0.3 to -0.9 V, and the peak intensity stronger than that of R-Cu₁/Cu_{NP} and P-Cu₁/Cu_{NP}, suggesting that M-Cu₁/Cu_{NP} preferred to generate *CO intermediate. Meanwhile, a peak at 1748 cm⁻¹ observed over M-Cu₁/Cu_{NP} from -0.3 to -0.9 V can be ascribed to the *CHO intermediate,^{12, 38, 39} and its change trend was similar to that of *CO intermediate, indicating that the *CHO was originated from the hydrogenation of *CO with the assistance of *H. Nevertheless, the *CHO peak was not observed on R-Cu₁/Cu_{NP} and P-Cu₁/Cu_{NP}, meaning that proper content ratio of Cu₁ to Cu_{NP} is necessary for *CHO formation.” Please see them in pages 14-15 of the revised manuscript.

16. DISCUSSION & SUMMARY : The discussion is weak, providing that the assumptions were mostly made on the basis of just one sample. Mixing Cu NPs and Cu atomic sites require good knowledge and verification of both individual systems. The manuscript has started well by presenting three different samples and analyzing the electrocatalytic data of each quite well. However, some more work has to be done on the spectroscopic site to understand the catalytic systems and to extract a relevant conclusion from it.

Response: We thank the referee again for the comment. As the suggestion of the referee, we have supplemented more data on spectroscopic characterization to further understand the catalytic system and discussed them in the revised manuscript.

For *in situ* surface-enhanced Raman spectroscopy, the spectra of P-Cu₁/Cu_{NP} and R-Cu₁/Cu_{NP} have been added and presented as Figures 3D-3F in the revised manuscript. The peaks associated with *CO intermediate appeared at more positive potential in the spectra of M-Cu₁/Cu_{NP} than in those of P-Cu₁/Cu_{NP} spectra, indicating that an increased

content ratio of Cu₁ to Cu_{NP} facilitated CO₂ conversion to adsorbed CO. However, the spectra of R-Cu₁/Cu_{NP} only showed a CO rotation peak within a narrow potential range and no peaks associated with *CO intermediate were observed in the spectra of Cu-N-C (Figure S26 in the revised supporting information), suggesting that high content of single atomic Cu is not conducive to the conversion of CO₂ to adsorbed CO. Therefore, the results of *in situ* surface-enhanced Raman spectroscopy suggested that CO₂-to-C₂ products proceeded a CO intermediate and moderate ratio of Cu₁ to Cu_{NP} facilitated CO₂ conversion to adsorbed CO.

For *in situ* attenuated total reflection-surface-enhanced IR absorption spectroscopy, the spectra of P-Cu₁/Cu_{NP} and R-Cu₁/Cu_{NP} have been also added and presented as Figures 3G-3I in the revised manuscript. As the potentials shifted to negative, the peak attributed to *CO in M-Cu₁/Cu_{NP} spectra appeared earlier than in those of R-Cu₁/Cu_{NP} and P-Cu₁/Cu_{NP}. Moreover, M-Cu₁/Cu_{NP} showed the highest *CO peak intensity among the three catalysts at the same potentials, which implied that moderate content ratio of Cu₁ to Cu_{NP} facilitated CO₂ conversion to adsorbed CO, as confirmed by *in situ* surface-enhanced Raman spectroscopy results. Additionally, a peak ascribed to *CHO intermediate was observed in the M-Cu₁/Cu_{NP} spectra, while not in the P-Cu₁/Cu_{NP} and R-Cu₁/Cu_{NP} spectra, and its change trend was similar to that of *CO intermediate. These findings suggest that the *CHO intermediate is produced via the hydrogenation of *CO with the assistance of *H, and a proper content ratio of Cu₁ to Cu_{NP} was necessary for *CHO formation.

For *in situ* X-ray adsorption spectroscopy, we have reconducted the characterization for M-Cu₁/Cu_{NP} and displayed the results as Figure 3C in the revised manuscript and Figure S24 in the revised supporting information. No obvious differences were observed in both XANES and EXAFS, which implied the structural stability of single atomic Cu and Cu nanoparticles.

In the revised manuscript, we have discussed the results of *in situ* spectroscopic characterization in detail and summarized them in the Discussion section by “In summary, a series of dual-active sites catalysts with co-loaded Cu NPs and atomic Cu sites on N-doped carbon matrix have been successfully synthesized and evaluated for CO₂ electroreduction. Among them, the M-Cu₁/Cu_{NP} exhibited a C₂ products FE of 75.4% with a corresponding partial current density of 289.2 mA cm⁻² at -0.6 V, as well as remarkable long-term stability. The detailed study revealed that atomic Cu sites could promote H₂O dissociation to provide *H, and the Cu NPs was beneficial for the adsorption and conversion of CO₂. The *in situ* spectroscopic characterization showed that moderate content ratio of atomic Cu sites to Cu NPs facilitated CO₂ conversion to adsorbed CO, resulting in M-Cu₁/Cu_{NP} possessing a lower onset potential for C₂ products. Furthermore, *in situ* ATR-SEIRAS revealed the presence of the intermediate *CHO over M-Cu₁/Cu_{NP}, which is originated from the hydrogenation of *CO with the assistance of *H. DFT calculations demonstrated that the C-C coupling step was promoted through *CHO dimerization reaction on Cu NPs, and the moderate *H coverage facilitated the formation of *CHO. Therefore, the excellent catalytic performance of fabricated dual-active sites catalyst originated from the cooperative effect of atomic Cu sites and Cu NPs. The atomic

Cu sites promoted H₂O dissociation to provide *H, which in turn migrated to Cu NPs and facilitated *CO protonation to form *CHO by modulating the *H coverage on Cu NPs, leading to high activity for C₂ products production. This work puts forward rational concept for promoting conversion CO₂ to C₂ products through modulate the adsorbed hydrogen coverage on Cu-based catalysts. We believe that it will inspire the design of more dual-/multi-active sites catalysts for multi-step reactions.” Please see them in page 18 of the revised manuscript.

17. EXPERIMENTAL : In general, the experimental section lacks several details that are important to follow the experiments. E.g., different types of flow cells exist, a more explicit description should be provided. If a gas-fed CO₂ type flow cell was used, please also add the side from which the gas products were determined and how often gas products were injected into the GC (if an online-GC was used). Please provide a scheme or a reference of the used cell. Please state type and company of membrane and used chemicals, type of the used gas diffusion electrode was used (type, content of PTFE, with or without microporous layer), type of the Ni foam. Please state the total volume of electrolyte (re)circulated, and how big the electrolyte volume in the cell is. The authors may also provide more details of the electrolyte (purity of the water, purity and company of the KOH, possible presaturation with CO₂). Experimental details on the ohmic resistance should be added. The authors may also provide the experimental details the double layer capacitance measurements. Ni foam as counter electrode not the state of the art to be used as counter electrode for CO₂RR, the authors may comment on their choice. The authors may also comment on their choice of and Hg/HgO reference electrode instead of other stable and non-toxic reference electrodes, such as RHE or Ag/AgCl. The authors may provide schemes or references to the used operando and in situ XAS, Raman and ATR-SEIRAS cells and a description the catalysts are exposed to the spectroscopic techniques. The authors may state which definition are the terms operando and in-situ chosen for each technique. The authors may also combine the Methods in the main manuscript and the methods in the Supporting information into one section.

Response: We thank the referee again for the comment. The scheme of flow cell has been added in the revised supporting information, and gas products were collected using a gas bag from CO₂ outlet, then analyzed by using an off-line GC. We have combined the methods in the main manuscript and supporting information into one section and more experiment details have been supplemented in the revised manuscript as “Materials. Copper nitrate hydrate (Cu(NO₃)₂·3H₂O, purity > 99%), guanidine thiocyanate (C₂H₆N₄S, purity > 99%), potassium hydroxide (KOH, purity > 85%), potassium bicarbonate (KHCO₃, purity > 99.5%), sodium sulfate (Na₂SO₄, purity > 99%), 2, 2-dimethyl-2-silapentane-5-sulfonate (DSS, 99%), Deuterium oxide (D₂O, purity > 99.9%), gas diffusion electrode (YLS-30) with 10% PTFE and microporous layer, anion exchange membrane (FAA-3-PK-130) and Ni foam (purity> 99.8%, thickness 0.5 mm) were purchased from Alfa Aesar China Co., Ltd. CO₂ (99.999%), N₂ (99.999%) and 10% H₂/Ar were provided by Beijing Analytical Instrument Company. All materials were used directly without further purification and all the aqueous solutions were prepared by Milli Q water (18.2 MΩ cm, 298 K).

Electrocatalytic CO₂ reduction. To prepared gas diffusion electrode, 1 mg catalyst was suspended in 500 μL isopropanol with 10 μL Nafion D-521 dispersion (5 wt%) to form a homogeneous ink. Then the catalyst ink was spread onto the gas diffusion electrode of 0.5 × 2 cm² in area by a micropipette and then dried under room temperature. All the electrochemical experiments were conducted on the electrochemical workstation (CHI 660E) equipped with a high current amplifier (CHI 680C) and the CO₂RR performance was investigated in flow cell (Figure S7). The as-prepared gas diffusion electrode and Ni foam were used as the cathode and anode, respectively. An anion exchange membrane was used to separate the cathode and anode. Aqueous KOH solution (5 M) was used as the electrolyte solution and the electrolyte volume was 30 mL, and the electrolyte volume in the cell is 0.5 mL. The catholyte solution and anolyte solution were recirculated by two pumps with flow rates of 10 mL min⁻¹ and 30 mL min⁻¹, respectively. Meanwhile, CO₂ gas was continuously supplied to the gas chamber by using a mass flow controller with a flow rate of 40 mL min⁻¹. All the potentials were measured against a Hg/HgO reference electrode and converted to versus RHE with *iR* (80%) compensations, *i* represents the current obtained at corresponding potential, *R* is ohmic resistance of the cell measured by electrochemical workstation:

$$E_{\text{RHE}} = E_{\text{Hg/HgO}} + 0.098 + 0.059 \times \text{pH} - iR \times 80\%$$

The chronopotentiometry method is used for evaluating long-term CO₂RR activity stability of electrocatalyst in 5 M KOH electrolyte. In order to address the issues of flooding and carbonation accumulation of GDE, the CO₂RR was interrupted every 5 hours and the GDE were removed, washed with deionized water and followed by dryness under N₂ atmosphere. Meanwhile, the electrolyte solution was refreshed for each interval.

Product Analysis. The gaseous products were collected using a gas bag and quantified by gas chromatography (GC 7890B). A thermal conductivity detector (TCD) and a flame ionization detector (FID) were used to quantify H₂, CO, and other alkane contents, respectively. The Faradaic efficiency (FE) of gaseous products was calculated by the equation:

$$FE = \frac{znVF}{Q} \times 100$$

Where *z* represents the number of electrons transferred for product formation, *n* is the volume concentration from GC, *V* is the total volume calculated by outlet flow rate, *F* is Faraday constant (96485 C/mol) and the *Q* is the amount of cumulative charge recorded by the electrochemical workstation.

The liquid product was analyzed by ¹H NMR (Bruker Advance III 400 HD spectrometer) in deuterioxide. To accurately integrate the products in NMR analysis, the sodium 2, 2-dimethyl-2-silapentane-5-sulfonate (DSS) was the reference for ethanol and acetic acid, and phenol was the reference for formate. The FE of liquid products was calculated by the equation:

$$FE = \frac{zCVF}{Q} \times 100$$

Where *z* represents the number of electrons transferred for product formation, *C* is the liquid concentration obtained

from NMR, V is electrolyte volume, F is Faraday constant (96485 C/mol) and the Q is the amount of cumulative charge recorded by the electrochemical workstation.” Please see them in pages 19-20 of the revised manuscript.

The oxygen evolution reaction (OER) occurred in anode chamber of flow cell during CO₂RR occurred in cathode chamber. Therefore, the low-cost, earth-abundant, and efficient OER catalyst is necessary for CO₂RR investigation. Although IrO₂ and Pt anode catalysts are high activity for OER, the scarcity and high cost limit their widespread use. Ni foam has the advantage of low cost, easy accessibility and high OER activity in an alkaline environment, which has been widely used as the anode for OER in flow cell with alkaline electrolyte (*Science* 2020, 367, 661-666; *Nat. Energy* 2020, 5, 478-486; *Nat. Commun.* 2021, 12, 5745; *J. Am. Chem. Soc.* 2022, 144, 3039-3049). The selection of reference electrode depends on the pH value of the electrolyte. The Hg/HgO electrode is ideal reference electrode for using in alkaline electrolyte. The Ag/AgCl reference electrode has limited life time in strong alkaline electrolyte. This is caused by hydroxide diffusing into the electrode filling solution, and forming silver hydroxy/oxide species on the silver wire (which becomes brown/blackened), leading to a shift in the electrode potential. For comparison with data reported in the literature, all potentials in the manuscript have been converted to the potentials with respect to RHE.

The photograph or schemes of cells used in *in situ* XAS, Raman and ATR-SEIRAS experiments have been added as Figures S23, S25 and S27 in the revised supporting information. There is a slight difference between *in situ* and *operando* characterization. For *in situ* characterization, the reaction is monitored in real time at the positions relevant to the catalytic operation, not consistent with genuine reaction conditions, such as temperature, pressure. However, *operando* characterization indicates the operation under genuine reaction conditions, where the catalyst structure (active sites and intermediates), as well as activity and selectivity, is measured simultaneously in real time (*Catal. Today*, 2005, 100, 71–77; *ACS Catal.*, 2021, 11, 1136–1178). Based on the above definition, we have used *in situ* to describe XAS, Raman and ATR-SEIRAS characterization in the revised manuscript.

Reviewer 3:

Authors reported the dual-active sites catalyst for electrocatalytic CO₂ reduction reaction. The atomic Cu sites and Cu NPs exhibited an efficient CO₂-to-C₂ reaction. M-Cu₁/Cu_{NP} delivered a high C₂ products FE of 75.4% with a partial current density of 289.2 mA cm⁻² and remarkable long-term stability. Based on the experimental and theoretical studies, the boosted H₂O dissociation and formed *CHO play the crucial role in the CO₂-to-C₂ reaction. However, the description of X-ray and FTIR spectroscopy is not convincing. The reviewer has several questions regarding the mechanistic insights into the proposed reaction model. Please address these issues in detail.

Response: We thank the referee very much for the comment, which help and guide us to improve and polish our work greatly. We have tried our best to answer the questions from the reviewers.

1. Line 92: XPS results show that the valence state of Cu in the catalysts increased in the order of P-Cu₁/Cu_{NP} < M-Cu₁/Cu_{NP} < R-Cu₁/Cu_{NP}. Can authors provide quantitative analysis?

Response: We thank the referee again for the comment. Considering that both atomically dispersed Cu site and Cu nanoparticles existed in the catalysts, XPS was used for qualitative analysis of Cu average valence state. In order to further explain the results of XPS, more detailed analysis has been added in the revised manuscript by “The XPS survey spectra show that the presence of Cu, N, C and O elements in P-Cu₁/Cu_{NP}, M-Cu₁/Cu_{NP} and R-Cu₁/Cu_{NP} (Figure S3), where the existence of O element can be attributed to the adsorbed oxygen on surface. The contents of Cu in P-Cu₁/Cu_{NP}, M-Cu₁/Cu_{NP}, and R-Cu₁/Cu_{NP} were 7.4, 4.3, and 1.9 at%, respectively, and the corresponding N and C content are displayed in Table S1. The Cu 2*p* spectra (Figure 1F) suggest that only Cu 2*p*_{3/2} peak attributed to Cu^{0/+} existed in the spectra, which moved to lower binding energy in the order of R-Cu₁/Cu_{NP} (933.0 eV), M-Cu₁/Cu_{NP} (932.7 eV) and P-Cu₁/Cu_{NP} (932.5 eV), indicating that the valence state of Cu in the catalysts increased gradually. The Cu LMM Auger spectra confirmed that Cu⁺ and Cu⁰ coexisted in P-Cu₁/Cu_{NP} M-Cu₁/Cu_{NP} and R-Cu₁/Cu_{NP} (Figure S4). Considering the valence state of Cu in Cu NP is 0, while that of atomically dispersed Cu site is Cu^{δ+}, the content of Cu₁ site in the catalyst increased in order of P-Cu₁/Cu_{NP} < M-Cu₁/Cu_{NP} < R-Cu₁/Cu_{NP}. Meanwhile, the high resolution XPS N 1*s* spectra of P-Cu₁/Cu_{NP}, M-Cu₁/Cu_{NP}, and R-Cu₁/Cu_{NP} are displayed in Figure S5, all spectra showed obvious Cu-N peak at around 399.2 eV, which suggests that the Cu₁ sites were coordinated by N atoms and other peaks could be attributed to pyridinic N (~398.5 eV), pyrrolic N (~399.9 eV), and graphitic (~401.0 eV), respectively.” Please see them in Page 6 of the revised manuscript.

2. Line 100: “Figure 1F shows that the characteristic peaks of Cu Kβ_{1,3} for Cu₁/Cu_{NP} samples lied between 0 and +2 in the sequence of P-Cu₁/Cu_{NP} < M-Cu₁/Cu_{NP} < R-Cu₁/Cu_{NP}”. In fact, it is difficult to distinguish the position of these characteristic peaks in figure 1F. Authors should present this figure clearly and provide detailed description.

Response: We thank the referee again for the comment. The XES experiments was adopted to identify the coordination atoms of Cu single atoms and Cu valence state. However, the data quality was much lower than that reported in literature due to the limitation of the detector accuracy in XES experiments, which affected the validity of the conclusions. Additionally, according to the results of XPS and XAS, it can be identified that Cu valence state increased as the sequence of P-Cu₁/Cu_{NP} < M-Cu₁/Cu_{NP} < R-Cu₁/Cu_{NP}, and Cu single atoms were coordinated by N atoms. Therefore, we have deleted the XES experiments data, because we have already obtained the same conclusion by the more accurate technique (XPS and XAS).

3. In Figure 3C, the increase of Cu valence state was studied using the absorption energy of white line position of XANES spectra. However, the average Cu valence state of pristine materials was examined using first derivative of XANES spectra (Figure 1H). Please provide the detailed XANES spectra of pristine materials in the supporting

information.

Response: We thank the referee again for the comment. As suggested by the reviewer, the Cu K-edge XANES spectra of R-Cu₁/Cu_{NP}, M-Cu₁/Cu_{NP} and P-Cu₁/Cu_{NP} were presented in the revised manuscript (Figure 1G). The adsorption edge of the catalysts shifted to higher energy in the sequence of P-Cu₁/Cu_{NP} < M-Cu₁/Cu_{NP} < R-Cu₁/Cu_{NP}, and all located between that of Cu foil and CuPc, indicating the average oxidation state of Cu in the catalysts laid between 0 and +2. In the revised manuscript, we have discussed them by “To further determine the chemical state and local coordination environment, X-ray absorption near-edge structure (XANES) and extended X-ray absorption fine structure (EXAFS) measurements were conducted. The Cu K-edge XANES of R-Cu₁/Cu_{NP}, M-Cu₁/Cu_{NP} and P-Cu₁/Cu_{NP} with the reference samples Cu foil and Cu phthalocyanine (CuPc) were presented in Figure 1G, the adsorption edge of all catalysts located between that of Cu foil and CuPc, indicating the average Cu valence state of Cu in the catalysts laid between 0 and +2. Meanwhile, the adsorption edge moved to higher energy in the sequence of P-Cu₁/Cu_{NP} < M-Cu₁/Cu_{NP} < R-Cu₁/Cu_{NP}, confirming that the average Cu valence state increased with the increase of Cu₁/Cu_{NP} ratio. It is in accordance with the XPS results.” Please see them in Page 6-7 of the revised manuscript.

4. In Figure 3C, the feature of Cu K-edge XANES spectrum of M-Cu₁/Cu_{NP} at OCP is very similar with that of Cu foil (Cu₀). Also, the shift in the absorption energy of white line position of M-Cu₁/Cu_{NP} and Cu foil is close. These results suggest that Cu valence state of M-Cu₁/Cu_{NP} could be metallic Cu (or close to Cu₀) at OCP, which is not consistent with the XPS/XAS results in Figure 1H and Figure S3. Authors should add XANES spectrum of pristine M-Cu₁/Cu_{NP} and Cu foil in figure 3C. The detailed discussion should be provided. More information could be obtained using linear combination fitting.

Response: We thank the referee again for the comment. We have reconducted *in situ* XAS experiments over M-Cu₁/Cu_{NP} and the spectra of Cu K-edge XANES are presented with reference sample Cu foil and CuPc as Figure S24 in the revised supporting information. The Cu K-edge XANES of M-Cu₁/Cu_{NP} under different potentials still located between 0 and +2 and no obvious difference was observed under different potentials, suggesting that the Cu average valence state kept stable during CO₂RR. Moreover, the peaks attributed to Cu-N and Cu-Cu coordination still existed in FT EXAFS (Figure 3C in the revised manuscript), with no notable changes observed in peak intensity, indicating the stability of the content ratio of Cu₁ to Cu_{NP} during CO₂RR. These results demonstrated that the structure and content of Cu₁ and Cu_{NP} remained stable during CO₂RR. We discussed them in the revised manuscript as “*In situ* X-ray absorption spectroscopy (XAS) experiments were performed on M-Cu₁/Cu_{NP} to investigate the changes of Cu valence state and structure during CO₂RR (Figure S23). In the XANES spectra (Figure S24), the Cu K-edge adsorption spectra did not show any significant differences under different potentials, indicating that the average Cu valence state kept stable during the reaction. Moreover, the peaks attributed to Cu-N and Cu-Cu

coordination still existed in FT EXAFS spectra (Figure 3C), with no notable change was observed in peak intensity, suggesting the stability of the content ratio of Cu₁ to Cu_{NP}. The results of *in situ* XAS experiments showed that the structure and content of Cu₁ and Cu_{NP} remained stable during CO₂RR.” Please see them in page 13-14 of the revised manuscript.

5. Line 193-195 (Figure 3C), authors claimed that the increase of Cu valence state could be attributed to the adsorption of CO₂ and H₂O molecular on Cu NPs and Cu sites. It is surprising that Cu valence state increases during CO₂RR. This result is not consistent with previous studies. Also, Cu K-edge XAS provides bulk information not surface information. These results and statements are not convincing. If the adsorption of CO₂ and H₂O does cause the oxidation process, authors should provide a series of operando XANES spectra obtained from positive potentials to negative potentials. The shifts in the absorption energy of white line position can be therefore discussed.

Response: We thank the referee again for the comment. We have reconducted *in situ* XAS experiments carefully and made the data more precise. The results of *in situ* XAS experiments showed that no obvious difference was observed in both XANES and EXAFS, which implied the structure stability of single atomic Cu and Cu nanoparticle. There results were in agreement with the previous literature.

In the revised manuscript, we have shown the modified Figure and discussed them by “*In situ* X-ray absorption spectroscopy (XAS) experiments were performed on M-Cu₁/Cu_{NP} to investigate the changes of Cu valence state and structure during CO₂RR (Figure S23). In the XANES spectra (Figure S24), the Cu K-edge adsorption spectra did not show obvious difference under different potentials, indicating that the average Cu valence state kept stable during the reaction. Moreover, the peaks attributed to Cu-N and Cu-Cu coordination still existed in FT EXAFS spectra (Figure 3C), with no notable changes observed in peak intensity, suggesting the stability of the content ratio of Cu₁ to Cu_{NP}. The results of *in situ* XAS experiments showed that the structure and content of Cu₁ and Cu_{NP} remained stable during CO₂RR.” Please see them in page 13-14 of the revised manuscript.

6. Figure S12-S14 show the TEM/XPS/XRD results of M-Cu₁/Cu_{NP} after CO₂RR. Is it after 5 hours or 40 hours? Please provide the information.

Response: We thank the referee again for the comment. The morphology, valence states and crystal structure of M-Cu₁/Cu_{NP} were characterized after 40 h CO₂RR experiment. More characterizations have been adopted to investigate the stability of M-Cu₁/Cu_{NP}. The TEM image, HRTEM image and aberration-corrected HAADF-STEM image of M-Cu₁/Cu_{NP} after 40 h electrolysis were added as Figure S15 in the revised supporting information. The Cu LMM Auger spectrum, XPS spectra of N 1s orbits and XRD pattern of M-Cu₁/Cu_{NP} after 40 h electrolysis were added as Figure S16 in the revised supporting information. In the revised manuscript, we have discussed them as “The morphology, valence states and crystal structure of M-Cu₁/Cu_{NP} after 40 h CO₂RR experiment were characterized

(Figures S15, S16). TEM image suggested the size Cu NPs is similar to that before used and the aberration-corrected HAADF-STEM image confirmed that Cu₁ site and Cu NPs still coexisted in M-Cu₁/Cu_{NP}. The Cu LMM Auger spectra show that Cu⁺ and Cu⁰ coexisted in M-Cu₁/Cu_{NP}. Meanwhile, no obvious difference was observed in N 1s XPS spectra between before and after CO₂RR experiment, and only the peaks corresponding to Cu were observed in XRD spectrum after CO₂RR experiment. All these results indicated the excellent stability of the catalyst.” Please see them in Page 10 of the revised manuscript.

7. The experimental description of *in situ* XAS/Raman/SEIRAS is not clear. Were all *in situ* XAS/Raman/SEIRAS experiments performed in the flow cell with KOH electrolyte? Since cell configuration of traditional SEIRAS measurement is H-type cell, how did authors perform SEIRAS experiment with flow cell? Please provide the cell configuration. The changes in the OH-associated peak should be further addressed. If authors performed the SEIRAS experiment with traditional cell configuration (H-type cell), the pH value of electrolyte changes during CO₂RR. Also, the pH value of electrolyte used in SEIRAS measurement is different from that in XAS/Raman experiments. How did authors correlate the FTIR results with XAS/Raman results?

Response: We thank the referee again for the comment and we answer the two questions as follows.

(1) The photograph or schemes of cells used in *in situ* XAS, Raman and ATR-SEIRAS experiments have been added as Figures S23, S25, S27, respectively, in the revised supporting information. More experiment details have been added in the Methods section of the revised manuscript as “*In situ* XAS measurements. A custom-designed flow cell was used to conduct *in situ* XAS measurements (Figure S23), the gas diffusion electrode loaded with catalyst (1 mg cm⁻²), Ni foam and Hg/HgO electrode were chosen as the working electrode, counter electrode and reference electrode, respectively. The 5 M KOH aqueous solution was used as electrolyte and were recirculated by pump with flow rates of 20 mL min⁻¹. CO₂ gas was continuously supplied to the gas chamber with a flow rate of 40 mL min⁻¹. Data were recorded in fluorescence excitation mode using a Lytle detector. During the *in situ* XAS experiments, the spectrum was recorded following the application of a corresponding potential to the electrode for a duration of 600 s. Beamline of Beijing Synchrotron Radiation Facility (BSRF) and the radiation was monochromatized by a Si (111) double-crystal monochromator. All collected spectra were analyzed using Athena and Artemis program within the IFEFFIT software packages.

In situ Raman measurements. A flow cell with a quartz window by GaossUnion (Tianjin) Photoelectric Technology Company was used to carry out *in situ* Raman measurements using a Horiba LabRAM HR Evolution Raman microscope (Figure S25). A 785 nm laser was used and signals were recorded using a 20 s integration and by averaging two scans. The gas diffusion electrode sprayed with the catalyst was used as working electrode and a graphite rod and a Hg/HgO electrode were used as counter and reference electrodes, respectively. The counter electrode was separated from the working electrode by anion exchange membrane. The 5 M KOH aqueous solution

was used as electrolyte and were recirculated by pump with flow rates of 20 mL min⁻¹. Meanwhile, CO₂ gas was continuously supplied to the gas chamber with a flow rate of 40 mL min⁻¹.

In situ ATR-SEIRAS measurements. The experiments were conducted in a modified electrochemical cell that integrated into a Nicolet 6700 FTIR spectrometer equipped with MCT detector cooled by liquid nitrogen (Figure S27). The catalysts ink was dropped on a germanium ATR crystal deposited with Au film. A platinum wire and a Hg/HgO electrode were used as counter and reference electrodes. Each spectrum was collected with 32 times with a resolution of 4 cm⁻¹. The CO₂-saturated 3 M KOH aqueous solution was used as electrolyte to avoid that the germanium ATR crystal was damaged by strong basic electrolyte. The background spectrum was collected at the potential of +0.1 V vs RHE.” Please see them in page 22-23 of the revised manuscript.

The *in situ* XAS and Raman experiments were performed in flow cell with 5 M KOH aqueous solution as electrolyte. The *in situ* ATR-SEIRAS experiments were performed in single-chamber electrolytic cell. As the strong basic electrolyte would damage the germanium ATR crystal, 3 M KOH aqueous was used as electrolyte during the *in situ* ATR-SEIRAS experiments. It is difficult to distinguish that the OH-associated peak is caused by hydroxide ion in electrolyte or formation from water dissolution, because there were abundant hydroxide ions in the alkaline electrolyte, Therefore, the changes in the OH-associated peak did not investigate in the manuscript.

(2) A mass of bubbles would be generated on the catalyst surface under high current, which affected the signal of *in situ* ATR-SEIRAS. Thus, in order to ensure the signal quality of *in situ* ATR-SEIRAS experiments, the highest current during the experiments kept a low value (< 4 mA). Meanwhile, the time of collection each spectrum was quick. The pH value of electrolyte changed from 10.21 to 10.15 after *in situ* ATR-SEIRAS test. Therefore, the effect of electrolyte pH change on CO₂RR was almost negligible. Although the pH value of electrolyte in *in situ* ATR-SEIRAS experiments was different from that in XAS and Raman experiments, the CO₂RR conditions were all in alkaline electrolyte, and the same intermediates were involved in CO₂RR process during *in situ* ATR-SEIRAS and Raman experiments. Thereby, it has a good correlation between the *in situ* ATR-SEIRAS and Raman/XAS results, which has gradually developed into useful techniques for the mechanism analysis for CO₂RR (*Nat. Catal.* 2020, 3, 75-82; *Nat. Catal.* 2018, 1, 922-934; *J. Am. Chem. Soc.* 2022, 144, 2079-2084; *J. Am. Chem. Soc.* 2020, 142, 9567-9581;).

8. Cu will be oxidized in the alkaline solution. Why did not XAS (Figure 3C)/Raman (Figure 3D) results show the formation of CuO/Cu(+2)?

Response: We thank the referee again for the comment. Previous works have reported that Cu²⁺ species can be easily reduced to Cu⁰ or Cu⁺ species when reduce potentials were applied on catalysts (*J. Am. Chem. Soc.* 2019, 141, 6986–6994; *J. Am. Chem. Soc.* 2022, 144, 2079–2084; *Nat. Commun.* 2022, 13, 4857). Therefore, the average Cu oxidation state was between 0 and +1 from *in situ* XAS analysis, and thus CuO/Cu²⁺ could not be formed under the

electrolysis conditions.

9. Since the S/N ratio of CHO-associated peak is poor in figure 3F, the role of CHO in the CO₂ reduction mechanism is questionable. To prove the proposed reaction model, authors should provide SEIRAS of P-Cu₁/Cu_{NP} and R-Cu₁/Cu_{NP} materials. The potential dependence of CHO-associated peak (at 1548 cm⁻¹) in different materials should be further discussed.

Response: We thank the referee again for the comment. According to the comment, we reconducted the *in situ* ATR-SEIRAS experiments over R-Cu₁/Cu_{NP}, M-Cu₁/Cu_{NP} and P-Cu₁/Cu_{NP}. The peak of *CO existed on all the three catalysts. However, as the potentials moved to negative, the *CO peak over M-Cu₁/Cu_{NP} appeared earlier than that of R-Cu₁/Cu_{NP} and P-Cu₁/Cu_{NP}, and had a stronger intensity. It indicated that M-Cu₁/Cu_{NP} facilitated CO₂ conversion to adsorbed *CO. On the other hand, the peak assigned *CHO only appeared over M-Cu₁/Cu_{NP}, and its change trend was similar to that of *CO intermediate, indicating that the *CHO was originated from the hydrogenation of *CO with the assistance of *H. The results suggested that proper ratio of Cu₁/Cu_{NP} is necessary for *CHO formation.

In the revised manuscript, we have discussed them by “*In situ* attenuated total reflection-surface-enhanced IR absorption spectroscopy (ATR-SEIRAS) spectra were further collected to trace the evolution of reaction intermediates during CO₂RR from -0.2 to -0.9 V over R-Cu₁/Cu_{NP}, M-Cu₁/Cu_{NP} and P-Cu₁/Cu_{NP} (Figures 3G-3I, Figure S27). The peak at around 2100 cm⁻¹ can be attributed to electrogenerated CO adsorbed (*CO) on catalyst surface,^{39, 40} which did not appear until the applied potential reached -0.5 V on R-Cu₁/Cu_{NP} and P-Cu₁/Cu_{NP}. However, the ν(*CO) peak over M-Cu₁/Cu_{NP} obviously existed from -0.3 to -0.9 V and the peak intensity was stronger than that of R-Cu₁/Cu_{NP} and P-Cu₁/Cu_{NP}, suggesting that M-Cu₁/Cu_{NP} preferred to generate *CO intermediate. Meanwhile, a peak at 1748 cm⁻¹ observed over M-Cu₁/Cu_{NP} from -0.3 to -0.9 V can be ascribed to the *CHO intermediate,^{12, 38, 39} and its change trend was similar to that of *CO intermediate, indicating that the *CHO was originated from the hydrogenation of *CO with the assistance of *H. Nevertheless, the *CHO peak was not observed on R-Cu₁/Cu_{NP} and P-Cu₁/Cu_{NP}, meaning that proper content ratio of Cu₁ to Cu_{NP} is necessary for *CHO formation.” Please see them in Page 14-15 of the revised manuscript.

Reviewer 4:

In this work, the authors report the design of catalysts for electrochemical CO₂ reduction to C₂₊ products based on a dual-active site strategy. They combine a water dissociation catalyst (single atom Cu) and a C-C coupling one (Cu nanoparticles) to produce Cu nanoparticles supported on Cu-N-C single atom matrix. They found that this combination enhances CO₂ reduction to C₂₊, achieving a C₂ Faradaic efficiency of 75% and a partial current density of 289 mA with a total FE at -0.6 V vs RHE. While the fabrication of the dual catalysts could be interesting, I do not find the designed catalysts represent a significant advance in the field of electrochemical CO₂ conversion. My

conclusion comes from both fundamental and applied perspectives as I explain below:

Response: We thank the referee very much for the comment, which help and guide us to think about and polish our work greatly. We have tried our best to answer the questions from the reviewers. We hope that we have addressed all of the questions satisfactorily. Many thanks for your time and efforts. We believe our manuscript meets the high standard of the journal, and also hope the reviewer could agree with us.

1. The electrochemical CO₂ conversion performance of the designed catalysts in this work is comparable to most of normal Cu nanoparticles catalysts. In a basic electrolyte, C₂ selectivity of 70-80% in the current density range of 200-500 mA/cm² has been frequently reported using Cu nanoparticle catalysts. A good catalyst for C₂ production should exhibit an FE of at least over 80%.

Response: We thank the referee again for the comment. Although 70%-80% C₂ products selectivity with 200-500 mA cm⁻² C₂ products partial current density has been reported in some papers, the performance was always obtained at high applied potential. For example, Zhang et al. designed a poly(ionic liquid) (PIL)-based Cu(0)-Cu(I) tandem catalyst for producing C₂₊ products, a high C₂₊ products selectivity of 76.1% with 304.2 mA cm⁻² partial current density was obtained at -0.85 V (*Angew. Chem. Int. Ed.* 2022,61, e202110657). Gao et al. engineered the GDL by growing hierarchically structured Cu dendrites consisting of sharp needles and achieved a C₂₊ products partial current density of 255 mA cm⁻² with a C₂₊ products selectivity up to 64% at -0.68 V (*J. Am. Chem. Soc.* 2021, 143, 21, 8011-8021). Zeng et al. synthesized hollow multi-shell structures Cu₂O with tunable shell numbers and obtained a maximum C₂₊ products selectivity of 77.0% with 513.7 mA cm⁻² partial current density at -0.88 V (*Angew. Chem. Int. Ed.* 2021, 61, e202113498). Obtaining good performance at low applied potential is significant for CO₂RR. In this work, the prepared M-Cu₁/Cu_{NP} exhibited a C₂ products Faradaic efficiency (FE_{C2}) of 75.4% with partial current density of C₂ products of 289.2 mA cm⁻² at low applied potential (-0.6 V), which lower than the reported electrocatalysts for CO₂RR to C₂₊ products in literature because the excellent synergistic effect between Cu₁ and Cu_{NP}. In the revised manuscript, we have discussed them by "Compared with reported electrocatalysts for CO₂RR to C₂₊ products in the literature, M-Cu₁/Cu_{NP} exhibited higher CO₂RR to C₂₊ products activity at low applied potential, especially in term of C₂₊ partial current density (Table S7)." Please see them in page 10 of the revised manuscript.

More importantly, this work has shown a dual-active sites catalyst model and proved that inducing an active site for generating moderate *H can efficiently improve the performance for CO₂RR through modifying *H coverage. In order to analyze the get insight into catalytic mechanism in detailed, a series of additional characterizations and experiments, including HAADF-STEM, XPS, in situ ATR-SEIRAS, Raman, XAS, etc., have been supplemented in the revised manuscript. We believe that more catalysts with higher activity for CO₂RR would be fabricated in the future according this catalyst model, through further optimizing the activity of H₂O dissociation site and CO₂

reduction site.

2. The high performance in this work was achieved with very basic electrolyte (5M KOH). Many recent analyses have pointed out the cost related to carbonate formation with alkaline electrolyte and suggest that this condition is not suitable for practical application. In this work, the author designed dual-active site catalysts that work well only in strong alkaline electrolyte (Figure S15). Therefore, it is unclear how the designed catalysts can be used in practical systems.

Response: We thank the referee again for the comment. Alkaline aqueous solution is often used as electrolyte for producing C₂₊ products, which can inhibit hydrogen evolution reaction and facilitate C-C coupling reaction (*Science* 2018, 360, 783-787; *Science* 2020, 367, 661-666; *Nat. Catal.* 2020, 3, 98-106). Carbonate formation is a common problem in alkaline electrolyte and neutral electrolyte. Although acid electrolyte can efficiently avoid carbonate formation, the existing equipment corrosion, metal-based catalyst instability problems also make it not suitable for practical application. Actually, the practical application of CO₂RR is a complex systematic problem, involving catalyst, electrolyte, ion exchange membrane, cell structure, mass transfer, et al. Recently, some works have been reported to try to solve these problems. It can be anticipated that the problem of carbonate formation in basic electrolyte would be solved through other methods and the practical application of CO₂RR would eventually be realized.

Although the CO₂RR activity in low concentration KOH electrolyte was lower than that in 5 M KOH electrolyte, the C₂ products selectivity of M-Cu₁/Cu_{NP} was still higher than that of R-Cu₁/Cu_{NP} and P-Cu₁/Cu_{NP}, suggesting that the cooperative effect of Cu NPs and atomic Cu sites still facilitated C₂ products formation. The dual-active site catalyst model that we proposed was still suitable for low concentration KOH electrolyte. In the revised manuscript, we have discussed them by “Furthermore, we studied the influence of electrolyte pH on CO₂RR performance over R-Cu₁/Cu_{NP}, M-Cu₁/Cu_{NP} and P-Cu₁/Cu_{NP}. Three different concentrations of KOH aqueous electrolytes, i.e., 0.1M, 3M and 5M, were employed to adjust the pH environment at the electrode/electrolyte interface. The results in Figure S18 show that the FE and formation rate of C₂ products increased with the increasing pH value of electrolyte, suggesting that strong basic local environment favored the C₂ products formation. Although CO₂RR activity of M-Cu₁/Cu_{NP} in low concentration KOH electrolytes was lower than that in 5 M KOH electrolyte, M-Cu₁/Cu_{NP} exhibited higher FE and formation rate of C₂ products than P-Cu₁/Cu_{NP} and R-Cu₁/Cu_{NP} in low concentration KOH electrolytes, which suggested that the cooperative effect of Cu NPs and atomic Cu sites still facilitated C₂ products formation. The ratio of C₂ products formation rate over M-Cu₁/Cu_{NP} to P-Cu₁/Cu_{NP}, i.e., Rate_M/Rate_P, in 5 M KOH (3.4) is higher than those in 3M KOH (3.0) and 0.1 M KOH (2.3). It demonstrated that the role of atomic Cu sites in accelerating the H₂O dissociation process was significantly more pronounced at higher pH values, leading to the enhanced C₂ products formation, even if the dissociation of H₂O in higher pH electrolyte was a sluggish step.” Please

see them in page 12-13 of the revised manuscript.

3. The mechanism for enhanced C_{2+} production based on dual-active sites is rather speculative as there are not enough experimental and theoretical evidence to support that claim. For example, the authors claim that Cu-N-C matrix can enhance water dissociation to provide protons for C_2 formation, but they did not provide a clear experimental and theoretical evidence for the enhanced water dissociation. Some obvious questions are: How does the water dissociate on Cu sites? What happens to the OH^- ? How does the H^+ transport to the surface of Cu nanoparticles (in the form of H^+ or H)?

Response: We thank the referee again for the comment. We proved Cu single atoms can enhance water dissociation through controlled experiments and kinetic isotope effect test. The Cu-N-C catalyst was obtained through treating M-Cu₁/Cu_{NP} with sulfuric acid aqueous solution and the results of TEM, XRD, XPS and XAFS confirmed that no Cu nanoparticles or clusters existed, while Cu species in the Cu-N-C existed as Cu single atom form, coordinated by N atoms. The characterization results were presented in the revised supporting information as Figures S11-S12 and we discussed them by “The Cu-N-C was obtained by acid treatment of M-Cu₁/Cu_{NP}: the M-Cu₁/Cu_{NP} was added into 50 mL 1 M sulfuric acid aqueous solution and heated at 80 °C for 48 h, then washed with deionized water several times and dried at 80°C overnight. No obvious nanoparticles were observed in TEM image and a large amount of isolated bright dots recognized as Cu single atoms were observed in aberration-corrected HAADF-STEM image. No diffraction peaks related to crystalline Cu species can be identified in XRD pattern. XPS spectra showed that Cu element existed in Cu-N-C catalyst and Cu⁺ was main specie. More importantly, the EXAFS profiles in the R-space of Cu-N-C suggested that only the peak attributed to Cu-N coordination located at around 1.4 Å, while the Cu-Cu coordination peak at around 2.2 Å disappeared, which confirmed that the Cu species only existed as single atom form. The results of least-squares EXAFS fitting confirmed that the Cu-N coordination numbers in Cu-N-C was 4.0 (Table S2), implying that the atomic Cu species mainly existed as Cu-N₄ structure.” Please see them in the figure description of Figure S11 and Figure S12 in the revised supporting information. The activity test of the Cu-N-C catalyst showed that H₂ was main products in the whole applied potentials range, and H₂O dissociation provided proton source in basic electrolyte, thereby the Cu-N-C catalyst enhanced H₂O dissociation even CO₂ presented in the reaction system.

Additionally, the kinetic isotope effect (KIE) test is an efficient method to justify whether H₂O dissociation is involved in the rate-determining step during CO₂ protonation process or not (*J. Am. Chem. Soc.* 2022, 144, 12807-12815; *Nat. Commun.* 2019, 12, 892; *Adv. Mater.* 2022, 34, 2200057). We have performed KIE test over all three catalysts and displayed the results in Figure 3A in the revised manuscript. If the KIE value (defined as the ratio of C₂H₄ formation rates in H₂O and D₂O) closes to 1, H₂O dissociation is not the rate-determining step over the catalyst. The KIE value of P-Cu₁/Cu_{NP} was calculated to be 2.87, demonstrating that proton transfer was limited. When the

Cu single atoms content was increased, the KIE value over M-Cu₁/Cu_{NP} and R-Cu₁/Cu_{NP} sharply dropped to 1.21 and 1.12 respectively, indicating that H₂O dissociation was accelerated, which is not involved in the rate-determining step. Therefore, the activity test and KIE results revealed that Cu single atoms were responsible for the dissociation of water to accelerate the proton transfer process. In the revised manuscript, we have discussed them by “The kinetic isotope effect (KIE) of H₂O/D₂O (H/D) experiments were performed to get insights into the role of H₂O dissociation in CO₂-to-C₂ products (Figure 3A). When the H₂O was replaced by D₂O in the electrolyte, the formation rate of the product (for example C₂H₄) decreased over P-Cu₁/Cu_{NP}, M-Cu₁/Cu_{NP} and R-Cu₁/Cu_{NP}, and the level of its decrease was closely related to the content ratio of Cu₁/Cu_{NP}. If the KIE value (defined as the ratio of C₂H₄ formation rates in H₂O and D₂O) closes to 1, H₂O dissociation is not the rate-determining step over the catalyst. The KIE value for R-Cu₁/Cu_{NP}, M-Cu₁/Cu_{NP} and P-Cu₁/Cu_{NP} were 1.12, 1.21 and 2.87, respectively, which indicated that H₂O dissociation was accelerated gradually as the increasing of Cu₁/Cu_{NP} content ratio. These results confirmed that the atomic Cu sites were responsible for accelerating H₂O dissociation and provided proton to adjacent Cu NPs, thus affecting CO₂-to-C₂ products. Meanwhile, the N-doped carbon matrix catalysts has been reported to favor the migration of proton.³⁰” Please see them in page 12 of the revised manuscript.

Although the OH-associated peak can be observed on *in situ* Raman spectra, considering that abundant hydroxide ion existed in alkaline electrolyte, it is difficult to distinguish that the OH-associated peak is caused by hydroxide ion in electrolyte or formation from water dissolution. Therefore, the changes in the OH⁻ did not investigate in the manuscript. H⁺ is not stable in the basic environment, which is easily neutralized by OH⁻ to generate H₂O again, so proton would be adsorbed on the surface of catalyst in the form of *H. It has been reported that hydrogen spillover can be occurred on N-doped carbon matrix (*Adv. Mater.* 2022, 34, 2200057), and doping N atoms into carbon matrix would induce C atoms possess positive charge, the positively charged C atoms could theoretically enhance the proton transfer through the repulsive interaction between the generated protons and the positively charged C atoms. Therefore, the proton is transported to adjacent Cu nanoparticles mainly via hydrogen spillover as *H form.

4. The conversion of Cu single atom to Cu nanoparticle/clusters during electrochemical CO₂ reduction are well-documented (*Nat. Commun.* 9, 415 (2018); *Angewandte Chemie*, 58, 15098 (2019)). The presence of Cu single site with operando XAS could be originated from non-electrically conductive sites (*Nat. Commun.* 13, 4190 (2022)). Therefore, it is unclear how the Cu single sites can help CO₂ reduction if they are not stable.

Response: We thank the referee again for the comment. Many literatures have reported that Cu single sites can keep stable during electrochemical CO₂ reduction (*Nat. Commun.* 2023, 14, 474; *Nat. Commun.* 2021, 12, 586; *Nat. Commun.* 2022, 13, 5496). Also, we have reconduted *in situ* XAS experiments over M-Cu₁/Cu_{NP} and the spectra of Cu K-edge XANES was presented with reference sample Cu foil and Cu Pc as Figure S24 in the revised

supporting information. The Cu K-edge XANES of M-Cu₁/Cu_{NP} under different potentials still located between 0 and +2 and no obvious difference was observed, suggesting that the Cu average valence state kept stable during CO₂RR. Moreover, the peaks attributed to Cu-N and Cu-Cu coordination still existed in FT EXAFS (Figure 3C in the revised manuscript), with no notable changes observed in peak intensity, indicating the stability of the content ratio of Cu₁ to Cu_{NP} during CO₂RR. These results suggested that the structure and content of Cu₁ and Cu_{NP} remained stable during CO₂RR.

In the revised manuscript, we have discussed them by “*In situ* X-ray absorption spectroscopy (XAS) experiments were performed on M-Cu₁/Cu_{NP} to investigate the changes of Cu valence state and structure during CO₂RR (Figure S23). In the XANES spectra (Figure S24), the Cu K-edge adsorption spectra did not show obvious difference under different potentials, indicating that the average Cu valence state kept stable during the reaction. Moreover, the peaks attributed to Cu-N and Cu-Cu coordination still existed in FT EXAFS spectra (Figure 3C), with no notable change was observed in peak intensity, suggesting the stability of the content ratio of Cu₁ to Cu_{NP}. The results of in situ XAS experiments showed that the structure and content of Cu₁ and Cu_{NP} remained stable during CO₂RR.” Please see them in page 13-14 of the revised manuscript.

5. Additional controlled samples should be tested to confirm the proposed mechanism. For example, mixing Cu-N-C with a commercially available Cu nanoparticles or other metals such as Ag or Au to see if this mechanism is universal (i.e., for CO₂ conversion to other products).

Response: We thank the referee again for the comment. It has been demonstrated that an optimal distribution and distance between the dual active sites is crucial for efficient active species (*CO or *H) transformation, which significantly impacted the synergistic effect of dual active sites. According to the comment, more controlled samples have been used in the CO₂RR to confirm the proposed mechanism.

The size of Cu nanoparticles in our fabricated dual-active site catalyst was around 4 nm. To exclude the size effect, we have prepared Cu and Ag nanoparticles with around 4 nm diameter according to the literature (Nanoscale 2021, 13, 4835-4844), and measured CO₂ activity of pure Cu nanoparticles and the catalyst fabricated via physically mixing the Cu/Ag nanoparticles with the Cu-N-C. After the Cu nanoparticles were mixed with the Cu-N-C, C₂₊ products selectivity decreased from 32.4% to 19.6%, while that of hydrogen increased from 42.5% to 63.7%. Meanwhile, mixing Ag nanoparticles with Cu-N-C resulted in decreased CO selectivity from 88.2% to 64.1% and an obvious enhancement in hydrogen selectivity. The dual-active site catalysts fabricated through physical mixing Cu-N-C with Cu/Ag nanoparticles were highly heterogeneous, with a considerable distance between the dual active sites that impeded the efficient transformation of *H species, which are dimerized to produce hydrogen. However, the distribution and distance between the dual active sites of the catalysts synthesized using one-step chemical methods can be effectively modified by altering the ratio of precursors used in the synthesis process, thus obtaining

highly effective catalytic performance. Therefore, the mechanism proposed in the manuscript is only applicable to catalysts synthesized using chemical methods with optimal distribution and distance between the dual active sites.

REVIEWER COMMENTS

Reviewer #1 (Remarks to the Author):

The reviewer appreciate the efforts from the authors to address the concerns and questions. The quality of manuscript has greatly improved. Thus, the reviewer recommend its publication in Nature Communications as it is.

Reviewer #2 (Remarks to the Author):

The authors revised their manuscript thoroughly by complementing their presented experimental data with the missing ones. To the reviewer's opinion, data quality, comprehensiveness and completeness has improved significantly and the datasets are now complete to allow any interpretation. Furthermore, the authors have addressed questions and concerns of the reviewers with a lot of detail.

Looking at the complete data sets, the reviewer is concerned that the improved catalytic activity of the M-sample is due to higher electrocatalytic surface area and the majority of the discussion and interpretation could thus be invalid. For example, the signal-to-noise ratio for the Raman data seems to be much lower for the M-sample than the P-sample. As Cu is active for surface enhancement, this might indicate a significant increase in surface area. Also, the CO-stripping voltammetry for the M-sample provides double of the current density compared to the other samples (and compared to the LSV after CO stripping in the supplementary information).

To study the sample loading-dependent catalytic selectivities is a creative way to circumvent the analysis of ECSA, it is however not convincing. First, the optimal loading for the M-sample was studied, but what happened with the other catalysts, where they adapted to the same loading? Unfortunately, it is best practice in the electrocatalysis community to analyse the ECSA method in order to avoid wrong interpretation of the catalytic activity. The reviewer expects a reasonable correction on this topic.

In the following, there are some minor comments:

- Figure 3d-f and 3g-i lacks the choice of electrolyte in the description. (And the CO₂-saturation in case of the IR measurement)
- L121/122: the content ratios are probably reversed, please double check the sentence.

Reviewer #3 (Remarks to the Author):

Although most of the minor questions have been addressed properly, the reviewer has quite substantial concerns in the mechanism study in the present study. The reviewer can not recommend the present study for further publication. These serious concerns should be further addressed.

It is agreed that the CO₂ reduction mechanism and electrochemical performance are different in the H-type cell and flow cell. In the flow cell, the CO₂ reduction reaction is carried out at GDE, which includes the CO₂ (gas)/electrocatalyst (solid)/electrolyte (liquid) reaction. It is a three-phase reaction. In the H-cell, CO₂ gas is dissolved in the electrolyte and forms bicarbonate/carbonate anions, which affect the pH of electrolyte during cycling. Also, mass diffusion is another problem. Thus, the product distribution and electrochemical performance are different in the H-type cell and flow cell, suggesting that the reaction mechanism is different in the H-type cell and flow cell. That is why the authors prepared GDE and perform in situ XAS/Raman spectroscopy in the flow cell in the present study (Figure S23 and figure S25). However, the authors performed in situ FTIR (SEIRAS) in the H-type cell and claimed that the results (mechanistic insights) obtained in the H-type cell and flow cell should be similar (for example: the correlation of SEIRAS/Raman/XAS, the negligible changes in the pH value). In fact, the pH of electrolyte in the SEIRAS (pH= ~10) is not the same with that in the Raman/XAS studies (concentrated KOH). The presence of anions is different in these cells. Moreover, a three-phase reaction is carried out in the flow cell with good mass diffusion. It is not about the different cells. It is about the cell configuration that affects the reaction mechanism.

If the authors insist that the results (mechanistic insights) obtained in the H-type cell and flow cell should be similar. The authors should perform the in situ Raman/XAS measurements/electrochemical performance in the “H-type cell” and provide a detailed comparison. To have a huge impact in the research community, authors should provide the mechanism insights precisely.

The reviewer understands that in situ SEIRAS measurement is available with the H-type cell. Since both Raman and SEIRAS measurements are used to study the formation of surface-adsorbed intermediates such as CO during CO₂ reduction reaction, do in situ SEIRAS results really provide crucial/additional mechanistic insight? Compared to the in situ Raman results, in situ SEIRAS also shows the formation of CHO species during CO₂ reduction reaction. Can this species be obtained in the Raman or ex-situ FTIR measurements (samples are prepared in the flow cell)? If so, the authors can consider to remove the SEIRAS results in the present study.

Reviewer #4 (Remarks to the Author):

The author have addressed most of my concerns.

I still believe that strong basic (KOH) electrolyte is not a suitable electrolyte to evaluate the intrinsic activity and selectivity of the catalysts for CO₂ conversion to multiple carbon products. As the authors shown in table S7, most of the catalysts tested using alkaline electrolyte show C₂⁺ selectivity in the range of 75-90%, which is comparable and higher than the performance reported in this work. Even commercial Cu nanoparticles show high C₂⁺ selectivity with alkaline electrolyte. Based on the

performance of the catalysts in strong alkaline electrolyte, I am not convinced that this represents a significant advance in the field.

Responses to the comments of the reviewers

Reviewer 1:

The reviewer appreciates the efforts from the authors to address the concerns and questions. The quality of manuscript has greatly improved. Thus, the reviewer recommends its publication in Nature Communications as it is.

Response: We thank the reviewer very much for the positive comment.

Reviewer 2:

The authors revised their manuscript thoroughly by complementing their presented experimental data with the missing ones. To the reviewer's opinion, data quality, comprehensiveness and completeness has improved significantly and the datasets are now complete to allow any interpretation. Furthermore, the authors have addressed questions and concerns of the reviewers with a lot of detail.

Response: We thank the referee very much for the comment, and we have carefully revised the manuscript based on the comments.

1. Looking at the complete data sets, the reviewer is concerned that the improved catalytic activity of the M-sample is due to higher electrocatalytic surface area and the majority of the discussion and interpretation could thus be invalid. For example, the signal-to-noise ratio for the Raman data seems to be much lower for the M-sample than the P-sample. As Cu is active for surface enhancement, this might indicate a significant increase in surface area. Also, the CO-stripping voltammetry for the M-sample provides double of the current density compared to the other samples (and compared to the LSV after CO stripping in the supplementary information). To study the sample loading-dependent catalytic selectivity is a creative way to circumvent the analysis of ECSA, it is however not convincing. First, the optimal loading for the M-sample was studied, but what happened with the other catalysts, where they adapted to the same loading? Unfortunately, it is best practice in the electrocatalysis community to analyse the ECSA method in order to avoid wrong interpretation of the catalytic activity. The reviewer expects a reasonable correction on this topic.

Response: We thank the referee again for the comment. The electrochemical active surface area (ECSA) is proportional to double layer capacitance, which measures the capacitive current associated with double-layer charging from the scan-rate dependence of cyclic voltammogram. As suggested by the referee, the ECSA values of the P-Cu₁/Cu_{NP}, M-Cu₁/Cu_{NP} and R-Cu₁/Cu_{NP} have been added as Figure S14 in the revised supporting information, and the results suggested that M-Cu₁/Cu_{NP} exhibited the largest ECSA, consistent with the phenomenon of CO-stripping voltammetry experiments. To exclude the influence of ECSA on catalytic activity, we further normalized

the j_{C_2} based on the ECSA. The results suggested that M-Cu₁/Cu_{NP} still exhibited the highest normalized j_{C_2} among the catalysts, which were displayed in Figure S15 of revised supporting information. Thus, the intrinsic catalytic activity of M-Cu₁/Cu_{NP} is higher than that of P-Cu₁/Cu_{NP} and R-Cu₁/Cu_{NP}.

The catalytic activity and ECSA were tested under the optimal loading of P-Cu₁/Cu_{NP}, M-Cu₁/Cu_{NP} and R-Cu₁/Cu_{NP}, respectively. To further eliminate the potential impact of differences in ECSA on catalytic activity, we increased the loading of P-Cu₁/Cu_{NP} and R-Cu₁/Cu_{NP} in accordance with the ECSA results, ensuring the catalytic activity were measured under similar ECSA conditions. The results were displayed in Figure S16 of the revised supporting information and indicated that M-Cu₁/Cu_{NP} still exhibited largest C₂ products selectivity and partial current density, further confirming that the intrinsic catalytic performance of M-Cu₁/Cu_{NP} is better than that of both P-Cu₁/Cu_{NP} and R-Cu₁/Cu_{NP}. In the revised manuscript, we have discussed them by “We further normalized the j_{C_2} on the basis of the electrochemical active surface area (ECSA), which was measured by the double-layer capacitance (C_{dl}) method (Figure S14). M-Cu₁/Cu_{NP} still exhibited the highest normalized j_{C_2} among the catalysts (Figure S15), confirming that the intrinsic catalytic activity of M-Cu₁/Cu_{NP} is higher than that of P-Cu₁/Cu_{NP} and R-Cu₁/Cu_{NP}. Moreover, we also increased the loading of P-Cu₁/Cu_{NP} and R-Cu₁/Cu_{NP} in accordance with the ECSA results, ensuring the catalytic activity were measured under similar ECSA condition. The results suggested that M-Cu₁/Cu_{NP} still exhibited largest C₂ products selectivity and partial current density (Figure S16), further confirming that the intrinsic catalytic activity of M-Cu₁/Cu_{NP} exceeds that of both P-Cu₁/Cu_{NP} and R-Cu₁/Cu_{NP}.” Please see them in page 10 of the revised manuscript.

We have added the experimental details of double layer capacitance measurements in the Methods section of the revised manuscript as “The electrochemical active surface area is proportional to double layer capacitance, which measures the capacitive current associated with double-layer charging from the scan-rate dependence of cyclic voltammogram. The double layer capacitance was determined in a single-compartment electrolytic cell with 0.5 M KHCO₃ aqueous solution as electrolyte. Ag/AgCl electrode and graphite rod were used as reference electrode and counter electrode, respectively. The scan rates of cyclic voltammogram were 20, 40, 60, 80, 100 mV s⁻¹.” Please see them in pages 21-22 of the revised manuscript.

2. In the following, there are some minor comments:

-Figure 3d-f and 3g-i lacks the choice of electrolyte in the description. (And the CO₂-saturation in case of the IR measurement)

-L121/122: the content ratios are probably reversed, please double check the sentence.

Response: We thank the referee again for the comment. The choice of electrolyte has been added in the description of these figures as “*In situ* surface-enhanced Raman spectra recorded at different applied potentials for (D) P-Cu₁/Cu_{NP}, (E) M-Cu₁/Cu_{NP} and (F) R-Cu₁/Cu_{NP} during CO₂RR in 5 M KOH electrolyte.” in the revised manuscript

(Page 16), and “*In situ* ATR-SEIRAS spectra recorded at different applied potentials for (A) P-Cu₁/Cu_{NP}, (B) M-Cu₁/Cu_{NP} and (C) R-Cu₁/Cu_{NP} during CO₂RR in CO₂-saturated 3 M KOH electrolyte” in Figure S33 of the revised supporting information. Additionally, the content ratios in L121/122 were reversed and we have modified them as “Thus, the content ratio of Cu₁ to Cu_{NP} of P-Cu₁/Cu_{NP}, M-Cu₁/Cu_{NP} and R-Cu₁/Cu_{NP} were 0.05, 0.25 and 0.39, respectively, which is consistent with our catalyst design expectation.” Please see them in Page 6 of the revised manuscript.

Reviewer 3:

Although most of the minor questions have been addressed properly, the reviewer has quite substantial concerns in the mechanism study in the present study. The reviewer cannot recommend the present study for further publication. These serious concerns should be further addressed.

It is agreed that the CO₂ reduction mechanism and electrochemical performance are different in the H-type cell and flow cell. In the flow cell, the CO₂ reduction reaction is carried out at GDE, which includes the CO₂ (gas)/electrocatalyst (solid)/electrolyte (liquid) reaction. It is a three-phase reaction. In the H-cell, CO₂ gas is dissolved in the electrolyte and forms bicarbonate/carbonate anions, which affect the pH of electrolyte during cycling. Also, mass diffusion is another problem. Thus, the product distribution and electrochemical performance are different in the H-type cell and flow cell, suggesting that the reaction mechanism is different in the H-type cell and flow cell. That is why the authors prepared GDE and perform *in situ* XAS/Raman spectroscopy in the flow cell in the present study (Figure S23 and figure S25). However, the authors performed *in situ* FTIR (SEIRAS) in the H-type cell and claimed that the results (mechanistic insights) obtained in the H-type cell and flow cell should be similar (for example: the correlation of SEIRAS/Raman/XAS, the negligible changes in the pH value). In fact, the pH of electrolyte in the SEIRAS (pH= ~10) is not the same with that in the Raman/XAS studies (concentrated KOH). The presence of anions is different in these cells. Moreover, a three-phase reaction is carried out in the flow cell with good mass diffusion. It is not about the different cells. It is about the cell configuration that affects the reaction mechanism.

If the authors insist that the results (mechanistic insights) obtained in the H-type cell and flow cell should be similar. The authors should perform the *in situ* Raman/XAS measurements/electrochemical performance in the “H-type cell” and provide a detailed comparison. To have a huge impact in the research community, authors should provide the mechanism insights precisely.

The reviewer understands that *in situ* SEIRAS measurement is available with the H-type cell. Since both Raman and SEIRAS measurements are used to study the formation of surface-adsorbed intermediates such as CO during CO₂ reduction reaction, do *in situ* SEIRAS results really provide crucial/additional mechanistic insight? Compared to the *in situ* Raman results, *in situ* SEIRAS also shows the formation of CHO species during CO₂ reduction reaction.

Can this species be obtained in the Raman or ex-situ FTIR measurements (samples are prepared in the flow cell)? If so, the authors can consider to remove the SEIRAS results in the present study.

Response: We thank the referee very much for the comment and agree with reviewer's opinion. Due to the inherent limitations of the test method, in situ SEIRAS can only be conducted in the H-type cell in electrocatalytic field. Therefore, in situ SEIRAS cannot directly verify the reaction mechanism in flow cell. In order to elucidate the mechanism with higher precision, we have performed online differential electrochemical mass spectrometry (DEMS) to monitor the intermediates during the reaction process. To maintain experimental conditions closely resembling those of the actual reaction, the online DEMS experiments were conducted in a flow cell equipped with a gas diffusion electrode and using 5 M KOH as the electrolyte. The scheme of the cell used for online DEMS is displayed in Figure S30 of the revised supporting information. As depicted in Figure S31 of the revised supporting information, the m/z signal of 29, corresponding to CHO, can be detected during five continuous cycles at -0.6 V vs. RHE for M-Cu₁/Cu_{NP}, while it was absent in both P-Cu₁/Cu_{NP} and R-Cu₁/Cu_{NP}. The results of online DEMS demonstrated that the proper content ratio of Cu₁ to Cu_{NP} is necessary for CHO formation. We have discussed them in the revised manuscript as "The online differential electrochemical mass spectrometry (DEMS) was conducted to further detected intermediate (Figure S30). During five continuous cycles at -0.6 V vs. RHE, the m/z signal of 29 that correspond to CHO could be detected over M-Cu₁/Cu_{NP}, while it was not detected over P-Cu₁/Cu_{NP} and R-Cu₁/Cu_{NP} (Figure S31). The results of online DEMS demonstrated that the crucial role of the proper Cu₁ to Cu_{NP} content ratio in CHO formation." Please see them in page 15 of the revised manuscript.

The in situ SEIRAS results can only be used as the supplementary result to confirm the intermediates (*Nat. Catal.* 2020, 3, 478-487; *J. Am. Chem. Soc.* 2022, 144, 14936-14944; *Nat. Commun.* 2022, 13, 3754). Therefore, according to the comment of the referee, we have removed them into the supporting information.

Reviewer 4:

The author has addressed most of my concerns. I still believe that strong basic (KOH) electrolyte is not a suitable electrolyte to evaluate the intrinsic activity and selectivity of the catalysts for CO₂ conversion to multiple carbon products. As the authors shown in table S7, most of the catalysts tested using alkaline electrolyte show C₂₊ selectivity in the range of 75-90%, which is comparable and higher than the performance reported in this work. Even commercial Cu nanoparticles show high C₂₊ selectivity with alkaline electrolyte. Based on the performance of the catalysts in strong alkaline electrolyte, I am not convinced that this represents a significant advance in the field.

Response: We thank the referee very much for the comment. We would like to emphasize the significance and novelty of our work briefly. It is known that design of novel catalysts and explore the reaction mechanism for CO₂RR is a very interesting topic, and many important issues need to be studied. In this study, we present a novel dual-active site catalyst model comprising of atomic Cu sites and Cu NPs, and have proved that the substantial enhancement of

CO₂RR performance through the synergistic effect between atomic Cu sites and Cu NPs. Meanwhile, the synergistic effect of atomic Cu sites and Cu NPs is also suitable for low concentration KOH electrolyte. We have combined experimental and theoretical studies to explore the reaction mechanism. It was revealed that Cu NPs facilitated the C-C coupling step through *CHO dimerization reaction to produce C₂ products, while the atomic Cu sites boosted H₂O dissociation to form *H. The generated *H migrated to Cu NPs and modulated the *H coverage on Cu NPs, and thus promoted *CO protonation pathway toward *CHO. Therefore, the synergistic effect of atomic Cu sites and Cu NPs resulted in the excellent performance of the catalyst. We believe that the catalyst model and mechanism will inspire the design of more dual-/multi-active sites catalysts for multi-step reactions.

As above, we believe our manuscript meets the high standard of the journal, and also hope the reviewer could agree with us.

REVIEWER COMMENTS

Reviewer #2 (Remarks to the Author):

The authors have addressed all of the reviewers comments, who now recommends the publication in Nature Communications.

Reviewer #3 (Remarks to the Author):

The authors have addressed most of the concerns. In the revised manuscript, the formation of CHO intermediates is studied using DEMS. Based on the authors' proposed reaction model, CHO is the important "surface-adsorbed intermediates" for further reduction process. However, the scheme of cell used for online DEMS measurements shows that the surface-adsorbed species can not be observed using DEMS. (Figure S30) Why can CHO be detected using DEMS? Do authors claim that CHO species will leave the surface during the further reduction process? The DEMS (for solution species) and FTIR/Raman (for surface-adsorbed species) results are contradictory.

Responses to the comments of the reviewers

Reviewer 2:

The authors have addressed all of the reviewers' comments, who now recommends the publication in Nature Communications.

Response: We thank the reviewer very much for the positive comment.

Reviewer 3:

The authors have addressed most of the concerns. In the revised manuscript, the formation of CHO intermediates is studied using DEMS. Based on the authors' proposed reaction model, CHO is the important "surface-adsorbed intermediates" for further reduction process. However, the scheme of cell used for online DEMS measurements shows that the surface-adsorbed species cannot be observed using DEMS. (Figure S30) Why can CHO be detected using DEMS? Do authors claim that CHO species will leave the surface during the further reduction process? The DEMS (for solution species) and FTIR/Raman (for surface-adsorbed species) results are contradictory.

Response: We thank the referee very much for the comment. The results of online DEMS and in situ FTIR are not contradictory. The online DEMS can provide the signals for the volatile intermediates generated during electrolysis, which is one of the important tools for studying electrode reaction mechanisms (*Chem. Soc. Rev.*, 2021, 50, 6720; *Nat. Catal.*, 2023, 6, 402). Online DEMS relies on the pressure difference in a vacuum as the driving force to extract volatile intermediates and products generated from electrode reactions on the electrode surface through a water-blocking permeable membrane (near the working electrode) into the mass spectrometer within milliseconds, enabling the detection of intermediates and products with high sensitivity and high temporal resolution. Therefore, it is suitable for detecting the intermediates during CO₂ reduction, such as the CHO species.

In the revised manuscript, we have added the illustration for online DEMS as "Online differential electrochemical mass spectrometry (DEMS) was conducted to investigate reaction mechanisms (Figure S30), which extracts volatile intermediates and products generated on the electrode surface into the mass spectrometer within milliseconds, utilizing the pressure difference in a vacuum as the driving force.^{39, 40}". Please see them in page 15 of the revised manuscript.

REVIEWERS' COMMENTS

Reviewer #3 (Remarks to the Author):

Authors have addressed the questions properly. I would like to recommend this study for further publication.

Responses to the comments of the reviewers

Reviewer 3:

Authors have addressed the questions properly. I would like to recommend this study for further publication.

Response: We thank the reviewer very much for the positive comment.